# Variational Flow Maps:
# Make Some Noise for One-Step Conditional Generation

Abbas Mammadov [* 1]   So Takao [* 2 3]   Bohan Chen [2]   Ricardo Baptista [4]   Morteza Mardani [5]
Yee Whye Teh [1]   Julius Berner [5]

## Abstract

Flow maps enable high-quality image generation in a single forward pass. However, unlike iterative diffusion models, their lack of an explicit sampling trajectory impedes incorporating external constraints for conditional generation and solving inverse problems. We put forth *Variational Flow Maps*, a framework for conditional sampling that shifts the perspective of conditioning from "guiding a sampling path", to that of "learning the proper initial noise". Specifically, given an observation, we seek to learn a *noise adapter model* that outputs a noise distribution, so that after mapping to the data space via flow map, the samples respect the observation and data prior. To this end, we develop a principled variational objective that jointly trains the noise adapter and the flow map, improving noise-data alignment, such that sampling from complex data posterior is achieved with a simple adapter. Experiments on various inverse problems show that VFMs produce well-calibrated conditional samples in a single (or few) steps. For ImageNet, VFM attains competitive fidelity while accelerating the sampling by orders of magnitude compared to alternative iterative diffusion/flow models. Code is available at https://github.com/abbasmammadov/VFM .

## 1. Introduction

Diffusion and flow-based methods have emerged as the dominant paradigm for high-fidelity generative modeling, achieving state-of-the-art results across images, audio, and video (Ho et al., 2020; Song & Ermon, 2020; Sohl-Dickstein et al., 2015; Karras et al., 2022; Lipman et al., 2022; Liu et al., 2022). These methods can be understood from the unified perspective of interpolating between two distributions; a simple noise distribution and a complex data distribution, and learning dynamics based on ordinary or stochastic differential equations (ODE/SDEs) that transport one to the other (Albergo et al., 2023). However, these share a fundamental limitation that generating a single sample requires dozens to hundreds of sequential function evaluations, creating high computational cost for real-time applications.

To address this issue, recent research have sought to dramatically reduce this sampling cost. Consistency models (Song et al., 2023b), for example, learn to map any point on the flow trajectory directly to the corresponding clean data, enabling few-step generation. Despite their promise, consistency models often suffer from training instabilities and frequently require re-noising steps for multi-step sampling to correct the drift trajectory, complicating the inference process (Geng et al., 2024). Flow maps (Boffi et al., 2025b;a) offer an alternative framework that seeks to learn ODE flows directly, by training on the mathematical structure of such flows. For example, the state-of-the-art Mean Flow model (Geng et al., 2025) presents a particular parameterisation of flow maps based on *average velocities*, and trained on the so-called Eulerian condition satisfied by ODE flows.

While flow maps excel at unconditional few-steps generation, many applications require *conditional* generation to produce samples that satisfy external constraints. Inverse problems provide a canonical example: given a degraded observation $y = A(x) + \varepsilon$ (e.g., a blurred, masked, or noisy image), we seek to recover plausible original signals $x$ consistent with both the observation and our learned prior $p(x)$. Iterative generative models naturally accommodate such conditioning through *guidance* mechanisms (Chung et al., 2022; 2024; Kawar et al., 2022; Song et al., 2023a), where the trajectory is iteratively nudged toward the conditional target. Flow maps, despite their efficiency, lack this iterative refinement mechanism: once the noise vector $z$ is chosen, the generated sample $z \mapsto x$ is fixed; there is no intermediate state to guide, nor a trajectory to steer, hence there is no

---
[*]Equal contribution  [1]Department of Statistics, University of Oxford [2]California Institute of Technology [3]PhysicsX [4]University of Toronto [5]NVIDIA. Correspondence to: Abbas Mammadov <abbas.mammadov@stats.ox.ac.uk>.

*Proceedings of the 43rd International Conference on Machine Learning*, Seoul, South Korea. PMLR 306, 2026. Copyright 2026 by the author(s).

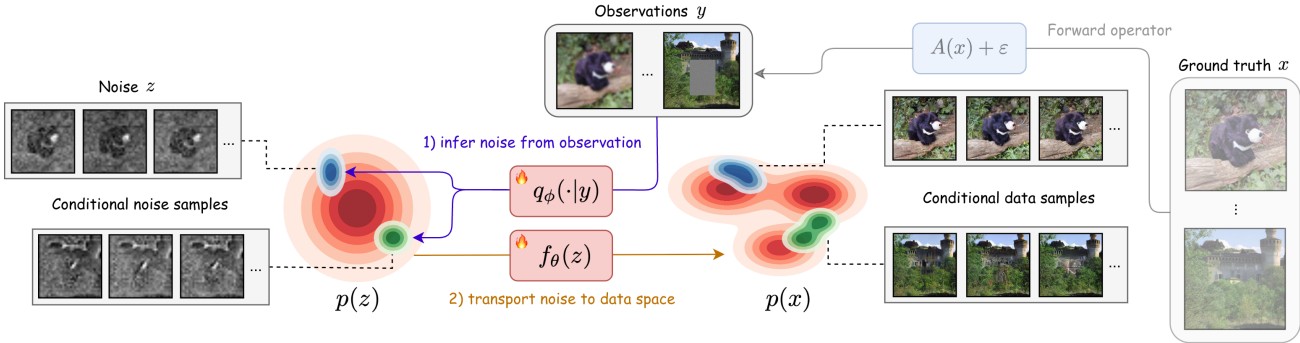

*Figure 1.* One-step conditional generation with Variational Flow Maps (VFM). Given an observation $y$, VFM learns a noise adapter network $q_\phi(z|y)$, which approximates the noise space posterior $p(z|y)$ via amortized variational inference. Conditional noise samples $z \sim q_\phi(z|y)$ are then mapped to data space in a single step via a learned flow map $x = f_\theta(z)$, producing conditional samples that approximate $p(x|y)$. In VFM, the networks $q_\phi$ and $f_\theta$ are trained jointly by extending the variational autoencoder framework to learn the correspondence between the triplet $(x, y, z)$. By jointly training, $f_\theta$ learns to compensate for the simple Gaussian assumption on $q_\phi$.

opportunity to incorporate measurement information during generation. This "guidance gap" has limited flow maps to unconditional settings, leaving their potential for conditional generation largely unexplored. Alternatively, directly training a conditional flow map is possible. However, handling all conditioning modalities simultaneously (e.g. class labels, text prompts, inpainting masks, etc.) requires entangling all these modalities into a single generator, which can be data-hungry and requires substantial architectural modifications. Thus a more modular approach that decouples the backbone flow map from a separate component that handles the inverse problem, is more desirable.

To address these problems, we introduce *Variational Flow Maps* (VFMs), a framework for conditional sampling that is compatible with one/few-step generation using flow maps. Our approach is based on the following perspective: rather than steer the generation process itself, we can *find the noise $z$ to generate from*, as each $z$ deterministically maps to a data $x = f_\theta(z)$ (see Figure 1). Specifically, given an observation $y$, we seek to produce a distribution of $z$'s, such that each $x = f_\theta(z)$ is a candidate data that produced $y$. Formulating this as a Bayesian inverse problem, we can derive a principled variational training objective to jointly learn the flow map $f_\theta$ and a *noise adapter model $q_\phi$* that produces appropriate noise $z$ from observations $y$.

We note the resemblance to variational autoencoders (VAEs) (Kingma & Welling, 2013), where $q_\phi$ plays the role of an encoder that takes $y$ to a latent $z$, and $f_\theta$ acts as a decoder from $z$ to data $x$. Our key innovation is in learning the alignment of all three variables $(x, y, z)$ *simultaneously*, allowing updates to $q_\phi$ to reshape the noise-to-data coupling by $f_\theta$ and vice versa. Notably, we observe that joint training can compensate for limited adapter expressivity by learning a noise-to-data coupling that makes the conditional posterior easier to represent in latent space.

Altogether, our contributions can be summarized as follows:

- We introduce Variational Flow Maps (VFMs), a new paradigm enabling **one and few-step conditional generation** with flow maps by learning an observation-dependent noise sampler.
- We derive a **principled variational objective** for joint adapter/flow map training, linking the mean flow loss to likelihood bounds.
- We demonstrate empirically and theoretically that joint training yields **better noise-data coupling** to fit complex posteriors in data space using **simple** variational posteriors in noise space.
- We extend the framework to **general reward alignment**, introducing a fast and scalable method for fine-tuning pre-trained flow maps to sample from reward-tilted distributions in a single step.

## 2. Background

### 2.1. Flow-based Generative Models and Flow Maps

Flow-based generative models learn to transport samples from a prior distribution $p_1(z) = \mathcal{N}(0, I)$ to the data distribution $p_0(x) = p_{\text{data}}(x)$ via an ODE:

$$\frac{dx_t}{dt} = v_t(x_t), \quad t \in [0, 1], \tag{1}$$

where $v_t$ is a time-dependent velocity field. Flow matching (Lipman et al., 2022; Liu et al., 2022; Albergo et al., 2023) provides a training objective to learn $v_t$: given $x_0 \sim p_{\text{data}}$ and $x_1 \sim \mathcal{N}(0, I)$, we construct a linear interpolant $x_t = (1 - t)x_0 + tx_1$ with conditional velocity $v_t = x_1 - x_0$. Then, $v_\theta(x_t, t) \approx v_t(x_t)$ is trained via:

$$\mathcal{L}_{\text{FM}}(\theta) = \mathbb{E}_{x_0, x_1, t} \left[ \|v_\theta(x_t, t) - (x_1 - x_0)\|^2 \right]. \tag{2}$$

At inference time, samples are generated by integrating the ODE backwards from $t = 1$ to $t = 0$, typically requiring

50–250 function evaluations.

To accelerate sample generation, *flow maps* (Boffi et al., 2025b;a) directly learn the solution operator of the ODE, instead of the instantaneous velocity $v_t$. Denoting by $\phi_{t,s} : x_t \mapsto x_s$ the backward flow of the ODE, the *two-time flow map* $f_\theta(x_t, s, t)$ learns to approximate $\phi_{t,s}(x_t)$ for any $0 \leq s < t \leq 1$. This enables generation with an arbitrary number of steps chosen post-training, e.g. a single evaluation $f_\theta(x_1, 0, 1)$ produces a one-step sample, while intermediate evaluations can be composed for multi-step refinement.

One such approach to learn flow maps is *mean flows* (Geng et al., 2025), which introduce the *average velocity* as an alternative characterization:

$$u(x_t, r, t) := \frac{1}{t-r} \int_r^t v_s(\phi_{t,s}(x_t)) \, ds. \qquad (3)$$

The average velocity satisfies $x_r = x_t - (t-r) \cdot u(x_t, r, t)$, enabling one-step generation via $x_0 = x_1 - u(x_1, 0, 1)$. Thus the corresponding flow map is given by $f_\theta(x_t, r, t) = x_t - (t-r) \cdot u_\theta(x_t, r, t)$. For simplicity, we denote the one-step flow map as $f_\theta(z) := z - u_\theta(z, 0, 1)$, mapping noise $z \sim \mathcal{N}(0, I)$ directly to data $x = f_\theta(z)$.

## 2.2. Inverse Problems

Inverse problem seeks to recover an unknown signal $x \in \mathbb{R}^d$ from noisy observations, given by

$$y = A(x) + \varepsilon, \quad \varepsilon \sim \mathcal{N}(0, \sigma^2 I), \qquad (4)$$

where $A : \mathbb{R}^d \to \mathbb{R}^m$ is a known forward operator and $\sigma > 0$ is the noise level. Given a prior $p(x)$ over signals, the Bayesian formulation seeks the posterior distribution:

$$p(x|y) \propto \exp\left(-\frac{\|y - A(x)\|^2}{2\sigma^2}\right) p(x). \qquad (5)$$

When $p(x)$ is defined implicitly by a generative model, guidance-based methods (Chung et al., 2024; Song et al., 2023a) approximate posterior sampling by incorporating likelihood gradients $\nabla_x \log p(y|x)$ at each denoising step. While effective, these methods inherently require iterative refinement and cannot be applied to one-step flow maps.

## 2.3. Variational Inference and Data Amortization

Variational inference seeks to approximate an intractable posterior $p(z|x)$ with a tractable disribution $q(z|x)$ by minimizing the Kullback-Leibler (KL) divergence:

$$\mathrm{KL}(q(z|x)\|p(z|x)) := \mathbb{E}_{q_\phi}[\log q(z|x) - \log p(z|x)]. \qquad (6)$$

Extending this, amortized inference uses a neural network to directly predict the variational distribution from the conditioning variable $x$, rather than optimizing separately for each

instance. For example, if we choose the variational family to be Gaussians with diagonal covariance, then amortized inference learns a neural network $x \mapsto (\mu_\phi(x), \sigma_\phi(x))$ with parameter $\phi$, such that $q_\phi(z|x) = \mathcal{N}(z|\mu_\phi(x), \mathtt{diag}(\sigma_\phi^2(x)))$ is close to $p(z|x)$ under the KL divergence.

A prototypical example is the *Variational Autoencoder (VAE)* (Kingma & Welling, 2013), which learns both an encoder $q_\phi(z|x)$ and a decoder $p_\theta(x|z)$ by optimizing the VAE objective $\mathcal{L}_{\mathrm{VAE}}(\theta, \phi) = \mathbb{E}_{p(x)}[\ell(\theta, \phi; x)]$, where

$$\ell(\theta, \phi; x) := -\mathbb{E}_{q_\phi(z|x)}[\log p_\theta(x|z)] + \mathrm{KL}(q_\phi(z|x)\|p(z)), \qquad (7)$$

is the negative evidence lower bound (ELBO), yielding $q_\phi(z|x) \approx p_\theta(z|x) \propto p_\theta(x|z)p(z)$ for any $x \sim p(x)$. Probabilistically, the VAE objective can be derived from the KL divergence between two representations of the *joint distribution* of $(x, z)$, i.e., $\mathrm{KL}(q_\phi(z, x)\|p_\theta(z, x))$, where $q_\phi(z, x) = q_\phi(z|x)p(x)$ and $p_\theta(z, x) = p_\theta(x|z)p(z)$. This perspective will be useful in the derivation of our loss later.

# 3. Variational Flow Maps (VFMs)

Our proposed method for one-step conditional generation, which we term *Variational Flow Maps (VFMs)*, is based on reformulating the inverse problem (5) in noise space. To motivate our methodology, we begin with a simple "strawman" approach that is intuitively sound but ultimately insufficient for our task: Let $x = f_\theta(z)$ denote a pretrained flow map. Then the posterior over latent noise variables induced by the inverse problem can be written as

$$p(z|y) \propto \exp\left(-\frac{\|y - A(f_\theta(z))\|^2}{2\sigma^2}\right) p(z). \qquad (8)$$

Although the posterior (8) is intractable, we can approximate it in the same spirit as VAEs. In particular, introducing a variational posterior $q_\phi(z|y) \approx p(z|y)$, we minimize the objective $\mathcal{L}_{\mathrm{VAE}}(\theta, \phi) = \mathbb{E}_{p(y)}[\ell(\theta, \phi; y)]$, where,

$$\ell(\theta, \phi; y) := -\mathbb{E}_{q_\phi(z|y)}[\log p_\theta(y|z)] + \mathrm{KL}(q_\phi(z|y)\|p(z)), \qquad (9)$$

and $p_\theta(y|z) := \mathcal{N}(y|A(f_\theta(z)), \sigma^2 I)$, the likelihood in noise space. A key advantage of working in the noise space rather than the original data space is that the noise prior $p(z)$ is simple and tractable (commonly $\mathcal{N}(0, I)$, which we assume hereafter). Thus, imposing a conjugate variational posterior, such as $q_\phi(z|y) = \mathcal{N}(z|\mu_\phi(y), \mathtt{diag}(\sigma_\phi^2(y)))$, makes the computation of the KL term in (9) tractable.

However, the objective (9) has two major limitations in our setting. First, it does not impose structural properties of flow maps, such as the semi-group property (Boffi et al., 2025a), known to be crucial for learning said maps. Second, when the flow map $f_\theta$ is pretrained and held fixed, a Gaussian variational posterior $q_\phi(z|y)$ may not be expressive enough to approximate the true posterior $p(z|y)$ accurately.

Motivated by this observation, we pursue training the parameters $\theta$ and $\phi$ *jointly*. By adapting the map $f_\theta : z \mapsto x$ alongside learning the variational posterior $q_\phi$, we can compensate for the limited expressibility of $q_\phi(z|y)$ by reshaping the correspondence between noise and data. In the next section, we formalize this idea by deriving a modified objective that enables joint training of $(\theta, \phi)$ while explicitly incorporating additional structural constraints to the flow.

## 3.1. Joint Training of the Flow Map and Noise Adapter

We now propose a joint training strategy that simultaneously aligns the data variable $x$, the observation $y$, and the latent noise variable $z$. Following the probabilistic perspective underlying VAEs (see Section 2.3), we achieve this by matching the following two factorizations of $p(x, y, z)$:

$$q_\phi(z|y)p(y|x)p(x) \approx p_\theta(x, y|z)p(z). \quad (10)$$

For simplicity, we assume a Gaussian decoder of the form

$$p_\theta(x, y|z) = \mathcal{N}(x|f_\theta(z), \tau^2 I) \, \mathcal{N}(y|A(f_\theta(z)), \sigma^2 I), \quad (11)$$

where we introduce a new hyperparameter $\tau > 0$ that relaxes the correspondence between $x$ and $z$. Taking the KL divergence between the two representations in (10) yields

$$\mathrm{KL}(q_\phi(z|y)p(y|x)p(x) \, || \, p_\theta(x, y|z)p(z)) \quad (12)$$
$$\leq \frac{1}{2\tau^2}\mathcal{L}_{\mathrm{data}}(\theta, \phi) + \frac{1}{2\sigma^2}\mathcal{L}_{\mathrm{obs}}(\theta, \phi) + \mathcal{L}_{\mathrm{KL}}(\phi),$$

(see Appendix A.1 for details), where

$$\mathcal{L}_{\mathrm{data}}(\theta, \phi) = \mathbb{E}_{q_\phi(z|y)p(y|x)p(x)} \left[ \|x - f_\theta(z)\|^2 \right], \quad (13)$$
$$\mathcal{L}_{\mathrm{obs}}(\theta, \phi) = \mathbb{E}_{q_\phi(z|y)p(y)} \left[ \|y - A(f_\theta(z))\|^2 \right], \quad (14)$$
$$\mathcal{L}_{\mathrm{KL}}(\phi) = \mathbb{E}_{p(y)} \left[ \mathrm{KL} \left( q_\phi(z|y) \, || \, p(z) \right) \right]. \quad (15)$$

We note that relative to (9), this formulation gives rise to an additional term $\mathcal{L}_{\mathrm{data}}(\theta, \phi)$ that measures closeness of the reconstructed state $f_\theta(z)$ and the ground-truth data $x$, where noise $z$ is drawn from the noise adapter $q_\phi(z|y)$, with observation $y$ taken from $x$. This term couples the adapter model and flow map more tightly, encouraging the samples $\{f_\theta(z)\}_{z \sim q_\phi(z|y)}$ to remain consistent with data manifold.

In the following result, we identify a concrete benefit of jointly learning $f_\theta$ and $q_\phi$ to target the true posterior $p(x|y)$, under a simple Gaussian setting. While this does not claim that the distribution of samples $\{f_\theta(z)\}_{z \sim q_\phi(z|y)}$ matches $p(x|y)$ exactly, it shows that joint training can at least match the posterior mean for every observation $y$. This sharply contrasts with separately training $f_\theta$ and $q_\phi$, which leads to bias almost surely, even at the level of the posterior mean.

**Proposition 3.1.** *Assume that $p(z) = \mathcal{N}(z|0, I)$, $p(x) = \mathcal{N}(x|m, C)$ for some $m \in \mathbb{R}^d$ and $C \in \mathbb{R}^{d \times d}$ symmetric positive definite, $f_\theta(z) = K_\theta z + b_\theta$ and $q_\phi(z|y) = \mathcal{N}(z|\mu_\phi(y), \mathtt{diag}(\sigma_\phi^2(y)))$. Then, for any linear observation $y = Ax + \varepsilon$, we have that*

1. ***Separate Training:*** *Training $f_\theta$ first to match $p(x)$ and then training $q_\phi$ via loss (12) with $\theta$ fixed almost surely fails to match the posterior mean, i.e., $\mathbb{E}_{z \sim q_\phi(z|y)}[f_\theta(z)] \neq \mathbb{E}_{p(x|y)}[x]$.*

2. ***Joint Training:*** *Joint optimization of $f_\theta$ and $q_\phi$ via loss (12) recovers the true posterior mean $\mathbb{E}_{p(x|y)}[x]$ exactly via the procedure $\mathbb{E}_{z \sim q_\phi(z|y)}[f_\theta(z)]$.*

*Proof.* See Proposition A.13 in Appendix A.2. $\square$

Next, we relate the new term $\mathcal{L}_{\mathrm{data}}(\theta, \phi)$ in (12) to the *mean flow loss* (Geng et al., 2025), which imposes structural constraints on the flow map.

**Connection to mean flows.** We briefly recall the mean flow objective from (Geng et al., 2025). Denoting

$$\mathcal{E}_\theta(x, z, r, t) := (t - r) \left[ u_\theta(\psi_t(x, z), r, t) - \dot{\psi}_t(x, z) \right],$$
$$\text{where} \quad \psi_t(x, z) := (1 - t)x + tz, 0 \leq r \leq t \leq 1, \quad (16)$$

is the linear interpolant between data $x$ and noise $z$, the mean flow loss is given by

$$\mathbb{E}_{x,z,r,t} \left[ \|\partial_t \mathcal{E}_\theta(x, z, r, t)\|^2 \right] \approx \mathcal{L}_{\mathrm{MF}}(\theta) \quad (17)$$
$$:= \mathbb{E}_{x,z,r,t} \left[ \|u_\theta(\psi_t(x, z), r, t) - \mathtt{stopgrad}(u_{\mathrm{tgt}})\|^2 \right],$$

where $u_{\mathrm{tgt}} := \dot{\psi}_t(x, z) - (t - r)\frac{d}{dt}u_\theta(\psi_t(x, z), r, t)$ is the effective regression target. Below, we establish a direct link between this objective and the term $\mathcal{L}_{\mathrm{data}}(\theta, \phi)$ in (12).

**Proposition 3.2.** *Let the noise-to-data map $f_\theta$ be defined by $f_\theta(z) := z - u_\theta(z, 0, 1)$. Then we have*

$$\|x - f_\theta(z)\|^2 \leq \int_0^1 \|\partial_t \mathcal{E}_\theta(x, z, 0, t)\|^2 dt. \quad (18)$$

*Proof.* See Appendix A.3. $\square$

This result shows that the mean flow loss in the anchored case $r = 0$ and $t \sim U([0, 1])$ acts as an upper bound proxy to the reconstruction error $\|x - f_\theta(z)\|^2$ in (28). This specialized setting targets direct one-step transport to $r = 0$. Motivated by this connection, we opt to use the general mean flow loss (17), which distributes learning over $(r, t)$ to additionally learn intermediate flow maps $f_\theta(x_t, r, t)$. While this does not ensure optimality for the one-step transport $x = f_\theta(z, 0, 1)$, in practice, it yields strong empirical performance and furthermore provides functionality for multi-step sampling (Section 3.3). Summarizing, we propose to train $(\theta, \phi)$ using the following objective:

$$\mathcal{L}_{\theta,\phi} := \frac{1}{2\tau^2}\mathcal{L}_{\mathrm{MF}}(\theta; \phi) + \frac{1}{2\sigma^2}\mathcal{L}_{\mathrm{obs}}(\theta, \phi) + \mathcal{L}_{\mathrm{KL}}(\phi), \quad (19)$$

---

**Algorithm 1** Multi-Step Conditional Sampling with VFM

1: **Input:** Observation $y$, inverse problem class $c$, time partition $1 = t_0 > \cdots > t_K = 0$, adapter mean and standard deviation $\mu_\phi, \sigma_\phi$, mean flow model $u_\theta$
2: $\epsilon \sim \mathcal{N}(0, I)$
3: $z \leftarrow \mu_\phi(y, c) + \sigma_\phi(y, c) \odot \epsilon$
4: $x \leftarrow z$
5: **for** $k = 1$ to $K$ **do**
6: $\quad x \leftarrow x + (t_k - t_{k-1})u_\theta(x, t_k, t_{k-1})$
7: **end for**
8: **Output:** $x$

---

**Algorithm 2** Joint training of the adapter and flow map

1: **Input:** Inverse problem classes $\mathcal{A}_1, \ldots, \mathcal{A}_C$, observation noise standard deviation $\sigma$, data misfit tolerance $\tau$, conditional noise proportion $\alpha$, learning rates $\eta_1, \eta_2$, EMA rate $\mu$, adaptive loss constants $\gamma, p$
2: $\theta^- \leftarrow \texttt{stopgrad}(\theta)$
3: **repeat**
4: $\quad$ Sample $c \sim p(c)$, $x \sim p(x)$
5: $\quad$ Sample forward operator $A_c^\omega \in \mathcal{A}_c$
6: $\quad y \leftarrow A_c^\omega x + \varepsilon, \quad \varepsilon \sim \mathcal{N}(0, \sigma^2 I)$
7: $\quad z \leftarrow \mu_\phi(y, c) + \sigma_\phi(y, c) \odot \epsilon, \quad \epsilon \sim \mathcal{N}(0, I)$
8: $\quad \mathcal{L}_{\text{obs}}(\phi) \leftarrow \|y - A_c^\omega(f_{\theta^-}(z, 0, 1))\|^2$
9: $\quad \mathcal{L}_{\text{KL}}(\phi) \leftarrow \text{KL}\Big(\mathcal{N}(\mu_\phi(y, c), \sigma_\phi^2(y, c)) \,\|\, \mathcal{N}(0, I)\Big)$
10: $\quad$ Sample $w \sim U([0, 1])$ and $(r, t) \sim p(r, t)$
11: $\quad$ **if** $w > \alpha$ **then**
12: $\quad\quad z \sim \mathcal{N}(0, I)$
13: $\quad$ **end if**
14: $\quad \mathcal{L}_{\text{MF}}(\theta; \phi) \leftarrow \text{MeanFlowLoss}(x, z, r, t)$
15: $\quad \mathcal{L}(\theta, \phi) \leftarrow \frac{1}{2\tau^2}\mathcal{L}_{\text{MF}}(\theta; \phi) + \frac{1}{2\sigma^2}\mathcal{L}_{\text{obs}}(\theta) + \mathcal{L}_{\text{KL}}(\phi)$
16: $\quad \mathcal{L}(\theta, \phi) \leftarrow \mathcal{L}(\theta, \phi)/\texttt{stopgrad}(\|\mathcal{L}(\theta, \phi) + \gamma\|^p)$
17: $\quad \theta \leftarrow \theta - \eta_1 \nabla_\theta \mathcal{L}(\theta, \phi)$
18: $\quad \phi \leftarrow \phi - \eta_2 \nabla_\phi \mathcal{L}(\theta, \phi)$
19: $\quad \theta^- \leftarrow \texttt{stopgrad}(\mu\theta^- + (1 - \mu)\theta)$
20: **until** convergence

---

where the mean flow term is evaluated using $(x, z)$-pairs sampled from the joint distribution $\pi_\phi(x, z) := \int q_\phi(z|y)p(y|x)p(x)dy$, in accordance with (28). This dependence induces an implicit coupling between $\theta$ and $\phi$. To promote stable optimization, we further limit the interaction to this term by replacing $\theta$ in the observation loss $\mathcal{L}_{\text{obs}}$ with its exponential moving average (EMA), yielding $\mathcal{L}_{\text{obs}}(\theta^-, \phi)$, where $\theta^-$ denotes the EMA of $\theta$.

*Remark* 3.3. Our framework can also be related to consistency model training by Proposition 6.1 in (Silvestri et al., 2025). In this case, the mean flow loss in (19) is replaced by an appropriate consistency loss.

### 3.2. Amortizing Over Multiple Inverse Problems

In many applications, one is interested not in a single inverse problem defined by a fixed forward operator $A$, but rather a *family of inverse problems*. To accommodate this setting, we extend our framework by amortizing inference over multiple forward operators $A_1, \ldots, A_C$. This allows for a single model to handle multiple tasks, such as denoising, inpainting, and deblurring.

To achieve this, we consider a class-conditional noise adapter $q_\phi(z|y, c) = \mathcal{N}(z|\mu_\phi(y, c), \texttt{diag}(\sigma_\phi^2(y, c)))$, where $c \in \{1, \ldots, C\}$ is a categorical variable indicating which forward operator $A_c$ was used to generate the observation $y$. Conditioning the adapter on $c$ enables the model to adapt its posterior approximation to the specific structure of each inverse problem. We may further extend this by amortizing over *inverse problem classes*, where $c$ now defines a collection of inverse problems $\mathcal{A}_c = \{A_c^\omega\}_{\omega \in \Omega}$. For example, these can define a family of random masks or a distribution of blurring kernels.

### 3.3. Single and Multi-Step Conditional Sampling

Given a trained noise adapter $q_\phi(z|y)$ and flow map $f_\theta(z)$, samples from the data-space posterior $p(x|y)$ can be approximately generated by first sampling $z \sim q_\phi(z|y)$ and then mapping $x = f_\theta(z)$. The validity of this procedure is justified by the following result.

**Proposition 3.4.** *Let the joint distribuion of $(x, y, z)$ be given by $p(x, y, z) = p_\theta(x, y|z)p(z)$, for $p_\theta(x, y|z)$ in (11). Then, for any fixed observation $y$, the data-space posterior $p(x|y)$ converges weakly to the pushforward of the noise-space posterior $p(z|y)$ under the map $f_\theta$, as $\tau \to 0$.*

*Proof.* See Appendix A.4. $\square$

The proposition states that in the limiting case $\tau \to 0$, sampling from $p(x|y)$ is equivalent in distribution to first sampling $z \sim p(z|y)$ (approximated by $q_\phi(z|y)$) and then applying $x = f_\theta(z)$. While sound in theory, we find that when $\tau \ll \sigma$, joint optimization of $(\theta, \phi)$ becomes difficult. This is likely due to the RHS distribution in (10) concentrating sharply around the submanifold $\{(x, y, z) : x = f_\theta(z)\}$, making it nearly impossible to match using the LHS representation of (10), which remains a full distribution over $(x, y, z)$. In practice, we find that using $\tau$ larger than $\sigma$ yields stable optimization and the best empirical results.

Sample quality can also be improved by considering *multi-step sampling* instead of single-step sampling, as described in Algorithm 1. Empirically, high-quality samples can be obtained with only a small number of steps $K$, substantially fewer than the number of integration steps required for solving a full generative ODE or SDE.

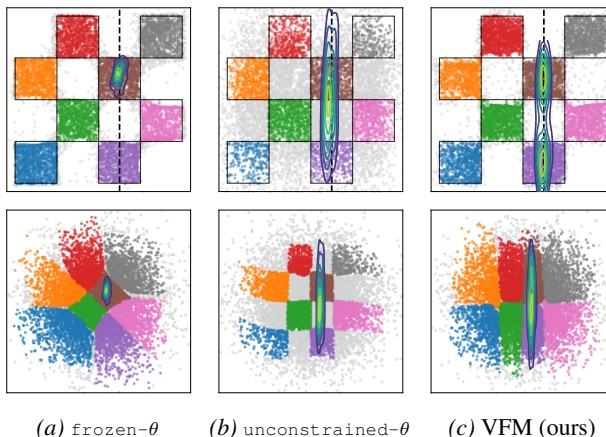

*(a)* `frozen-`$\theta$     *(b)* `unconstrained-`$\theta$     *(c)* VFM (ours)

*Figure 2.* Prior 2D samples and posterior densities in data space (top row) and noise space (bottom row). We observe the $x$-component (black dashed lines) with $\sigma = 0.1$. The unconditional samples are color-coded by checkerboard cell; light grey for off-manifold samples. VFM successfully captures the bimodal nature of the posterior, while the baselines struggle to do so.

### 3.4. Other Training Considerations

**Mixing in the unconditional loss:** We observe that training solely using the objective (19) can degrade the quality of unconditional samples $x = f_\theta(z)$, with $z \sim \mathcal{N}(0, I)$. This behaviour arises because latent samples drawn from $q_\phi(z|y)$ retain structural details of $y$, and therefore are not fully representative of pure noise drawn from $\mathcal{N}(0, I)$. Thus, during training, the mean flow loss is never evaluated on pure noise, imparining the model to generate unconditional samples. To address this, we modify the computation of the mean flow loss $\mathcal{L}_{\mathrm{MF}}(\theta; \phi)$, by sampling $(x, z) \sim \pi_\phi(x, z)$ with probability $\alpha$ and with remaining probablility $1 - \alpha$, we sample $z \sim \mathcal{N}(0, I)$ independently of $x$.

**Adaptive loss:** Similar to the mean flow training procedure of (Geng et al., 2025), we consider an adaptive loss scaling to stabilize optimization. Specifically, we use the rescaled loss $w \cdot \mathcal{L}_{\theta,\phi}$, where the weight $w$ is given by $w = 1/\mathtt{stopgrad}(\|\mathcal{L}_{\theta,\phi} + \gamma\|^p)$ for constants $\gamma, p > 0$.

We summarize the full training procedure in Algorithm 2.

## 4. Experiments
### 4.1. Illustration on a 2D Example

In this experiment, we illustrate the effects of jointly training $(\theta, \phi)$ on a toy 2D example, and perform ablations on key design choices in VFM. Specifically, we take $p(x)$ to be a $4 \times 4$ checkerboard distribution supported on $[-2, 2] \times [-2, 2]$. For the forward problem, we observe only the first coordinate, i.e. $y = Ax + \varepsilon$ with $A = \begin{pmatrix} 1 & 0 \end{pmatrix}$ and $\varepsilon \sim \mathcal{N}(0, \sigma^2)$ with $\sigma = 0.1$. We refer the readers to Appendix

**Baselines and evaluation metrics.** We consider two baselines: the first, `frozen-`$\theta$ trains only the noise adapter $q_\phi(z|y)$ via loss (9) (amortized over $y$), while keeping $\theta$ fixed to a pretrained flow map. The second, `unconstrained-`$\theta$, optimizes the same objective but learns $\theta$ jointly with $\phi$. These baselines are chosen to illustrate (i) the effect of joint optimization of $\theta$ and $\phi$, and (ii) the failure mode that can occur when $\theta$ is trained without the structural constraints imposed by the mean flow loss.

For model evaluation, we use the following metrics: (1) The negative log predictive density (NLPD), evaluates how well generated samples are consistent with observations $y$; (2) the continuous ranked probability score (CRPS) measures uncertainty calibration around the ground truth $x$ that generated $y$; (3) the maximum mean discrepancy (MMD) provides a sample-based distance between the true and approximate posteriors (Gretton et al., 2012); (4) the support accuracy (SACC) measures the proportion of samples $x = f_\theta(z)$ that lie on the checkerboard support. We compare MMD and SACC on both unconditional samples $\{f_\theta(z)\}_{z \sim \mathcal{N}(0, I)}$ and conditional samples $\{f_\theta(z)\}_{z \sim q_\phi(z|y)}$ to evaluate the quality of both prior and posterior approximations, respectively. For details, see Appendix B.1.3.

**Ablation on the loss components.** We compare VFM against `frozen-`$\theta$ and `unconstrained-`$\theta$ to isolate the effect of the mean flow term $\mathcal{L}_{\mathrm{MF}}(\theta; \phi)$ in (19); results displayed in Figure 2. The `frozen-`$\theta$ baseline (Figure 2a) fails to capture the bimodality of the true posterior (support in the brown and purple cells), due to the limited flexibility of $q_\phi$. On the other hand, `unconstrained-`$\theta$ (Figure 2b) is able to sample from both brown and purple cells, however, also produces many off-manifold samples. VFM (Figure 2c, $\tau = 100$, $\alpha = 1$) successfully captures both modes while preserving the checkerboard pattern; joint training improves the noise-to-data coupling, while $\mathcal{L}_{\mathrm{MF}}$ pull samples towards the structured data manifold. This observation is supported by the improvements in CRPS and posterior MMD (see Figures 6 & 7, Appendix), and high support accuracy comparable to the pretrained flow map used in `frozen-`$\theta$. Finally, removing $\mathcal{L}_{\mathrm{KL}}(\phi)$ from (19) makes training unstable (Figure 8d, Appendix), owing to the ill-posedness of the inverse problem without prior regularization.

**Ablation on $\tau$ and $\alpha$.** We sweep $\tau \in [10^{-2}, 10^2]$, and report metrics using a single-step and 4-step sampler (See Figures 6 & 7, Appendix). When $\tau \lesssim \sigma$, performance across metrics is generally worse than `frozen-`$\theta$ (Figures 9a & 9b, Appendix). For $\tau \geq 1$, results improve substantially, especially CRPS and posterior MMD, while SACC and prior MMD approach the strong values already achieved by the pretrained flow used in `frozen-`$\theta$. We also ablate on $\alpha$, fixing $\tau = 100$ (Figure 10, Appendix). Setting $\alpha = 0$

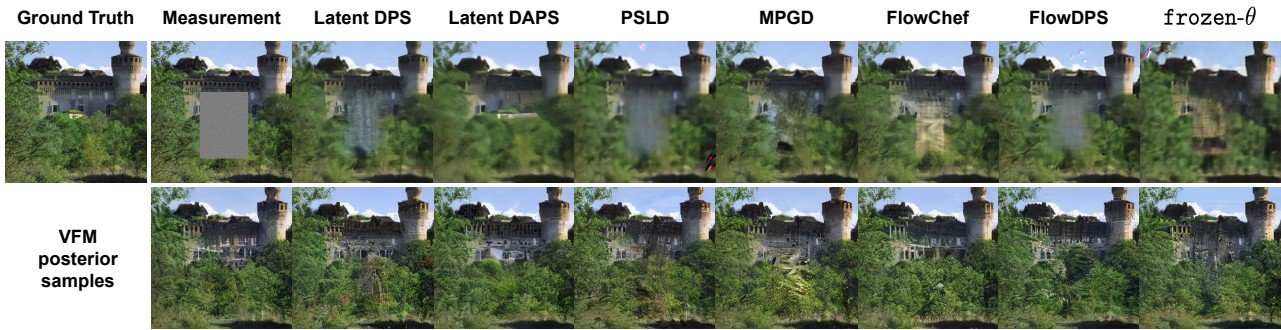

*Figure 3.* Qualitative comparison on ImageNet 256×256 box inpainting. Top row: ground truth, measurement, and reconstructions from guidance-based baselines. Bottom row: conditional samples produced by VFM, showing diversity in the inpainted region.

decouples the training of mean flow and the adapter, yielding behaviour close to frozen-$\theta$. Increasing $\alpha$ strengthens the coupling, inducing a more pronounced warping of the latent space. In practice, $\alpha < 1$ is more stable and yields better prior fit (lower prior MMD, compare Figures 6d and 7d), whereas $\alpha = 1$ gives the best posterior fit (lower posterior MMD, see Figures 6c vs 7c, Appendix).

**To EMA or not to EMA.** Finally, we examine the role of using an EMA of $\theta$ in the observation loss $\mathcal{L}_{\text{obs}}(\theta, \phi)$. Without EMA, i.e., allowing $\theta$-gradients to propagate through $\mathcal{L}_{\text{obs}}$, both prior and posterior support accuracy deteriorate as $\tau$ increases (orange curves in Figures 6 and 7). This can be explained by the fact that in the limit $\tau \to \infty$, this pushes training toward the unconstrained-$\theta$ failure mode, leading to unstructured sample generation. This can be seen in Figure 8c, where the no-EMA variant when $\tau = 100$ yields results similar to unconstrained-$\theta$.

### 4.2. Image Inverse Problems

We evaluate VFM on standard image inverse problems using ImageNet 256×256, comparing against established guidance-based solvers, as well as the frozen-$\theta$ baseline considered in our earlier 2D experiment. For VFM, we amortize over the problems, as described in Section 3.2. All methods operate in the latent space of SD-VAE (Rombach et al., 2022). We provide further details of our experimental settings in Appendix B.2.

**Comparison with guidance-based methods.** Table 1 reports quantitative results on box inpainting and Gaussian deblurring tasks (additional tasks are in Table 2, Appendix). For VFM, we report results for both single posterior samples and averaged estimates over 10 posterior samples, shown as {sample}/{average}. For all guidance-based baselines, we use the same flow-matching backbone (SiT-B/2) used to initialize our mean-flow model.

Across both tasks, we observe that VFM is consistently better than the baselines on distributional metrics (FID, MMD

& CRPS), e.g., on box inpainting, the FIDs on the baselines range between 63–83, while we achieve an FID of 41.76. These improvements align with the qualitative results in Figure 3, where we observe that VFM exhibits notable diversity in the inpainted region, while maintaining visual sharpness. Guidance-based methods generally struggle with box-inpainting, especially when operating in latent space.

On pixel-space fidelity metrics (PSNR, SSIM), guidance methods consistently scores higher than a single VFM draw. However, both PSNR and SSIM typically reward mean behavior and thus prefer smoother results (Zhang et al., 2018). To confirm this, we observe that averaging multiple VFM samples narrows this gap and even exceeds the baselines in some instances, e.g., on Gaussian deblurring. On LPIPS, which is a feature-space perceptual similarity metric, we find that VFM is competitive even without averaging; this is consistent with LPIPS being more aligned with the perceptual quality than PSNR or SSIM (Zhang et al., 2018).

We also note the significant speed advantage of VFM at inference time: we used 250 sampling steps for the guidance methods with an additional ×2 cost for classifier-free guidance (Ho & Salimans, 2022), while VFM requires only one step to achieve competitive results, as displayed. This results in around two orders of magnitude lower wall-clock time, e.g., DAPS (Zhang et al., 2025) has an inference cost close to a minute; in comparison, the $\sim 0.03$s cost of VFM is instantaneous.

**Benefits of joint training.** The frozen-$\theta$ baseline, while fastest at inference time, performs poorly across all metrics, exhibiting visible artifacts and blurriness. This highlights the importance of jointly training the flow map $f_\theta$ and the adapter $q_\phi$, consistent with our observations from the 2D experiment that the flow map itself needs to adjust for the adapter to approximate the conditional distributions well. By training jointly, we observe surprisingly strong perceptual quality, despite the simple Gaussian structural assumption used in the variational posterior.

| Task | Method | NFE | PSNR (↑) | SSIM (↑) | LPIPS (↓) | FID (↓) | MMD (↓) | CRPS$_{\text{DINO}}$ (↓) | CRPS$_{\text{Inc}}$ (↓) | Time (s) (↓) |
|---|---|---|---|---|---|---|---|---|---|---|
| | Latent DPS | 250×2 | 22.80 | 0.704 | 0.349 | 62.89 | 0.132 | 0.511 | 0.389 | 7.223 |
| | Latent DAPS | 250×2 | **23.98** | **0.707** | 0.348 | – | – | 0.468 | 0.365 | 43.93 |
| | PSLD | 250×2 | 22.61 | 0.699 | 0.346 | 67.22 | 0.153 | 0.536 | 0.435 | 10.07 |
| Inpaint | MPGD | 250×2 | 22.76 | 0.705 | 0.350 | 62.35 | 0.132 | 0.510 | 0.388 | 7.487 |
| (box) | FlowChef | 250×2 | 22.80 | 0.704 | 0.349 | 63.20 | 0.133 | 0.512 | 0.389 | 7.612 |
| | FlowDPS | 250×2 | 23.21 | 0.706 | 0.364 | 75.62 | 0.166 | 0.606 | 0.482 | 14.47 |
| | LFlow | 250×2 | 22.47 | 0.680 | 0.395 | 82.95 | 0.184 | 0.616 | 0.491 | 6.854 |
| | `frozen-`$\theta$ | 1 | 19.41 | 0.531 | 0.528 | 136.12 | 0.255 | 0.814 | 0.601 | **0.015** |
| | VFM (ours) | 1 / 10 | 20.79 / 22.92 | 0.570 / 0.673 | 0.354 / **0.331** | **41.76** | **0.051** | **0.455** | **0.329** | 0.025 / 0.252 |
| | Latent DPS | 250×2 | 23.21 | 0.592 | 0.434 | 83.11 | 0.180 | 0.613 | 0.498 | 7.724 |
| | Latent DAPS | 250×2 | 21.46 | 0.500 | 0.432 | – | – | 0.529 | 0.422 | 46.86 |
| | PSLD | 250×2 | 23.01 | 0.591 | 0.459 | 101.23 | 0.223 | 0.675 | 0.559 | 10.28 |
| Gaussian | MPGD | 250×2 | 23.22 | 0.593 | 0.435 | 83.86 | 0.183 | 0.612 | 0.498 | 7.695 |
| deblur | FlowChef | 250×2 | 23.21 | 0.592 | 0.434 | 83.19 | **0.180** | 0.613 | 0.499 | 7.525 |
| | FlowDPS | 250×2 | 23.41 | 0.615 | 0.449 | 92.13 | 0.209 | 0.699 | 0.569 | 14.91 |
| | LFlow | 250×2 | 22.40 | 0.580 | 0.480 | 113.52 | 0.242 | 0.715 | 0.552 | 7.179 |
| | `frozen-`$\theta$ | 1 | 20.02 | 0.419 | 0.597 | 172.74 | 0.306 | 0.927 | 0.657 | **0.015** |
| | VFM (ours) | 1 / 10 | 20.75 / **23.92** | 0.503 / **0.632** | 0.392 / **0.369** | **44.12** | **0.044** | **0.496** | **0.328** | 0.027 / 0.268 |

*Table 1.* Quantitative comparison on ImageNet for box inpainting and Gaussian deblurring. Best results are in **bold**, second best are underlined. ↑: higher is better, ↓: lower is better. For VFM, we display results for single samples and average over 10 samples, displayed as {sample} / {average}. VFM achieves the best results on LPIPS, FID, MMD, CRPS, with a significantly reduced wall-clock time.

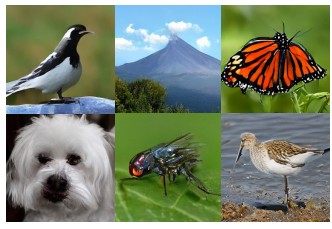

| | NFE | FID (↓) |
|---|---|---|
| iCT | 1 | 34.24 |
| Shortcut-B/2 | 1 | 40.30 |
| IMM-B/2 | 1×2 | 9.60 |
| MF-B/2 | 1 | 6.17 |
| DMF-B/2 | 1 | 5.63 |
| **VFM-B/2** | 1 | 10.77 |
| | 2 | 9.22 |

*Figure 4.* Unconditional generation on ImageNet $256 \times 256$. **Left:** unconditional samples from VFM-B/2. **Right:** unconditional FID comparison versus mean-flow baselines. VFM retains competitive performance despite it being trained for posterior sampling.

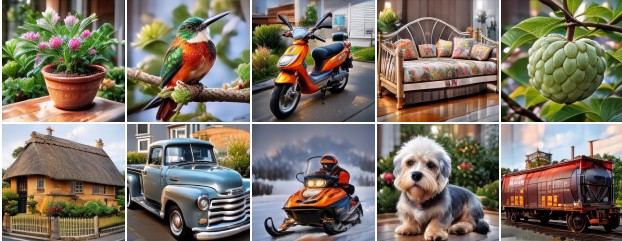

*Figure 5.* One-step reward-aligned generation using VFM fine-tuning. Starting from a pre-trained ImageNet flow map, VFM efficiently adapts the latent noise space and flow trajectories to sample from a reward-tilted distribution, achieving strong visual alignment with a target reward $R(x, c)$ in a single forward pass while preserving image quality.

**Unconditional generation.** To assess the robustness of VFM, we also evaluate unconditional generation from the trained flow map. In Figure 4, we compare the FID on 50,000 unconditional samples generated from the flow map in VFM, against various baselines with similar architecture sizes (Song & Dhariwal, 2023; Frans et al., 2025; Lee et al., 2025; Zhou et al., 2025a). We fine-tune the SiT-B/2 model (trained for 80 epochs) for an additional 50 epochs (30 epochs using only the mean flow loss + 20 epochs with the full VFM loss). We note, however, that the baselines are trained for longer ($\sim$240 epochs).

Unconditional generation of VFM remains competitive, with 2-step sampling results achieving FID below 10 (see Figure 4 for visual results). To achieve this result, we emphasize the important role of the $\alpha$ parameter; we observe that the adapter's noise outputs retain some structure from the observations (see Figure 17, Appendix) and are therefore not representative of pure standard Gaussian noise. Thus, using $\alpha < 1$ is necessary to achieve good unconditional performance. In our experiments, we used $\alpha = 0.8$.

### 4.3. General Reward Alignment via VFM Fine-Tuning

Beyond solving standard inverse problems, the Variational Flow Map presents a highly efficient framework for general reward alignment. The goal is to fine-tune a pre-trained model such that its generated samples maximize a differentiable reward function $R(x, c)$ conditioned on a context $c$, while staying close to the original data distribution. This objective effectively corresponds to sampling from a reward-tilted distribution $p_{\text{reward}}(x|c) \propto p_{\text{data}}(x) \exp(\beta R(x, c))$.

Traditional flow and diffusion reward fine-tuning methods require expensive backpropagation through iterative sampling trajectories (Denker et al., 2025; Domingo-Enrich et al., 2025; Venkatraman et al., 2025b) or rely on approximations (Clark et al., 2024; Choi et al., 2026). In contrast, VFM achieves this by learning an amortized noise adapter $q_\phi(z|c)$ that directly maps the condition $c$ to a high-reward region of the latent space, while simultaneously fine-tuning

the flow map $f_\theta$ to decode this noise into high-quality data. We formulate this by replacing the standard observation loss with a reward maximization objective:

$$\mathcal{L}_{\text{reward}}(\theta, \phi) = -\lambda \, \mathbb{E}_{c \sim p(c), z \sim q_\phi(z|c)} \left[ R(f_\theta(z), c) \right] \quad (20)$$

where $\lambda$ controls the reward strength. In this context, the reward $R(x, c)$ can be viewed as the (unnormalized) log-likelihood of the context $c$ (e.g., a text prompt) given the generated sample.

To the best of our knowledge, this is the first rigorous, scalable framework for fine-tuning flow maps to arbitrary differentiable rewards. In particular, the fine-tuning process is very fast and stable. Starting from a pre-trained flow map, VFM achieves strong reward alignment in under $0.5$ epochs. The resulting model enables sampling from the reward-tilted distribution in a single neural function evaluation (1 NFE). We provide qualitative results in Figure 5 and present further training and generation details in Appendix B.3.

## 5. Related Works

Variational/amortized inference with diffusion-based priors has been explored in previous works: (Feng et al., 2023) explores usage of score-based prior in variational inference to approximate posteriors $p(x|y)$ in data space and (Mammadov et al., 2024) extends this to the amortized inference setting. However, these approaches rely on normalizing flows to ensure sufficient flexibility for the variational posterior, making scaling to high-resolution settings difficult. The work (Mardani et al., 2023) on the other hand, uses a Gaussian variational posterior similar to ours, but still performs variational inference in data space.

Noise space posterior inference for arbitrary generative models has been considered in (Venkatraman et al., 2025a). However, their method considers a frozen generator and compensates with a more flexible noise adapter based on neural SDEs, making training significantly more complex. In comparison, VFM uses a simpler adapter and instead unfreeze the generative flow map, so the model itself can adapt to the conditional task while keeping the objective simple. The works (Zhou et al., 2025b; Eyring et al., 2026) on the other hand learn perturbations to noise that nudge generated samples towards regions of higher reward. However, these approaches target image quality refinement and are not suited for tackling inverse problems.

We also note the work (Silvestri et al., 2025), which introduces Variational Consistency Training (VCT) to address instability issues in consistency model training by learning data-dependent noise couplings through a variational encoder that maps data into a better-behaved latent representation. While conceptually related to our work, the goal is different: VCT is aimed at improving stability of uncon-

ditional consistency training, whereas VFM is designed to amortize posterior sampling for conditional generation.

Finally, Noise Consistency Training (NCT) (Luo et al., 2025) also targets one-step conditional sampling, but via a different construction: they consider a diffusion process in $(z, y)$-space and learns a consistency map from intermediate states to $(x, y)$. This is strongly tied to consistency models and therefore do not generalize naturally to flow maps, considered state-of-the-art in one-step generative modeling.

## 6. Limitations & Future Work

A limitation of VFM is that it is fundamentally a training-based method for solving inverse problems, unlike purely inference-time methods such as DPS. However, we view this one-off training in VFM as a favorable trade-off for many practical scenarios where one requires fast posterior draws for a fixed class of problems (e.g. uncertainty quantification, real-time restoration). Adapting to unseen inverse problems is also challenging under our current framework, requiring re-training without catastrophic forgetting. In addition, the joint training in VFM requires modifying the backbone flow map, which can lead to slight degradations in the unconditional performance. To address these issues, we believe that further improving the modularity of the approach is an important direction for future work.

## 7. Conclusion

We proposed Variational Flow Maps (VFMs) to enable highly efficient posterior sampling and reward fine-tuning with just a single (or few) sampling steps. VFM leverages a principled variational objective to jointly train a flow map alongside an amortized noise adapter, which infers optimal initial noise from noisy observations, class labels, or text prompts. A natural next step is to relax our current Gaussian adapter assumption by using more expressive noise models, such as normalizing flows or energy-transformers (Hoover et al., 2023), which can capture richer, non-Gaussian conditional structures. Another exciting direction for future research is to extend the VFM framework to other distillation methods and modalities; for instance, one could tackle video inverse problems, where latent noise evolution could be leveraged to promote temporal coherence among frames.

## Impact Statement

The overarching goal of reducing the computational cost for conditional generation and posterior sampling has the potential not only to drive practical applications in scientific and engineering workflows that rely on fast generation of posterior samples, but also to help reduce the high energy cost for inference. This is especially valuable as generative

models see widespread use in today's society; thus, the problem of lowering inference costs is an increasingly important challenge for machine learning. Variational flow maps take a step in this direction by enabling low-cost conditional sampling without sacrificing performance.

## Acknowledgments

AM is supported by the Clarendon Fund Scholarship, University of Oxford. ST is supported by a Department of Defense Vannevar Bush Faculty Fellowship held by Prof. Andrew Stuart, and by the SciAI Center, funded by the Office of Naval Research (ONR), under Grant Number N00014-23-1-2729. The authors acknowledge the use of resources provided by the Isambard-AI National AI Research Resource (AIRR) (McIntosh-Smith et al., 2024). Isambard-AI is operated by the University of Bristol and is funded by the UK Government's Department for Science, Innovation and Technology (DSIT) via UK Research and Innovation; and the Science and Technology Facilities Council [ST/AIRR/I-A-I/1023].

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

## A. Theory

### A.1. Derivation of the loss

We recall that the VFM objective is obtained by matching the following two representations of $p(x, y, z)$ using the KL divergence:

$$q_\phi(z|y)p(y|x)p(x) \approx p_\theta(x, y|z)p(z), \tag{21}$$

where we assumed that

$$p_\theta(x, y|z) = \mathcal{N}(x|f_\theta(z), \sigma^2 I)\, \mathcal{N}(y|Af_\theta(z), \tau^2 I). \tag{22}$$

By direct computation, this yields

$$\text{KL}(q_\phi(z|y)p(y|x)p(x) \,\|\, p_\theta(x, y|z)p(z)) \tag{23}$$

$$= -\int \log \frac{p_\theta(x, y|z)p(z)}{q_\phi(z|y)p(y|x)p(x)} q_\phi(z|y)p(y|x)p(x)\, dx\, dy\, dz \tag{24}$$

$$= -\mathbb{E}_{q_\phi(z|y)p(y|x)p(x)}\left[\log p_\theta(x, y|z)\right] + \mathbb{E}_{p(y|x)p(x)}\left[\text{KL}\left(q_\phi(z|y)\,\|\,p(z)\right)\right] + \underbrace{\mathbb{E}_{p(y|x)p(x)}[\log(p(y|x)p(x))]}_{\leq 0} \tag{25}$$

$$\leq -\mathbb{E}_{q_\phi(z|y)p(y|x)p(x)}\left[\log p_\theta(x, y|z)\right] + \mathbb{E}_{p(y)}\left[\text{KL}\left(q_\phi(z|y)\,\|\,p(z)\right)\right] \tag{26}$$

$$\overset{(22)}{=} -\mathbb{E}_{q_\phi(z|y)p(y)}\left[\log \mathcal{N}(x|f_\theta(z), \sigma^2 I) + \log \mathcal{N}(y|Af_\theta(z), \tau^2 I)\right] + \mathbb{E}_{p(y|x)p(x)}\left[\mathcal{KL}\left(q_\phi(z|y)\,\|\,p(z)\right)\right], \tag{27}$$

where we used that $\mathbb{E}_{p(y|x)p(x)}[\log(p(y|x)p(x))] \leq 0$ since this is the negative Shannon entropy of the joint distribution $H(p(x, y)) := -\mathbb{E}_{p(x,y)}[\log(p(x, y))] \geq 0$. This yields

$$\text{KL}(q_\phi(z|y)p(y|x)p(x) \,\|\, p_\theta(x, y|z)p(z)) \leq \frac{1}{2\tau^2}\mathcal{L}_{\text{data}}(\theta, \phi) + \frac{1}{2\sigma^2}\mathcal{L}_{\text{obs}}(\theta, \phi) + \mathcal{L}_{\text{KL}}(\phi),$$

where

$$\mathcal{L}_{\text{data}}(\theta, \phi) = \mathbb{E}_{q_\phi(z|y)p(y|x)p(x)}\left[\|x - f_\theta(z)\|^2\right], \tag{28}$$

$$\mathcal{L}_{\text{obs}}(\theta, \phi) = \mathbb{E}_{q_\phi(z|y)p(y)}\left[\|y - Af_\theta(z)\|^2\right], \tag{29}$$

$$\mathcal{L}_{\text{KL}}(\phi) = \mathbb{E}_{p(y)}\left[\text{KL}\left(q_\phi(z|y)\,\|\,p(z)\right)\right]. \tag{30}$$

### A.2. Proof of Proposition 3.1

This section provides the formal proofs for Proposition 3.1 within a Linear-Gaussian framework. We analyze the interaction between the generative map $f_\theta$ and the variational posterior $q_\phi$ to demonstrate that joint optimization is necessary for exact posterior mean recovery under diagonal constraints. The derivation proceeds from characterizing the optimal parameters to proving the almost sure failure of separate training in Proposition A.13. We conclude with Remark A.14, which discusses the extension of these results to non-linear cases through the lens of Jacobian alignment and symmetry restoration.

**Data and Observation Model.** We assume the ground truth data $x \in \mathbb{R}^d$ follows a Gaussian distribution:

$$x \sim p_{data}(x) = \mathcal{N}(x|m, C), \tag{31}$$

where $m \in \mathbb{R}^d$ is the data mean and $C \in \mathbb{R}^{d \times d}$ is the symmetric positive definite (SPD) covariance matrix. The observation $y \in \mathbb{R}^{d_y}$ is obtained via a linear operator $A \in \mathbb{R}^{d_y \times d}$ with additive Gaussian noise:

$$y = Ax + \epsilon, \quad \epsilon \sim \mathcal{N}(0, \sigma^2 I), \tag{32}$$

where $\sigma > 0$ is the noise level. Consequently, the marginal distribution of observations is given by

$$p(y) = \mathcal{N}(y|\mu_y, \Sigma_y), \quad \text{where } \mu_y = Am, \quad \Sigma_y = ACA^\top + \sigma^2 I. \tag{33}$$

**Generative Model.** We define the generative model $f_\theta : \mathbb{R}^d \to \mathbb{R}^d$ as a linear map acting on a standard Gaussian latent variable $z$:

$$z \sim p(z) = \mathcal{N}(z|0, I), \tag{34}$$

$$x = f_\theta(z) = K_\theta z + b_\theta, \tag{35}$$

where $\theta = \{K_\theta, b_\theta\}$ are the learnable parameters with $K_\theta \in \mathbb{R}^{d \times d}$ and $b_\theta \in \mathbb{R}^d$. The induced model distribution is $p_\theta(x) = \mathcal{N}(b_\theta, K_\theta K_\theta^\top)$.

**Amortized Inference (Adapter).** We parameterize the variational posterior (noise adapter) $q_\phi(z|y)$ as a multivariate Gaussian distribution:

$$q_\phi(z|y) = \mathcal{N}(\mu_\phi(y), \Sigma_\phi(y)), \tag{36}$$

where $\mu_\phi : \mathbb{R}^{d_y} \to \mathbb{R}^d$ and $\Sigma_\phi : \mathbb{R}^{d_y} \to \mathbb{R}^{d \times d}$ are generally parameterized by neural networks. While one may optionally restrict $\Sigma_\phi(y)$ to be a diagonal matrix for computational efficiency.

In the following sections, we will derive the optimal solutions for $\theta = \{K_\theta, b_\theta\}$ and $\phi$ under the separate training and joint training paradigms, respectively.

**Training Objective.** Recall that in the general framework, we minimized a joint objective consisting of a data matching term, observation matching term, and a KL divergence term:

$$\mathcal{L}(\theta, \phi) = \underbrace{\mathbb{E}_{y \sim p_{\mathrm{data}}(y)} \mathbb{E}_{z \sim q_\phi(z|y)} \left[ \frac{1}{2\sigma^2} \|y - A f_\theta(z)\|^2 \right]}_{\mathcal{L}_{\mathrm{obs}}:\text{ Observation Loss}}$$

$$+ \underbrace{\mathbb{E}_{x \sim p_{\mathrm{data}}(x)} \mathbb{E}_{y \sim p(y|x)} \mathbb{E}_{z \sim q_\phi(z|y)} \left[ \frac{1}{2\tau^2} \|x - f_\theta(z)\|^2 \right]}_{\mathcal{L}_{\mathrm{data}}:\text{ Data Fitting Loss}} \tag{37}$$

$$+ \underbrace{\mathbb{E}_{y \sim p_{\mathrm{data}}(y)} \left[ \mathrm{KL}(q_\phi(z|y) \| p(z)) \right]}_{\mathcal{L}_{\mathrm{KL}}:\text{ KL Loss}}.$$

In the linear-Gaussian theoretical analysis, the generative map $f_\theta(z) = K_\theta z + b_\theta$ is explicitly parameterized as a single-step affine transformation. Note that $\mathcal{L}_{\mathrm{data}}$ corresponds to the negative expected log-likelihood term $-\mathbb{E}\left[\log \mathcal{N}(x|f_\theta(z), \tau^2 I)\right]$.

**Definition A.1** (Matrix Sets and Measure). We denote the set of $d \times d$ orthogonal matrices as the orthogonal group $\mathbb{O}(d) := \{Q \in \mathbb{R}^{d \times d} \mid Q^\top Q = I\}$. The space $\mathbb{O}(d)$ is equipped with the unique normalized Haar measure $\nu_{\mathbb{O}(d)}$, representing the uniform distribution over the group. Furthermore, let $\mathbb{S}^d$ represent the space of $d \times d$ real symmetric matrices. The subsets of symmetric positive semi-definite (SPSD) and symmetric positive definite (SPD) matrices are denoted by $\mathbb{S}_+^d := \{M \in \mathbb{S}^d \mid x^\top M x \geq 0, \forall x \in \mathbb{R}^d\}$ and $\mathbb{S}_{++}^d := \{M \in \mathbb{S}^d \mid x^\top M x > 0, \forall x \in \mathbb{R}^d \setminus \{0\}\}$, respectively. We denote the set of $d \times d$ real diagonal matrices as $\mathbb{D}(d) := \{\mathrm{diag}(d_1, \ldots, d_d) \mid d_i \in \mathbb{R}\}$. We denote the determinant of a square matrix $M$ by $|M|$.

**Lemma A.2** (Optimal Generative Parameters via KL Minimization). *Consider the data distribution $p_{\mathrm{data}}(x) = \mathcal{N}(m, C)$ and the induced model distribution $p_\theta(x) = \mathcal{N}(b_\theta, \Sigma_\theta)$ with $\Sigma_\theta = K_\theta K_\theta^\top$. Let $C = U\Lambda^2 U^\top$ be the eigen-decomposition of the data distribution covariance, where $U \in \mathbb{O}(d)$ and $\Lambda \in \mathbb{D}(d)$ has positive entries. The set of optimal parameters $\Theta^* := \arg\min_\theta \mathrm{KL}(p_{\mathrm{data}}(x) \| p_\theta(x))$ is given by:*

$$\Theta^* = \{\{K_\theta, b_\theta\} \mid b_\theta = m, \ K_\theta = U\Lambda Q, \ \forall Q \in \mathbb{O}(d)\}. \tag{38}$$

*Proof.* The KL divergence between two multivariate Gaussians is minimized if and only if their first and second moments match, i.e., $b_\theta = m$ and $\Sigma_\theta = C$. Substituting the parameterization $\Sigma_\theta = K_\theta K_\theta^\top$ and the eigen-decomposition of $C$, the second condition becomes $K_\theta K_\theta^\top = U\Lambda^2 U^\top = (U\Lambda)(U\Lambda)^\top$. This equality holds if and only if $K_\theta = U\Lambda Q$ for some $Q \in \mathbb{R}^{d \times d}$ such that $QQ^\top = I$, which implies $Q \in \mathbb{O}(d)$. $\qquad\square$

**Definition A.3** (Optimal Loss Value). We define the optimal loss value for any $\theta \in \Theta^*$ and any $\phi$ as:

$$\mathcal{L}_{\text{opt}} = \min_{\theta \in \Theta^*, \phi} \mathcal{L}(\theta, \phi). \tag{39}$$

**Lemma A.4.** *Consider the joint training objective $\mathcal{L}(\theta, \phi)$ in the Linear-Gaussian setting. For fixed generative parameters $\theta = \{K_\theta, b_\theta\}$, the optimal variational posterior $q_{\phi^*}(z|y) = \mathcal{N}(\mu^*(y), \Sigma^*(y))$ that minimizes the loss (37) (under the constratint that $\Sigma(y) \in \mathbb{S}_{++}^d$) is given by:*

$$\mu^*(y) := K_\phi y + b_\phi, \tag{40}$$
$$\Sigma^*(y) := \Sigma_\phi, \tag{41}$$

*where*

$$\Sigma_\phi := \left( I_d + \frac{1}{\tau^2} K_\theta^\top K_\theta + \frac{1}{\sigma^2} K_\theta^\top A^\top A K_\theta \right)^{-1}, \tag{42}$$

$$K_\phi := \Sigma_\phi K_\theta^\top \left( \frac{1}{\sigma^2} A^\top + \frac{1}{\tau^2} K \right), \tag{43}$$

$$b_\phi := \Sigma_\phi K_\theta^\top \left[ -\frac{1}{\sigma^2} A^\top A b_\theta + \frac{1}{\tau^2} (I_d - KA)m - \frac{1}{\tau^2} b_\theta \right], \tag{44}$$

*and $K = CA^\top (ACA^\top + \sigma^2 I_{d_y})^{-1}$ denotes the Kalman gain matrix associated with the data distribution. In particular, this shows that the optimal covariance $\Sigma^*$ is independent of $y$, and the optimal mean $\mu^*(y)$ is an affine function of $y$*

*Proof.* The total loss is expressed as the expectation $\mathcal{L} = \mathbb{E}_{y \sim p(y)}[J(y; \mu, \Sigma)]$, where $\mu := \mu_\phi(y)$ and $\Sigma := \Sigma_\phi(y)$. The pointwise objective $J(y; \mu, \Sigma)$ is

$$\begin{aligned} J(y; \mu, \Sigma) = \; & \frac{1}{2\sigma^2} \left( \|y - Ab_\theta - AK_\theta\mu\|^2 + \text{Tr}(K_\theta^\top A^\top A K_\theta \Sigma) \right) \\ & + \frac{1}{2\tau^2} \left( \mathbb{E}_{x|y}[\|x - b_\theta - K_\theta\mu\|^2] + \text{Tr}(K_\theta^\top K_\theta \Sigma) \right) \\ & + \frac{1}{2} \left( \text{Tr}(\Sigma) + \|\mu\|^2 - \ln|\Sigma| \right). \end{aligned} \tag{45}$$

Differentiating $J$ with respect to $\Sigma$ yields

$$\frac{\partial J}{\partial \Sigma} = \frac{1}{2} \left( \frac{1}{\sigma^2} K_\theta^\top A^\top A K_\theta + \frac{1}{\tau^2} K_\theta^\top K_\theta + I_d \right) - \frac{1}{2} \Sigma^{-1}. \tag{46}$$

The stationary point of this gradient corresponds to the constant optimal covariance matrix $\Sigma_\phi$ defined in (42), naturally satisfying the SPD restriction. Similarly, the gradient with respect to the variational mean $\mu$ is given by

$$\nabla_\mu J = -\frac{1}{\sigma^2} K_\theta^\top A^\top (y - Ab_\theta - AK_\theta\mu) - \frac{1}{\tau^2} K_\theta^\top (\mathbb{E}[x|y] - b_\theta - K_\theta\mu) + \mu. \tag{47}$$

Rearranging the terms for the condition $\nabla_\mu J = 0$, it follows that

$$\left( I_d + \frac{1}{\sigma^2} K_\theta^\top A^\top A K_\theta + \frac{1}{\tau^2} K_\theta^\top K_\theta \right) \mu = \frac{1}{\sigma^2} K_\theta^\top A^\top (y - Ab_\theta) + \frac{1}{\tau^2} K_\theta^\top (\mathbb{E}[x|y] - b_\theta). \tag{48}$$

Observing that the coefficient matrix on the left-hand side is $\Sigma_\phi^{-1}$, we obtain the expression for the optimal mean

$$\mu^*(y) = \Sigma_\phi K_\theta^\top \left[ \frac{1}{\sigma^2} A^\top y - \frac{1}{\sigma^2} A^\top Ab_\theta + \frac{1}{\tau^2} \mathbb{E}[x|y] - \frac{1}{\tau^2} b_\theta \right]. \tag{49}$$

Substituting the conditional expectation of the data distribution $\mathbb{E}[x|y] = Ky + (I_d - KA)m$ into (49) results in

$$\mu^*(y) = \Sigma_\phi K_\theta^\top \left( \frac{1}{\sigma^2} A^\top + \frac{1}{\tau^2} K \right) y + \Sigma_\phi K_\theta^\top \left[ -\frac{1}{\sigma^2} A^\top Ab_\theta + \frac{1}{\tau^2} (I_d - KA)m - \frac{1}{\tau^2} b_\theta \right]. \tag{50}$$

This affine structure identifies $K_\phi$ and $b_\phi$ as defined in (43) and (44). $\qquad \square$

**Corollary A.5.** *We can optimize $\phi \in \Phi$ where*

$$\Phi := \{(K_\phi, b_\phi, \Sigma_\phi) \mid K_\phi \in \mathbb{R}^{d \times d_y}, b_\phi \in \mathbb{R}^d, \Sigma_\phi \in \mathbb{S}^d_{++}\}. \tag{51}$$

*Proof.* The functional forms derived in Proposition A.4 show that any $q_\phi$ not belonging to this parametric family is strictly sub-optimal for the joint loss $\mathcal{L}(\theta, \phi)$, thus reducing the search space to the coefficients $\{K_\phi, b_\phi, \Sigma_\phi\}$. $\square$

**Definition A.6** (Separate Training). The separate training paradigm consists of a two-stage sequential optimization. First, the generative parameters $\theta = \{K_\theta, b_\theta\}$ are obtained by minimizing the unconditional KL divergence

$$\theta^* = \operatorname*{argmin}_{\theta} \mathrm{KL}\left((f_\theta)_\sharp \mathcal{N}(0, I) \,\|\, p_{\text{data}}(x)\right), \tag{52}$$

which, in the linear-Gaussian case, implies $b_{\theta^*} = m$ and $K_{\theta^*} K_{\theta^*}^\top = C$. Subsequently, the variational parameters are determined by fixing $\theta^*$ and minimizing the joint objective

$$\phi^* = \operatorname*{argmin}_{\phi} \mathcal{L}(\theta^*, \phi). \tag{53}$$

**Definition A.7** (Joint Training). The joint training paradigm optimizes $\theta$ and $\phi$ simultaneously by minimizing the regularized objective with $\alpha > 0$,

$$\min_{\theta, \phi} \mathcal{L}(\theta, \phi) \\ \text{s.t. } (f_\theta)_\sharp \mathcal{N}(0, I) = p_{\text{data}}. \tag{54}$$

For the linear-Gaussian framework, this constraint restricts the search space of $\theta$ to the manifold

$$\Theta^* = \{\{K_\theta, b_\theta\} \mid b_\theta = m, K_\theta K_\theta^\top = C\}. \tag{55}$$

**Definition A.8** (Solution Sets). Let $\Theta^*$ be the set of optimal generative parameters from Proposition A.2, and the set $\Phi$ is defined in (51). We define the *diagonal parameter space by restricting the covariance matrix to be diagonal, yielding*

$$\Phi_{\mathbb{D}} := \{(K_\phi, b_\phi, \Sigma_\phi) \in \Phi \mid \Sigma_\phi \in \mathbb{S}^d_{++} \cap \mathbb{D}(d)\}. \tag{56}$$

The solution sets for the training paradigms are defined as

$$\mathcal{S}^{\text{sep}} := \{(\theta^*, \phi), \theta^* \in \Theta^* \mid \phi = \operatorname*{argmin}_{\phi' \in \Phi} \mathcal{L}(\theta^*, \phi')\}, \tag{57}$$

$$\mathcal{S}^{\text{sep}}_{\text{diag}} := \{(\theta^*, \phi), \theta^* \in \Theta^* \mid \phi = \operatorname*{argmin}_{\phi' \in \Phi_{\mathbb{D}}} \mathcal{L}(\theta^*, \phi')\}, \tag{58}$$

$$\mathcal{S}^{\text{joint}} := \{(\theta, \phi) \mid (\theta, \phi) = \operatorname*{argmin}_{\theta' \in \Theta^*, \phi' \in \Phi} \mathcal{L}(\theta', \phi')\}, \tag{59}$$

$$\mathcal{S}^{\text{joint}}_{\text{diag}} := \{(\theta, \phi) \mid (\theta, \phi) = \operatorname*{argmin}_{\theta' \in \Theta^*, \phi' \in \Phi_{\mathbb{D}}} \mathcal{L}(\theta', \phi')\}. \tag{60}$$

**Lemma A.9.** *For $Q \in \mathbb{O}(d)$ and $\theta(Q) := (U\Lambda Q, m) \in \Theta^*$, there exists a corresponding optimal parameter $\phi(Q) := (K_\phi(Q), b_\phi(Q), \Sigma_\phi(Q)) \in \Phi$ such that the joint loss (37) is invariant to the choice of $Q$, i.e., $\mathcal{L}(\theta(Q), \phi(Q)) = \mathcal{L}_{\text{opt}}$. In particular, we have the explicit expressions*

$$\Sigma_\phi(Q) := Q^\top \left(I_d + \frac{1}{\tau^2}\Lambda^2 + \frac{1}{\sigma^2}\Lambda U^\top A^\top A U \Lambda\right)^{-1} Q, \tag{61}$$

$$K_\phi(Q) := \Sigma_\phi(Q) Q^\top \Lambda U^\top \left(\frac{1}{\sigma^2} A^\top + \frac{1}{\tau^2} K\right), \tag{62}$$

$$b_\phi(Q) := \Sigma_\phi(Q) Q^\top \Lambda U^\top \left[-\frac{1}{\sigma^2} A^\top A m + \frac{1}{\tau^2}(I_d - KA)m - \frac{1}{\tau^2}m\right]. \tag{63}$$

*Consequently, the solution sets for separate and joint training are*

$$\mathcal{S}^{\text{sep}} = \{(\theta(Q_{\text{sep}}), \phi(Q_{\text{sep}})) \mid Q_{\text{sep}} \in \mathbb{O}(d) \text{ is fixed}\}, \tag{64}$$

$$\mathcal{S}^{\text{joint}} = \{(\theta(Q), \phi(Q)) \mid \forall Q \in \mathbb{O}(d)\}. \tag{65}$$

*Proof.* For a fixed $Q \in \mathbb{O}(d)$, let $K_\theta = U\Lambda Q$ and $b_\theta = m$. Substituting these into the optimality conditions (42)–(44) yields the parameterized forms of $\Sigma_\phi(Q)$, $K_\phi(Q)$, and $b_\phi(Q)$. The optimal precision matrix $P(Q) := (\Sigma_\phi(Q))^{-1}$ satisfies

$$P(Q) = I_d + \frac{1}{\tau^2}Q^\top \Lambda U^\top U\Lambda Q + \frac{1}{\sigma^2}Q^\top \Lambda U^\top A^\top AU\Lambda Q$$
$$= Q^\top \left(I_d + \frac{1}{\tau^2}\Lambda^2 + \frac{1}{\sigma^2}\Lambda U^\top A^\top AU\Lambda\right)Q := Q^\top HQ, \tag{66}$$

where $H$ is a SPD matrix independent of $Q$ defined by

$$H := I_d + \frac{1}{\tau^2}\Lambda^2 + \frac{1}{\sigma^2}\Lambda U^\top A^\top AU\Lambda. \tag{67}$$

According to Lemma A.4, the optimal covariance is given by (61),

$$\Sigma_\phi(Q) = Q^\top H^{-1}Q. \tag{68}$$

According to equations (49) and (50), we have the optimal mean of $q_\phi(z|y)$ as

$$\mu_Q(y) = \Sigma_\phi(Q)K_\theta^\top v(y), \tag{69}$$

where $v(y)$ is independent of $Q$. Specifically, by plugging the equation (69) and $K_\theta = U\Lambda Q$ into (69), we know the optimal solution $K_\phi(Q)$ and $b_\phi(Q)$ as equations (62) and (63), respectively.

Then we plug (68), (69) and $K_\theta = U\Lambda Q$ into the pointwise objective $J(y; \mu_Q, \Sigma_Q)$ (45). All terms related to $Q$ will be canceled out because $QQ^\top = Q^\top Q = I$, reducing $J(y; \mu_Q, \Sigma_Q)$ to an expression independent of $Q$. Therefore, for every $Q \in \mathbb{O}(d)$, the pair $(\theta(Q), \phi(Q))$ achieves the global minimum $\mathcal{L}_{\mathrm{opt}}$, forming the manifold $\mathcal{S}^{\mathrm{joint}}$. The separate training paradigm uniquely determines $Q_{\mathrm{sep}}$ during the pre-training of the generative map, restricting the solution to a singleton. $\square$

**Lemma A.10.** *For any $(\theta, \phi) \in \mathcal{S}^{\mathrm{joint}}$, the product of the generative and variational weight matrices equals the Kalman gain $K = CA^\top(ACA^\top + \sigma^2 I_{d_y})^{-1}$, i.e., $K_\theta K_\phi = K$. Furthermore, the expected output of the generative inference process recovers the exact Bayesian posterior mean,*

$$\mathbb{E}_{z \sim q_\phi(z|y)}[f_\theta(z)] = \mathbb{E}_{p_{\mathrm{data}}(x|y)}[x]. \tag{70}$$

*Specifically, this holds for the separate training where $\mathcal{S}^{\mathrm{sep}} = \{(\theta(Q_{\mathrm{sep}}), \phi(Q_{\mathrm{sep}}))\} \subset \mathcal{S}^{\mathrm{joint}}$ for a fixed $Q_{\mathrm{sep}} \in \mathbb{O}(d)$.*

*Proof.* Substituting $\Sigma_\phi$ from (42) into the expression for $K_\phi$ in (43), and applying the push-through identity $K_\theta(I_d + K_\theta^\top \mathcal{M}K_\theta)^{-1} = (I_d + K_\theta K_\theta^\top \mathcal{M})^{-1}K_\theta$ with $\mathcal{M} := \sigma^{-2}A^\top A + \tau^{-2}I_d$, we have

$$K_\theta K_\phi = (I_d + C(\sigma^{-2}A^\top A + \tau^{-2}I_d))^{-1}C\left(\sigma^{-2}A^\top + \tau^{-2}K\right)$$
$$= (C^{-1} + \sigma^{-2}A^\top A + \tau^{-2}I_d)^{-1}\left(\sigma^{-2}A^\top + \tau^{-2}K\right). \tag{71}$$

Using the identity of the Kalman gain, $(C^{-1} + \sigma^{-2}A^\top A)K = \sigma^{-2}A^\top$, and adding $\tau^{-2}K$ to both sides, we have

$$(C^{-1} + \sigma^{-2}A^\top A + \tau^{-2}I_d)K = \sigma^{-2}A^\top + \tau^{-2}K. \tag{72}$$

Left-multiplying by $(C^{-1} + \sigma^{-2}A^\top A + \tau^{-2}I_d)^{-1}$ and comparing with (71), we obtain $K_\theta K_\phi = K$.

Consider $\mathbb{E}_{z \sim q_\phi(z|y)}[f_\theta(z)] = K_\theta(K_\phi y + b_\phi) + b_\theta$. Since $b_\theta = m$ and $K_\theta K_\phi = K$, expanding $K_\theta b_\phi$ via (44) yields

$$K_\theta b_\phi = K_\theta \Sigma_\phi K_\theta^\top \left[-\sigma^{-2}A^\top Am + \tau^{-2}(I_d - KA)m - \tau^{-2}m\right]$$
$$= -(C^{-1} + \sigma^{-2}A^\top A + \tau^{-2}I_d)^{-1}(\sigma^{-2}A^\top A + \tau^{-2}KA)m$$
$$= -(C^{-1} + \sigma^{-2}A^\top A + \tau^{-2}I_d)^{-1}(\sigma^{-2}A^\top + \tau^{-2}K)Am = -KAm. \tag{73}$$

Therefore

$$\mathbb{E}_{z \sim q_\phi(z|y)}[f_\theta(z)] = Ky - KAm + m = m + K(y - Am) = \mathbb{E}_{p_{\mathrm{data}}(x|y)}[x]. \tag{74}$$

$\square$

**Proposition A.11.** *Assume that the observation operator $A$ and data covariance $C$ are in general position such that they are not simultaneously diagonalizable in the canonical basis. For separate training with $Q_{\text{sep}}$ uniformly randomly sampled from $\mathbb{O}(d)$, the following properties hold:*

1. **Sub-optimality of Separate Training:** $\mathcal{S}^{\text{sep}} \cap \mathcal{S}^{\text{sep}}_{\text{diag}} = \emptyset$ *a.s. w.r.t.* $\nu_{\mathbb{O}(d)}$ *(Definition A.1).*

2. **Optimality of Joint Training:** $\mathcal{S}^{\text{joint}} \cap \mathcal{S}^{\text{joint}}_{\text{diag}} \neq \emptyset$ *and* $\mathcal{S}^{\text{joint}}_{\text{diag}} \subset \mathcal{S}^{\text{joint}}$

*Proof.* For $Q \in \mathbb{O}(d)$ and $\theta(Q) \in \Theta^*$, recall from the proof in Lemma A.9 by

$$P(Q) := (\Sigma^*_\phi(Q))^{-1} = Q^\top H Q, \quad \text{where} \quad H := I_d + \frac{1}{\tau^2}\Lambda^2 + \frac{1}{\sigma^2}\Lambda U^\top A^\top A U \Lambda. \tag{75}$$

Let $\Sigma_\phi = \text{diag}(\sigma_1^2, \sigma_2^2, \ldots, \sigma_d^2) \in \mathbb{D}(d)$. The covariance-dependent objective $J(\Sigma_\phi)$ according to (45) and its minimizer $\Sigma^*_{\text{diag},\phi}$ are:

$$J(\Sigma_\phi) = \frac{1}{2}\sum_{i=1}^{d}\left(P_{ii}(Q)\sigma_i^2 - \ln \sigma_i^2\right) + \text{const}, \tag{76}$$

$$[\Sigma^*_{\text{diag},\phi}(Q)]^{-1} = \text{diag}(P(Q)) = \text{diag}(\Sigma^*_\phi(Q)^{-1}), \tag{77}$$

where $\Sigma^*_\phi(Q) = P(Q)^{-1}$. The optimality gap $\Delta J(Q)$ between $\Sigma^*_{\text{diag},\phi}(Q)$ and the unconstrained $\Sigma^*_\phi(Q) = P(Q)^{-1}$ is:

$$\begin{aligned}
\Delta J(Q) &:= J(\Sigma^*_{\text{diag},\phi}(Q)) - J(\Sigma^*_\phi(Q)) \\
&= \frac{1}{2}\left(\text{Tr}(P(Q)\Sigma^*_{\text{diag},\phi}(Q)) - \ln|\Sigma^*_{\text{diag},\phi}(Q)|\right) - \frac{1}{2}\left(\text{Tr}(P(Q)\Sigma^*_\phi(Q)) - \ln|\Sigma^*_\phi(Q)|\right) \\
&= \frac{1}{2}\left(\sum_{i=1}^{d}P_{ii}(Q)P_{ii}(Q)^{-1} + \ln\prod_{i=1}^{d}P_{ii}(Q)\right) - \frac{1}{2}\left(\text{Tr}(I_d) + \ln|P(Q)|\right) \\
&= \frac{1}{2}\left(d + \ln\prod_{i=1}^{d}P_{ii}(Q)\right) - \frac{1}{2}\left(d + \ln|P(Q)|\right) \\
&= \frac{1}{2}\ln\left(\frac{\prod_{i=1}^{d}P_{ii}(Q)}{|P(Q)|}\right).
\end{aligned} \tag{78}$$

By Hadamard's inequality, $\Delta J(Q) \geq 0$ and the equality holds if and only if $P(Q) \in \mathbb{D}(d)$.

In separate training, $Q_{\text{sep}}$ is fixed during pre-training and global optimality requires $P(Q_{\text{sep}}) = Q_{\text{sep}}^\top H Q_{\text{sep}} \in \mathbb{D}(d)$. By the general position assumption of $A$ and $C$, $H = I_d + \tau^{-2}\Lambda^2 + \sigma^{-2}\Lambda U^\top A^\top A U \Lambda$ is not a diagonal matrix. The set of matrices $\{Q \in \mathbb{O}(d) \mid Q^\top H Q \in \mathbb{D}(d)\}$ corresponds exclusively to the orthogonal matrices whose columns are the eigenvectors of $H$. Because this forms a finite set of permutation and sign-flip matrices, it holds a measure of zero with respect to the normalized Haar measure $\nu_{\mathbb{O}(d)}$ on the continuous manifold $\mathbb{O}(d)$ (refer to the Definition A.1).

According to the equality condition of the Hadamard's inequality, $P(Q_{\text{sep}}) \notin \mathbb{D}(d)$ a.s., leading to $\Delta J(Q) > 0$, which implies that the optimal loss value reached by the diagonal constrained $\Sigma^*_{\text{diag},\phi}(Q)$ is bigger than $\mathcal{L}_{\text{opt}}$. According to Lemma A.9, $\mathcal{S}^{\text{sep}}$ has the optimal loss $\mathcal{L}_{\text{opt}}$. Therefore $\mathcal{S}^{\text{sep}} \cap \mathcal{S}^{\text{sep}}_{\text{diag}} = \emptyset$.

In joint training, $Q$ is a learnable parameter optimized over $\mathbb{O}(d)$. The global minimum $\mathcal{L}_{\text{opt}}$ under the diagonal constraint is achieved if and only if the optimality gap vanishes, $\Delta J(Q) = 0$. By Hadamard's inequality, this condition holds if and only if $P(Q) = Q^\top H Q \in \mathbb{D}(d)$, which restricts $Q$ to the set of eigen-bases $\mathcal{V}(H) := \{Q \in \mathbb{O}(d) \mid Q^\top H Q \in \mathbb{D}(d)\}$. For any $Q \in \mathcal{V}(H)$, the optimal variational covariance $\Sigma^*_\phi(Q) = P(Q)^{-1}$ inherently belongs to $\mathbb{D}(d)$. Because these specific configurations satisfy the diagonal constraint while simultaneously achieving the unconstrained global minimum, it follows that $\mathcal{S}^{\text{joint}}_{\text{diag}} = \{(\theta(Q), \phi(Q)) \mid Q \in \mathcal{V}(H)\} \subset \mathcal{S}^{\text{joint}}$, thereby confirming $\mathcal{S}^{\text{joint}} \cap \mathcal{S}^{\text{joint}}_{\text{diag}} \neq \emptyset$. $\qquad\square$

**Lemma A.12.** *Let $\mathbb{O}(d)$ be the orthogonal group equipped with the normalized Haar measure $\nu_{\mathbb{O}(d)}$. Let $Sym_0(d) := \{M \in \mathbb{R}^{d\times d} \mid M = M^\top, diag(M) = 0\}$. Define the map $G : \mathbb{O}(d) \to Sym_0(d)$ by $G(Q) = QHQ^\top - diag(QHQ^\top)$,*

*where $H \in \mathbb{R}^{d \times d}$ is a fixed symmetric matrix with distinct eigenvalues. Let $V \subset \mathbb{R}^{d \times d}$ be a proper subspace such that $\mathrm{Sym}_0(d) \not\subseteq V$. Then the set*

$$S := \{Q \in \mathbb{O}(d) \mid G(Q) \in V\}$$

*has measure $\nu_{\mathbb{O}(d)}(S) = 0$.*

*Proof.* The orthogonal group $\mathbb{O}(d)$ is a compact real analytic manifold. Let $\mathfrak{so}(d) = \{B \in \mathbb{R}^{d \times d} \mid B = -B^\top\}$ denote its Lie algebra. The map $G$ is real analytic since its entries are polynomial functions of the elements of $Q$. Let $P_{V^\perp}$ be the projection operator onto the orthogonal complement of $V$. The condition $G(Q) \in V$ is equivalent to $f(Q) := P_{V^\perp} G(Q) = 0$. Now, $f = P_{V^\perp} \circ G$ is a composition of a linear projection and a polynomial map, which is real analytic on the manifold. Therefore $\nu_{\mathbb{O}(d)}(S) = 0$ follows if $f$ is not identically zero on the connected components of $\mathbb{O}(d)$ by the identity theorem.

Since $H$ is symmetric with distinct eigenvalues, there exists $Q_0 \in \mathbb{O}(d)$ such that $Q_0 H Q_0^\top = \Lambda = \mathrm{diag}(\lambda_1, \ldots, \lambda_d)$, where $\lambda_i \neq \lambda_j$ for $i \neq j$. We evaluate the differential $\mathrm{D}G(Q_0)$ by considering the variation $Q(\epsilon) = e^{\epsilon B} Q_0$ for $B \in \mathfrak{so}(d)$. The directional derivative at $Q_0$ is given by

$$\mathrm{D}G(Q_0)[B] = [B, \Lambda] - \mathrm{diag}([B, \Lambda]).$$

For the off-diagonal entries $i \neq j$, the commutator yields $[B, \Lambda]_{ij} = \sum_k (B_{ik} \Lambda_{kj} - \Lambda_{ik} B_{kj}) = B_{ij} \lambda_j - \lambda_i B_{ij} = (\lambda_j - \lambda_i) B_{ij}$. For the diagonal entries, $[B, \Lambda]_{ii} = B_{ii} \lambda_i - \lambda_i B_{ii} = 0$, which implies $\mathrm{diag}([B, \Lambda]) = 0$. Thus, for any $i \neq j$, we have

$$(\mathrm{D}G(Q_0)[B])_{ij} = (\lambda_j - \lambda_i) B_{ij}.$$

Given that $\{\lambda_i\}$ are pairwise distinct, for any target matrix $M \in \mathrm{Sym}_0(d)$, we can uniquely determine $B \in \mathfrak{so}(d)$ by setting $B_{ij} = M_{ij}/(\lambda_j - \lambda_i)$ for $i < j$. This proves that the differential $\mathrm{D}G(Q_0) : \mathfrak{so}(d) \to \mathrm{Sym}_0(d)$ is a linear isomorphism.

Since $\mathrm{D}G(Q_0)$ is an isomorphism onto $\mathrm{Sym}_0(d)$ and $\mathrm{Sym}_0(d) \not\subseteq V$, there exists $B \in \mathfrak{so}(d)$ such that $\mathrm{D}G(Q_0)[B] \notin V$. It follows that $P_{V^\perp} \mathrm{D}G(Q_0)[B] \neq 0$, implying that $f$ is not identically zero in a neighborhood of $Q_0$. By the identity theorem for real analytic functions, the zero set $S \cap \mathbb{O}(d)^\circ$ has Haar measure zero, where $\mathbb{O}(d)^\circ$ denotes the connected component containing $Q_0$. A similar argument holds for the remaining connected component of $\mathbb{O}(d)$ since $\mathbb{O}(d)$ has two connected components, i.e. $|Q| = 1$ and $|Q| = -1$. $\qquad\square$

**Proposition A.13** (Mean Recovery Gap under Diagonal Constraint). *Assuming $A$ and $C$ are in general position, the following properties hold under the diagonal constraint $\Sigma_\phi \in \mathbb{D}(d)$:*

1. ***Separate Training:*** *For any $(\theta, \phi) \in \mathcal{S}_{\mathrm{diag}}^{\mathrm{sep}}$, the inference process fails to recover the posterior mean almost surely:*

$$\mathbb{E}_{z \sim q_\phi(z|y)}[f_\theta(z)] \neq \mathbb{E}_{p_{\mathrm{data}}(x|y)}[x] \quad \textit{a.s. w.r.t. } y \sim \mathcal{N}(Am, \Sigma_y), \ Q \sim \nu_{\mathbb{O}(d)}. \tag{79}$$

2. ***Joint Training:*** *For any $(\theta, \phi) \in \mathcal{S}_{\mathrm{diag}}^{\mathrm{joint}}$, the inference process recovers the exact posterior mean:*

$$\mathbb{E}_{z \sim q_\phi(z|y)}[f_\theta(z)] = \mathbb{E}_{p_{\mathrm{data}}(x|y)}[x]. \tag{80}$$

*Proof.* Let $\mathrm{Sym}_0(d) := \{M \in \mathbb{R}^{d \times d} \mid M = M^\top, \mathrm{diag}(M) = 0\}$. The expected reconstruction is $\hat{x} = \mathbb{E}_{z \sim q_\phi(z|y)}[f_\theta(z)] = K_\theta K_\phi y + K_\theta b_\phi + b_\theta$, while the analytical posterior mean is $\mathbb{E}[x|y] = m + K(y - Am)$. In the separate training paradigm, where $(\theta(Q_{\mathrm{sep}}), \phi(Q_{\mathrm{sep}})) \in \mathcal{S}_{\mathrm{diag}}^{\mathrm{sep}}$, the rotation $Q_{\mathrm{sep}}$ is fixed. Let $P := Q_{\mathrm{sep}} H Q_{\mathrm{sep}}^\top$ and denote $E := [\mathrm{diag}(P)]^{-1} P - I$. Under the diagonal constraint, the gain matrix becomes $K_{\mathrm{diag}} = K_\theta [\mathrm{diag}(P)]^{-1} K_\theta^\top (\sigma^{-2} A^\top + \tau^{-2} K)$. The recovery error simplifies to

$$\hat{x} - \mathbb{E}[x|y] = (K_{\mathrm{diag}} - K)(y - Am) = K_\theta E K_\theta^{-1} K(y - Am). \tag{81}$$

Since $A$ and $C$ are in general position, $Q_{\mathrm{sep}}$ does not diagonalize $H$ almost surely, implying that $P$ is non-diagonal and thus $E$ is a non-zero matrix with a vanishing diagonal. In addition, this general position assumption implies that $H$ has distinct eigenvalues and that $K = CA^\top (ACA^\top + \sigma^2 I)^{-1}$ has rank $d_y$.

Now define $V := \{M \in \mathrm{Sym}_0(d) \mid K_\theta M K_\theta^{-1} K = 0\}$ as a subspace of $\mathbb{R}^{d \times d}$. Since $K_\theta$ is invertible due to $K_\theta K_\theta^\top = C$, the condition $M \in V$ is equivalent to $M(K_\theta^{-1} K) = 0$. By the general position assumption, $K$ is non-zero, meaning the

matrix $K_\theta^{-1}K$ contains at least one non-zero column $w$. If $Mw = 0$ for all $M \in \mathrm{Sym}_0(d)$, then applying symmetric matrices $M$ with a single pair of off-diagonal ones (and zeros elsewhere) would force all components of $w$ to be zero, contradicting $w \neq 0$. Thus, there exists some $M \in \mathrm{Sym}_0(d)$ such that $MK_\theta^{-1}K \neq 0$, ensuring that $V$ is a proper subspace of $\mathrm{Sym}_0(d)$ (i.e., $V \subsetneq \mathrm{Sym}_0(d)$). Therefore we can use the Lemma A.12 for $H$ and $V$ to show that

$$\nu_{\mathbb{O}(d)}(\{Q \in \mathbb{O}(d) | QHQ^\top - \mathrm{diag}(QHQ^\top) \in V\}) = 0, \tag{82}$$

To connect this with the recovery error, let $M_{\mathrm{sep}} = Q_{\mathrm{sep}}HQ_{\mathrm{sep}}^\top - \mathrm{diag}(Q_{\mathrm{sep}}HQ_{\mathrm{sep}}^\top)$. By definition, the error matrix $E$ satisfies $E = [\mathrm{diag}(P)]^{-1}M_{\mathrm{sep}}$. Because $[\mathrm{diag}(P)]^{-1}$ is an invertible diagonal matrix, the condition $K_\theta EK_\theta^{-1}K = 0$ holds if and only if $M_{\mathrm{sep}}K_\theta^{-1}K = 0$, which is exactly $M_{\mathrm{sep}} \in V$. According to (82), we have

$$\nu_{\mathbb{O}(d)}(\{Q_{\mathrm{sep}} \in \mathbb{O}(d) \mid K_\theta EK_\theta^{-1}K = 0\}) = 0. \tag{83}$$

This ensures that for almost every $Q_{\mathrm{sep}}$ sampled from $\mathbb{O}(d)$, the linear mapping matrix $K_\theta EK_\theta^{-1}K$ is strictly non-zero. Consequently, the null space $\{y \in \mathbb{R}^{d_y} \mid K_\theta EK_\theta^{-1}K(y - Am) = 0\}$ constitutes a proper affine subspace of $\mathbb{R}^{d_y}$ with dimension strictly less than $d_y$. Since the marginal distribution $p(y) = \mathcal{N}(y|Am, \Sigma_y)$ is a non-degenerate continuous Gaussian, it assigns zero probability mass to any strictly lower-dimensional subspace. It follows directly that the recovery error $\hat{x} - \mathbb{E}[x|y] \neq 0$ almost surely with respect to the joint measure of $p(y)$ and $\nu_{\mathbb{O}(d)}$, i.e.

$$\mathbb{E}_{z \sim q_\phi(z|y)}[f_\theta(z)] \neq \mathbb{E}_{p_{\mathrm{data}}(x|y)}[x] \quad \text{a.s. w.r.t. } y \sim \mathcal{N}(Am, \Sigma_y), \ Q \sim \nu_{\mathbb{O}(d)}. \tag{84}$$

For joint training, Proposition A.11 establishes that $\mathcal{S}_{\mathrm{diag}}^{\mathrm{joint}} \subset \mathcal{S}^{\mathrm{joint}}$. Since every pair in $\mathcal{S}^{\mathrm{joint}}$ satisfies the unconstrained optimality condition $\hat{x} = \mathbb{E}[x|y]$ by Lemma A.10, the identity holds for all $(\theta, \phi) \in \mathcal{S}_{\mathrm{diag}}^{\mathrm{joint}}$ for all $y \in \mathbb{R}^{d_y}$, i.e.

$$\mathbb{E}_{z \sim q_\phi(z|y)}[f_\theta(z)] = \mathbb{E}_{p_{\mathrm{data}}(x|y)}[x]. \tag{85}$$

$\square$

*Remark* A.14 (Coordinate Alignment and Non-linear Extensions). Propositions A.11 and A.13 characterize the interaction between the generative map and the amortized inference network under structural constraints. In the separate training paradigm, the fixed generative map imposes a rigid coordinate system in the latent space. Restricting the variational posterior to a diagonal covariance $\Sigma_\phi$ forces it to approximate a structurally dense precision matrix $P(Q_{\mathrm{sep}})$, which inherently induces a systematic recovery gap. Joint training resolves this limitation by optimizing the orthogonal matrix $Q \in \mathbb{O}(d)$ to align the principal axes of the posterior precision with the canonical basis of the prior. This alignment guarantees that the diagonal parameterization attains the unconstrained global minimum $\mathcal{L}_{\mathrm{opt}}$.

Furthermore, this geometric alignment property extends to non-linear generative models. During joint optimization, the generator adapts its representation such that the local geometry of the data distribution corresponds with the inductive bias of the variational distribution. By adjusting its Jacobian $\nabla_z f_\theta(z)$, the generator can approximately diagonalize the pull-back metric in the latent space, providing a mathematical justification for the deployment of factorized posterior approximations in more general inference settings.

### A.3. Proof of Proposition 3.2

First, noting that $f_\theta(z) = z - u_\theta(z, 0, 1)$, we have

$$\|x - f_\theta(z)\|^2 = \|x - (z - u_\theta(z, 0, 1))\|^2 = \|u_\theta(z, 0, 1) - (z - x)\|^2. \tag{86}$$

Then, by Jensen's inequality, and recalling that $\psi_t(x, z) := tz + (1 - t)x$, we get

$$\int_0^1 \|\partial_t \mathcal{E}_\theta(x, z, 0, t)\|^2 dt \tag{87}$$

$$= \int_0^1 \left\| \frac{d}{dt} \left[ tu_\theta(\psi_t(x, z), 0, t) - \int_0^t \dot{\psi}_t(x, z) ds \right] \right\|^2 dt \tag{88}$$

$$\overset{\text{Jensen}}{\geq} \left\| \int_0^1 \frac{d}{dt} \left[ tu_\theta(\psi_t(x, z), 0, t) - t(z - x) \right] dt \right\|^2 \tag{89}$$

$$= \|u_\theta(z, 0, 1) - (z - x)\|^2. \tag{90}$$

Putting these together, we establish our desired bound. $\square$

## A.4. Proof of Proposition 3.4

From our assumptions, we can compute

$$p_\tau(x|y) := \int_{\mathbb{R}^d} \mathcal{N}(x|f_\theta(z), \tau^2 I) p(z|y) dz, \tag{91}$$

where

$$p(z|y) = \frac{\mathcal{N}(y|Af_\theta(z), \sigma^2 I) p(z)}{\int_{\mathbb{R}^d} \mathcal{N}(y|Af_\theta(z), \sigma^2 I) p(z) dz}. \tag{92}$$

Denoting by $\mu_\tau^y(dx) := p_\tau(x|y) dx$ and $\nu^y(dz) := p(z|y) dz$ the posterior measures in $x$ and $z$ spaces, respectively, for any $g \in C_b(\mathbb{R}^d)$, we have

$$\int_{\mathbb{R}^d} g(x) \mu_\tau^y(dx) \overset{(91)}{=} \int_{\mathbb{R}^d} \int_{\mathbb{R}^d} g(x) \mathcal{N}(x|f_\theta(z), \tau^2 I) \nu^y(dz) dx \tag{93}$$

$$\overset{\tau \to 0}{\longrightarrow} \int_{\mathbb{R}^d} g(f_\theta(z)) \nu^y(dz) \tag{94}$$

$$= \int_{\mathbb{R}^d} g(x) (f_\theta)_\sharp \nu^y(dx), \tag{95}$$

where we used the standard result that $\mathcal{N}(x|f_\theta(z), \tau^2 I)$ converges weakly to the delta measure around $f_\theta(z)$ as $\tau \to 0$ (Billingsley, 2013), and we used the dominated convergence theorem and Fubini's theorem, both justified by the bound

$$\left| \int_{\mathbb{R}^d} g(x) \mathcal{N}(x|f_\theta(z), \tau^2 I) dx \right| \leq \|g\|_\infty. \tag{96}$$

This proves the weak convergence of measures $\mu_\tau^y \Rightarrow (f_\theta)_\sharp \nu^y$ as $\tau \to 0$. $\qquad \square$

# B. Experimental Details

## B.1. 2D Checkerboard Data

We use a 2D checkerboard distribution supported on alternating squares in $[-2, 2]^2$. To sample, we first draw $u \sim \text{Unif}([0, 1]^2)$ and partition the unit square into a $4 \times 4$ uniform grid. Then, we accept samples that lie on one of the checkerboard cells. Finally, we center and scale via $x = 4(u - (0.5, 0.5))$, so the support lies in $[-2, 2]^2$ and each retained square has side length $1$. We used $20,000$ samples from this distribution to train our models.

### B.1.1. MODEL ARCHITECTURES

For the mean-flow network $u_\theta$, we use a SiLU MLP with six layers and width 512. We initialize this model from a flow-matching velocity network pretrained on the checkerboard samples. The noise adapter is a smaller SiLU MLP with four layers and width 256, trained from scratch. Each model is trained for 50,000 iterations with batch size 2048 using the `AdamW` optimizer with learning rate $2 \times 10^{-4}$ and weight decay $1 \times 10^{-4}$.

### B.1.2. PROBLEM FORMULATION

The task in this experiment is to solve the Bayesian inverse problem

$$p(x|y) \propto \exp\left(-\frac{|y - Ax|^2}{2\sigma^2}\right) p(x), \tag{97}$$

where $p(x)$ is the 2D checkerboard distribution and the forward operator is given by $A = \begin{pmatrix} 1 & 0 \end{pmatrix}$, that is, observing only the first component. For the observation noise, we take $\sigma = 0.1$.

### B.1.3. METRICS

To evaluate our results, we use the following metrics.

**Negative log predictive density (NLPD).** Given an observation $y \in \mathbb{R}$ and posterior samples $\{x^{(j)}\}_{j=1}^J$ with $x^{(j)} \sim p(x|y)$, the predictive density is approximated by Monte Carlo:

$$p(y'|y) = \int p(y' \mid x)\, p(x|y)\, dx \approx \frac{1}{J}\sum_{j=1}^J \mathcal{N}(y'|Ax^{(j)},\ \sigma^2), \tag{98}$$

where $y'$ is a fresh observation independent of $y$. We report the *negative log predictive density (NLPD)*,

$$\mathrm{NLPD}(y'; y) = -\log p(y'|y) \approx -\log\left(\frac{1}{J}\sum_{j=1}^J \mathcal{N}(y'|Ax^{(j)},\ \sigma^2)\right), \tag{99}$$

which is a proper scoring rule. To sample from $p(x|y)$ approximately using VFM, we first sample $z^{(j)} \sim q_\phi(z|y)$ and then set $x^{(j)} = f_\theta(z^{(j)})$. We report the averaged NLPD over a batch $\{y_b', y_b, \{x_b^{(j)}\}_{j=1}^J\}_{b=1}^B$. We take $B = 10,000$ and $J = 100$.

**Continuous ranked probability score (CRPS).** Given ground-truth targets $x^\dagger \in \mathbb{R}^2$ and $J$ predictive samples $\{x^{(j)}\}_{j=1}^J$ corresponding to an observation $y^\dagger = Ax^\dagger + \varepsilon^\dagger$ for some noise realisation $\varepsilon^\dagger$ (i.e. we take $x^{(j)} = f_\theta(z^{(j)})$ for $z^{(j)} \sim q_\phi(z|y^\dagger)$), we estimate the CRPS as:

$$\mathrm{CRPS}(x^\dagger; y^\dagger) \approx \frac{1}{J}\sum_{j=1}^J \|x^{(j)} - x^\dagger\| - \frac{1}{2J^2}\sum_{j=1}^J\sum_{k=1}^J \|x^{(j)} - x^{(k)}\|. \tag{100}$$

The first term measures the average distance of samples to the truth, while the second term rewards diversity. We report the averaged CRPS over a batch $\{x_b^\dagger, y_b^\dagger, \{x_b^{(j)}\}_{j=1}^J\}_{b=1}^B$. We take $B = 10,000$ and $J = 100$.

**Maximum mean discrepancy (MMD).** To compare two measures $\mu_P$ and $\mu_Q$, we can compute their maximum mean discrepancy, which is a distance on the space of measures, whose square is given by (Gretton et al., 2012)

$$\mathrm{MMD}^2(X, Y) = \mathbb{E}[k(x, x')] + \mathbb{E}[k(y, y')] - 2\,\mathbb{E}[k(x, y)], \tag{101}$$

with $x, x' \sim \mu_P$ and $y, y' \sim \mu_Q$ i.i.d., and $k(\cdot, \cdot)$ is a choice of kernel such as the squared exponential kernel

$$k(u, v) := \exp\left(-\frac{\|u - v\|_2^2}{2\ell^2}\right). \tag{102}$$

In practice, we use the unbiased estimator:

$$\widehat{\mathrm{MMD}}^2 = \frac{1}{N(N-1)}\sum_{i \neq i'} k(x^{(i)}, x^{(i')}) + \frac{1}{M(M-1)}\sum_{j \neq j'} k(y^{(j)}, y^{(j')}) - \frac{2}{NM}\sum_{i=1}^N\sum_{j=1}^M k(x^{(i)}, y^{(j)}), \tag{103}$$

For the lengthscale hyperparameter $\ell$, we choose the median heuristic computed from pairwise distances between samples. In our computations, we choose $N = M = 10,000$ samples to compare the prior distributions and the posterior distributions. Here, our true prior distribution is the checkerboard distribution, and our approximate prior is obtained by $\{f_\theta(z)\}_{z \sim \mathcal{N}(0,I)}$. For the true posterior, we compute it using rejection sampling (see Algorithm 3) and the approximate posterior is obtained by $\{f_\theta(z)\}_{z \sim q_\phi(z|y)}$.

**Support accuracy (SACC).** We measure *support accuracy* as the proportion (percentage) of generated samples that fall inside one of the filled checkerboard squares. Concretely, for samples $\{x^{(j)}\}_{j=1}^J$, we compute

$$\mathrm{Acc}(\{x^{(j)}\}_{j=1}^J) = \frac{1}{J}\sum_{j=1}^J \mathbf{1}\Big[x^{(j)} \text{ lies in a checkerboard cell}\Big]. \tag{104}$$

We compute the support accuracy for both prior samples $\{f_\theta(z)\}_{z \sim \mathcal{N}(0,I)}$ and posterior samples $\{f_\theta(z)\}_{z \sim q_\phi(z|y)}$.

---

**Algorithm 3** Rejection sampling for $p(x \mid y)$

---

1: **Input:** observation $y \in \mathbb{R}$, noise $\sigma > 0$, number of samples $J$, prior $p(x)$
2: Initialize accepted set $\mathcal{S} \leftarrow \varnothing$
3: **while** $|\mathcal{S}| < J$ **do**
4:     Propose $x \sim p(x)$
5:     Compute $a \leftarrow \exp\left(-\frac{(y - Ax)^2}{2\sigma^2}\right)$
6:     Draw $u \sim \mathrm{Unif}(0, 1)$
7:     **if** $u < a$ **then**
8:         Append $x$ to $\mathcal{S}$
9:     **end if**
10: **end while**
11: **Output:** $\{x^{(j)}\}_{j=1}^{J} \leftarrow \mathcal{S}$

---

### B.1.4. ABLATION PLOTS

- Figure 6: Ablation of VFM for all metrics with respect to the parameter $\tau$. The parameter $\alpha$ is set to 0.5. We also display the results of the `frozen`$-\theta$ baseline for reference.

- Figure 7: Ablation of VFM for all the metrics with respect to $\tau$. The parameter $\alpha$ is set to 1.0. We also display the results of the `frozen`$-\theta$ baseline for reference.

- Figure 8: Plots displaying the noise-to-data alignment in VFM with or without various modeling choices in the loss to isolate their effects on the final results. In particular, we consider: (1) `frozen`$-\theta$, (2) `unconstrained`$-\theta$, (3) VFM with no EMA, (4) VFM without KL loss.

- Figure 9: Plots displaying how the noise-to-data alignment for VFM changes with respect to $\tau$. Here, $\alpha$ is set to 1.0.

- Figure 10: Plots displaying how the noise-to-data alignment for VFM changes with respect to $\alpha$. Here, $\tau$ is set to 100.0.

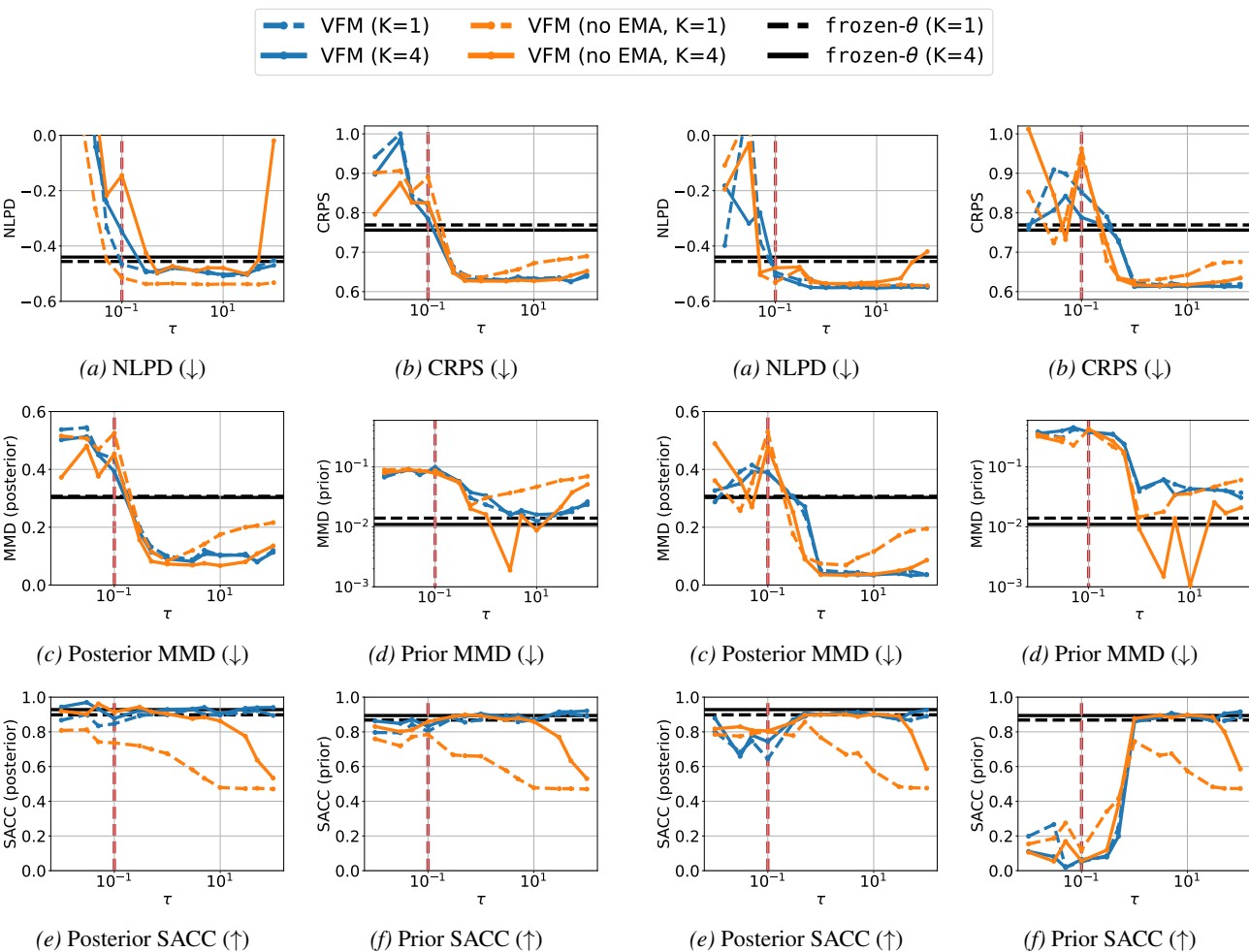

*Figure 6.* Metrics for VFM with $\alpha = 0.5$ and varying $\tau$. Dashed vertical (red) line indicates the reference value $\sigma = 0.1$. The baseline model (black lines) is frozen$-\theta$. We compare the results of VFM with EMA used in the observation loss term (blue lines) vs. without using EMA (orange line) for $K = 1, 4$.

*Figure 7.* Metrics for VFM with $\alpha = 1.0$ and varying $\tau$. Dashed vertical (red) line indicates the reference value $\sigma = 0.1$. The baseline model (black lines) is frozen$-\theta$. We compare the results of VFM with EMA used in the observation loss term (blue lines) vs. without using EMA (orange line) for $K = 1, 4$.

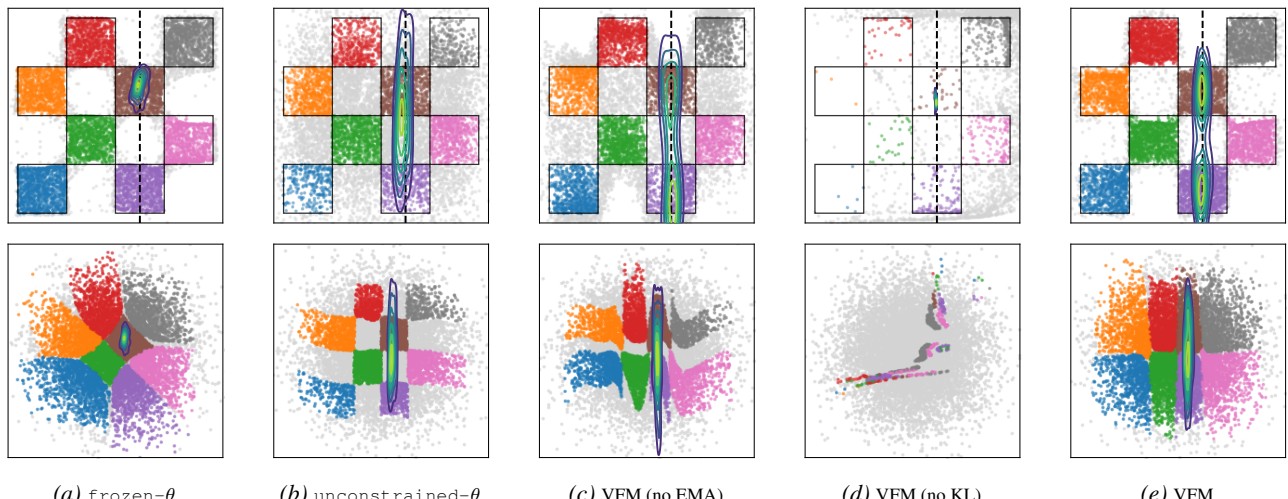

*(a)* `frozen-`$\theta$     *(b)* `unconstrained-`$\theta$     *(c)* VFM (no EMA)     *(d)* VFM (no KL)     *(e)* VFM

*Figure 8.* Ablation of VFM with respect to key modeling choices in the loss. Observation in black dots and $\sigma = 0.1$. For VFM (8c, 8d, 8e), we used $\tau = 100.0$, $\alpha = 1.0$ and $K = 4$. We observe that 8a: `frozen-`$\theta$ fails to capture the bimodal nature of the posterior; 8b: `unconstrained-`$\theta$ produces many off-manifold samples; 8c: removing EMA from the term $\mathcal{L}_{\mathrm{obs}}$ in VFM also produces many off-manifold samples when $\tau$ is large; 8d: removing the KL term $\mathcal{L}_{\mathrm{KL}}$ in the VFM loss leads to unstable optimization and results in poor approximations of both the prior and posterior.

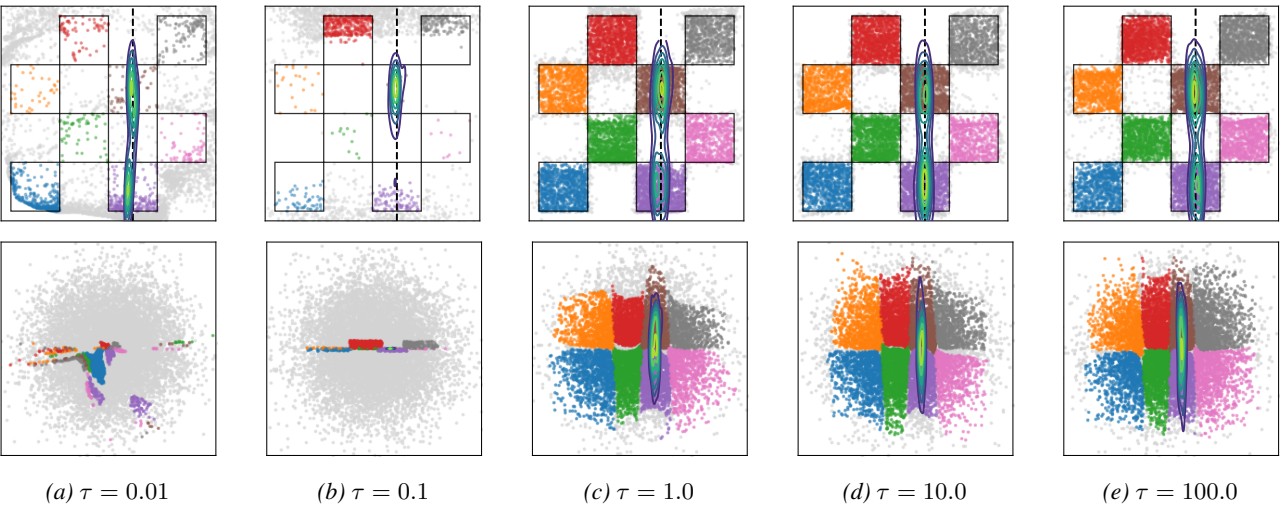

*(a)* $\tau = 0.01$     *(b)* $\tau = 0.1$     *(c)* $\tau = 1.0$     *(d)* $\tau = 10.0$     *(e)* $\tau = 100.0$

*Figure 9.* Ablation of VFM with respect to the $\tau$ parameter. For each plot, we set $\alpha = 1.0$ and $K = 4$. We fix $y = 0.5$ and $\sigma = 0.1$. We observe that for $\tau \lesssim \sigma$, the quality of prior/posterior approximations are poor, yielding many off-manifold samples. This is likely due to the difficulty of optimization as we tighten the correspondence between $x$ and $z$. For $\tau \geq 1$, we observe significant improvements in results and surprising robustness with respect to large values of $\tau$.

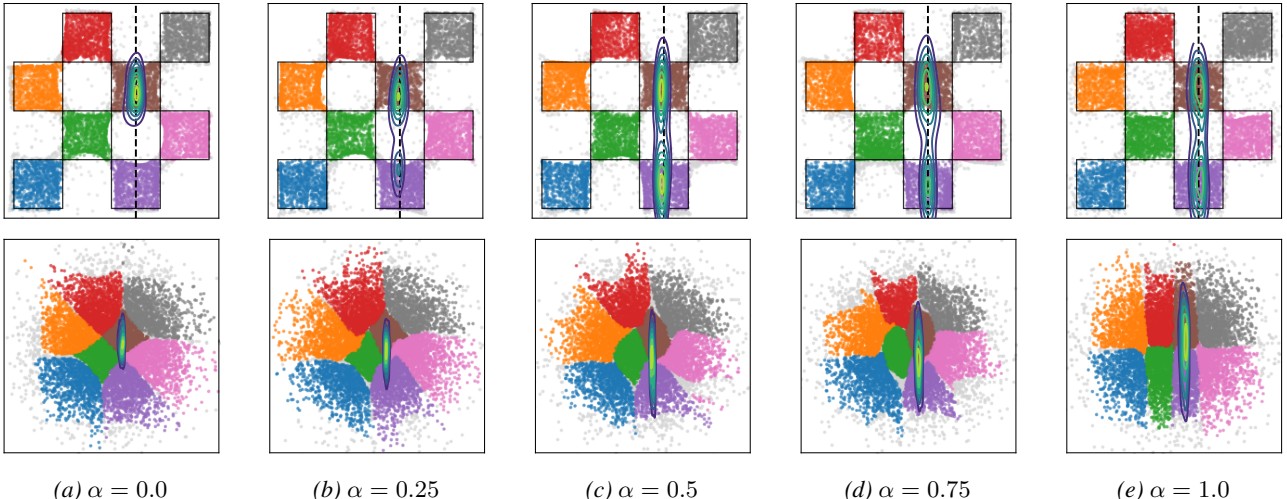

*(a)* $\alpha = 0.0$  *(b)* $\alpha = 0.25$  *(c)* $\alpha = 0.5$  *(d)* $\alpha = 0.75$  *(e)* $\alpha = 1.0$

*Figure 10.* Ablation of VFM with respect to the $\alpha$ parameter. For each plot, we set $\tau = 100.0$ and $K = 4$. We observe that the warping of the latent space becomes stronger as $\alpha \to 1$, making it easier to sample from the bimodal posterior using the simple Gaussian variational posterior in latent space.

## B.2. ImageNet experiment

In this section, we provide a detailed breakdown of the architectures, training objectives, and the extensive tuning process conducted for the baselines used in the ImageNet $256 \times 256$ experiments.

### B.2.1. MODEL ARCHITECTURES

**Flow Map Backbone ($f_\theta$).** We employ a SiT-B/2 architecture (Ma et al., 2024) (130M parameters) initialized from a flow-matching model pre-trained for 80 epochs. Following the design of Decoupled Mean Flow (DMF) (Lee et al., 2025), we utilize decoupled encoder/decoder embeddings for the timesteps to better capture the flow dynamics. Our fine-tuning is performed for 50 epochs (30 epochs using only the mean flow loss + 20 epochs with the full VFM loss), making the total training process to be 130 epochs.

**Noise Adapter ($q_\phi$).** To map high-dimensional observations $y$ and inverse problem classes $c$ to the latent noise distribution $q_\phi(z|y, c)$, we design a lightweight U-Net style adapter (10M parameters). The adapter is conditioned on the inverse problem class $c$ using Feature-wise Linear Modulation (FiLM) (Perez et al., 2018). The class embedding modifies the features at multiple resolutions via affine transformations $\gamma \cdot x + \beta$. The network processes the $256 \times 256$ input observation through a series of residual blocks and downsampling layers (channel multipliers: 1, 2, 4, 4), which compresses the spatial resolution to $32 \times 32$. The final projection layer outputs the mean $\mu$ and log-variance $\log \sigma^2$ (clamped between -10.0 and 2.0) for the latent distribution, from which we sample $z$ using the reparameterization trick.

**Latent space encoding.** Modern generative models often operate in a lower-dimensional latent space obtained via an autoencoder or similar compression mechanism (Rombach et al., 2022). We adopt this setting in this experiment by defining the flow map and adapter in the latent space of SD-VAE (Rombach et al., 2022) rather than pixel space, and applying the forward operator to decoded samples. Measurement encoding can be incorporated directly into the adapter architecture; specifically, our U-Net-based architecture for the adapter maps the high-dimensional input observations into a lower-dimensional latent representation, which allows us to estimate the mean ($\mu$) and variance ($\sigma$) within the latent space.

**Training.** For VFM training, we employ standard model guidance techniques during the flow map training phase. Following previous works (Tang et al., 2025; Lee et al., 2025), we utilize a prefixed CFG probability to redefine the target velocity, which allows us to perform robust one-step generation during sampling. After extensive experiments and ablations on the $\tau$ parameter, we found that setting the coefficient of $\mathcal{L}_{data}$ to 1.0 works best in practice. All training and inference were conducted using 8 and 1 NVIDIA GH200 GPUs, respectively.

### B.2.2. BASELINES AND TUNING

We compare VFM against a comprehensive suite of guidance-based solvers. A major challenge in this comparison is the high sensitivity of these methods to hyperparameters. To ensure a fair comparison, we performed an exhaustive hyperparameter sweep for every baseline, task, and backbone. We found that most inference-time methods require significant per-task tuning, which makes them computationally burdensome compared to the one-step nature of VFM.

Unless otherwise stated, all baselines use 250 ODE steps. To maximize their performance, we also applied Classifier-Free Guidance (CFG) with a scale of 2.0, which we found empirically boosts results across methods, even those that do not originally prescribe it.

**Latent DPS (Chung et al., 2024).** We extend Diffusion Posterior Sampling (DPS) to the latent flow matching setting. Through extensive sweeping, we identified a novel gradient scaling technique that provided the best stability. We normalize the likelihood gradient update to have a magnitude of 1, i.e., using a step size of $1/||\nabla_z \log p(y|z)||$.

**Latent DAPS (Zhang et al., 2025).** We implemented DAPS in the latent flow matching space, strictly following the original paper's settings. This involves 5 ODE rollout steps followed by 50 annealing steps and 50 Langevin steps, which makes the optimization extremely slow.

**PSLD (Rout et al., 2023)** Our implementation follows the original paper. We tuned the coefficients and found the optimal values to match the original recommendations, where DPS and gluing coefficients are chosen to be 1.0 and 0.1, respectively.

**MPGD (He et al., 2023).** We extended Manifold Preserving Guidance (MPGD) to flow matching, which approximates the Jacobian as identity. We utilized DDIM-type deterministic velocity maps and, similar to DPS, found that a gradient scaling of $1/||\nabla||$ yielded the best performance.

**FlowChef (Patel et al., 2025).** We followed the exact implementation from the original paper. After heavy tuning, we found that it behaved similarly to MPGD and performed best with the $1/||\nabla||$ gradient scaling.

**FlowDPS (Kim et al., 2025).** We followed the official implementation. Tuning revealed that a larger step size coefficient of $10.0/||\nabla||$ was optimal. We adhered to the original protocol of repeating the update 3 times per ODE iteration. Importantly, we disabled the stochasticity parameter as it was found to degrade performance, instead we relied on deterministic velocity updates.

**LFlow (Askari et al., 2026).** Our implementation follows the original paper. Due to LFlow's appeal of being robust to hyperparameters, we used the recommended setting in the paper with $K = 2$ and $t_s = 0.8$. We note that this method is developed specifically for linear inverse problems, hence we do not use this baseline for nonlinear problems.

### B.2.3. METRICS AND EVALUATION

We evaluate performance using two distinct categories of metrics:

**Pixel-Space Fidelity (PSNR/SSIM).** While we report these standard metrics, we note that inference-time optimization methods (like DPS) tend to produce smooth estimates that maximize these scores by converging toward the conditional mean. This often results in a loss of high-frequency texture and realistic detail (Zhang et al., 2018).

**Semantic and Distributional Fidelity (LPIPS, FID, MMD, CRPS).** To assess whether the model captures the true posterior distribution rather than just the mean, we prioritize metrics in embedding space. We evaluate methods by using standard LPIPS and FID by using 1024 reconstructions from the validation set of ImageNet. We further evaluate Maximum Mean Discrepancy (MMD) metric in the embedding space of Inception network (used also in FID). This measures the distance between the true and approximate posterior distributions in the semantic space. To evaluate the generation quality along with its diversity (which is very important in posterior sampling and uncertainty quantification), we also use the Continuous Ranked Probability Score (CRPS) scoring rule. It assesses the calibration and coverage of the posterior. We compute this in the embedding spaces of both Inception and DINO models to ensure semantic consistency. Refer to Appendix B.1.3 for further details on the computation of MMD and CRPS.

We evaluate PSNR, SSIM, LPIPS, FID, and MMD on the randomly selected 1024 samples from the validation set of the ImageNet dataset. As for CRPS metric, we generate 10 different reconstructions of 128 samples from validation set. We follow this recipe for all the baselines and our VFM experiments, except for Latent DAPS, where, due to the slower generation we only generated 128 samples instead of 1024 (all the rest of the settings are followed as stated above).

Our results show that while baselines may achieve high PSNR/SSIM due to mean-seeking behavior, VFM significantly outperforms them on distributional metrics (FID, MMD, CRPS), which indicates superior perceptual quality and a more accurate approximation of the complex posterior. We also observe that generating multiple samples through VFM in 1-step and then taking the average smoothes the reconstructions, which achieves competitive or better PSNR/SSIM values as well.

### B.2.4. INVERSE PROBLEMS AND EVALUATION SETUP

We evaluate VFM and all baselines on a diverse set of standard linear and nonlinear inverse problems frequently used in the literature. Our VFM model was trained jointly to handle denoising, random inpainting, box inpainting, super-resolution, Gaussian deblurring, motion deblurring and nonlinear high dynamic range (HDR) imaging via the amortized conditioning mechanism described in Section 3.2.

For quantitative evaluation, we focus on the structurally challenging tasks (inpainting, super-resolution, and deblurring) and omit pure denoising. To ensure a rigorous and fair comparison, all baselines utilize the exact same pre-trained SiT-B/2 backbone that was used to initialize VFM. This strictly isolates the performance differences to the sampling method (iterative guidance vs. one-step VFM) rather than the generative prior quality. Consequently, the reported numbers for VFM can serve as a reliable reference for future benchmarking on the SiT-B/2 architecture. We also followed the best practices from

| Task | Method | NFE | PSNR (↑) | SSIM (↑) | LPIPS (↓) | FID (↓) | MMD (↓) | CRPS$_{\text{DINO}}$ (↓) | CRPS$_{\text{Inc}}$ (↓) | Time (s) (↓) |
|---|---|---|---|---|---|---|---|---|---|---|
| Inpaint (random) | Latent DPS | 250×2 | 26.01 | 0.721 | 0.337 | 55.81 | 0.113 | 0.472 | 0.363 | 7.2164 |
| | Latent DAPS | 250×2 | 25.09 | 0.671 | 0.384 | – | – | 0.474 | **0.356** | 44.347 |
| | PSLD | 250×2 | 25.63 | 0.713 | 0.338 | 56.13 | 0.123 | 0.462 | 0.386 | 10.286 |
| | MPGD | 250×2 | **26.03** | 0.720 | 0.339 | 55.82 | 0.112 | 0.470 | 0.363 | 7.3512 |
| | FlowChef | 250×2 | 26.01 | 0.720 | 0.338 | 55.73 | 0.111 | 0.471 | 0.364 | 7.3885 |
| | FlowDPS | 250×2 | 25.80 | **0.729** | 0.344 | 62.62 | 0.139 | 0.557 | 0.453 | 14.054 |
| | LFlow | 250×2 | 25.21 | 0.694 | 0.383 | 71.21 | 0.159 | 0.525 | 0.430 | 6.9216 |
| | `frozen-`$\theta$ | 1 | 21.07 | 0.534 | 0.530 | 126.45 | 0.236 | 0.787 | 0.580 | **0.015** |
| | VFM (ours) | 1 / 10 | 21.74 / 24.58 | 0.556 / 0.674 | 0.369 / **0.335** | **40.20** | **0.036** | **0.439** | 0.313 | 0.025 / 0.252 |
| Super-res. (×4) | Latent DPS | 250×2 | 23.91 | 0.641 | 0.388 | 68.73 | 0.154 | 0.554 | 0.447 | 7.4195 |
| | Latent DAPS | 250×2 | 21.73 | 0.511 | 0.473 | – | – | 0.575 | 0.400 | 44.424 |
| | PSLD | 250×2 | 23.92 | 0.639 | 0.401 | 74.59 | 0.169 | 0.565 | 0.453 | 10.375 |
| | MPGD | 250×2 | 23.93 | 0.642 | 0.388 | 69.01 | 0.157 | 0.553 | 0.446 | 7.3801 |
| | FlowChef | 250×2 | 23.91 | 0.641 | 0.388 | 68.63 | 0.154 | 0.553 | 0.447 | 7.4914 |
| | FlowDPS | 250×2 | 24.13 | 0.655 | 0.413 | 81.47 | 0.193 | 0.633 | 0.547 | 14.303 |
| | LFlow | 250×2 | 23.07 | 0.581 | 0.439 | 69.57 | 0.136 | 0.589 | 0.418 | 6.8775 |
| | `frozen-`$\theta$ | 1 | 20.61 | 0.469 | 0.557 | 148.50 | 0.270 | 0.837 | 0.637 | **0.015** |
| | VFM (ours) | 1 / 10 | 21.74 / **24.37** | 0.559 / **0.673** | 0.354 / **0.325** | **39.08** | **0.032** | 0.475 | **0.320** | **0.015** / 0.148 |
| Motion deblur | Latent DPS | 250×2 | 22.17 | 0.555 | 0.478 | 103.35 | 0.203 | 0.716 | 0.519 | 7.5214 |
| | Latent DAPS | 250×2 | 21.26 | 0.499 | 0.480 | – | – | **0.558** | 0.392 | 46.691 |
| | PSLD | 250×2 | 21.62 | 0.537 | 0.516 | 136.63 | 0.260 | 0.819 | 0.588 | 10.129 |
| | MPGD | 250×2 | 22.20 | 0.557 | 0.478 | 102.97 | 0.203 | 0.715 | 0.519 | 7.5031 |
| | FlowChef | 250×2 | 22.18 | 0.556 | 0.477 | 103.35 | 0.203 | 0.715 | 0.519 | 7.4681 |
| | FlowDPS | 250×2 | **22.31** | **0.579** | 0.498 | 122.09 | 0.240 | 0.804 | 0.597 | 14.715 |
| | LFlow | 250×2 | 21.10 | 0.535 | 0.530 | 145.26 | 0.272 | 0.852 | 0.593 | 6.9493 |
| | `frozen-`$\theta$ | 1 | 18.30 | 0.348 | 0.651 | 214.29 | 0.365 | 1.099 | 0.720 | **0.015** |
| | VFM (ours) | 1 / 10 | 18.51 / 20.28 | 0.413 / 0.507 | 0.470 / **0.469** | **54.99** | **0.073** | 0.608 | **0.385** | **0.015** / 0.148 |
| Nonlinear HDR | Latent DPS | 250×2 | 21.40 | 0.624 | 0.421 | 69.95 | 0.142 | 0.554 | 0.420 | 7.1892 |
| | Latent DAPS | 250×2 | **23.46** | **0.693** | 0.364 | – | – | 0.511 | 0.378 | 43.322 |
| | VFM (ours) | 1 / 10 | 21.48 / 22.99 | 0.561 / 0.654 | 0.365 / **0.347** | **41.78** | **0.041** | **0.510** | **0.352** | **0.015** / 0.148 |

*Table 2.* Quantitative comparison on ImageNet for various inverse problems. Best results are in **bold**, second best are underlined. ↑: higher is better, ↓: lower is better.

SiT-B/2 unconditional sampling to get the best results.

The specific forward operators for the evaluated tasks are defined as follows. For random inpainting, we apply a random noise mask where the occlusion probability is sampled uniformly from the interval $(0.3, 0.7)$ for each image. In the case of box inpainting, we utilize a rectangular mask with a random location and aspect ratio, where the height and width are sampled independently from the interval $(32, 128)$. Super-resolution (x4) is implemented by downsampling the input image by a factor of 4 using bicubic interpolation. Finally, for the deblurring tasks, we employ a $61 \times 61$ kernel size, using a standard deviation of $\sigma = 3.0$ for Gaussian deblurring and an intensity value of $0.5$ for motion deblurring. Additionally, for all inverse problems, the measurements are further corrupted by additive Gaussian noise with a standard deviation of $\sigma = 0.05$.

### B.3. General Reward Alignment with VFM

In Section 4.3, we introduced Variational Flow Maps for general reward alignment. Given a base data distribution $p_{data}(x)$ and a differentiable reward model $R(x, c)$ conditioned on context $c$ (e.g., a class label or text prompt), the goal of reward alignment is to sample from the reward-tilted distribution:

$$p_{\text{reward}}(x|c) \propto p_{\text{data}}(x) \exp(\beta R(x, c)), \tag{105}$$

where $\beta > 0$ is a temperature parameter controlling the strength of the reward. In the context of a flow map $x = f_\theta(z)$, this target data distribution induces a corresponding target posterior in the latent noise space:

$$p(z|c) \propto p(z) \exp(\beta R(f_\theta(z), c)). \tag{106}$$

In the reward-alignment setting, there is no standard structural degradation $y = A(x) + \varepsilon$. However, reward maximization can still be naturally cast as an inverse problem. In this view, the context $c$ (e.g., a text prompt) takes the place of the

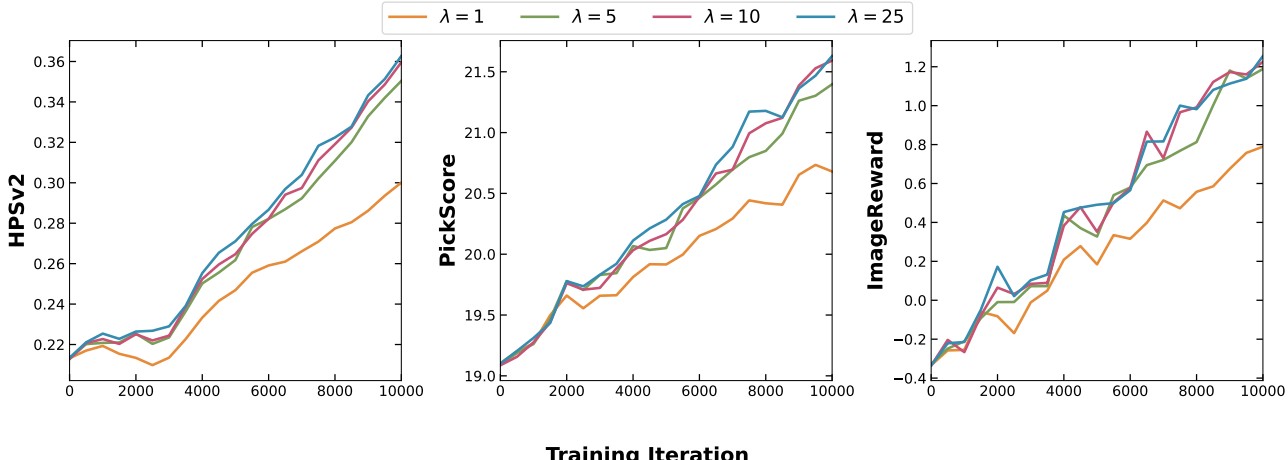

**Training Iteration**

*Figure 11.* Quantitative evaluation of generated samples over 10,000 training iterations for varying values of $\lambda$. We report HPSv2 (left), PickScore (middle), and ImageReward (right). Higher scores indicate better alignment with human preferences.

observation. Probabilistically, we treat the evaluated reward $R(\cdot, c)$ as the (unnormalized) log-likelihood of this context given the generated sample. Substituting the standard inverse problem observation loss $-\frac{1}{2\sigma^2}\|y - A(f_\theta(z))\|^2$ with this reward-based log-likelihood $\lambda R(f_\theta(z), c)$ naturally yields the objective:

$$\mathcal{L}(\theta, \phi) = -\lambda \, \mathbb{E}_{c \sim p(c), z \sim q_\phi(z|c)}[R(f_\theta(z), c)] + \mathcal{L}_{\text{KL}}(\phi) + \frac{1}{2\tau^2}\mathcal{L}_{\text{data}}(\theta; \phi). \tag{107}$$

**Motivation.** Crucially, this fine-tuning objective is a principled Variational Inference (VI) formulation derived directly from the Evidence Lower Bound (ELBO). At its global optimum, it recovers the true reward-tilted distribution $p_{\text{reward}}(x|c)$. The $-\lambda R(f_\theta(z), c)$ term corresponds to maximizing the expected log-likelihood of the ideal observation, pushing the adapter to find, and the flow map to decode, regions of high reward. The $\mathcal{L}_{\text{KL}}$ term matches the variational posterior to the prior $p(z)$, preventing the latent space from collapsing to a single deterministic point. Finally, the $\mathcal{L}_{\text{data}}$ term ensures that the generator $f_\theta$ remains a valid transport map anchored to the true data manifold. By minimizing this principled objective, the framework naturally prevents the generator from collapsing into an adversarial state purely to cheat the reward model.

**Fine-tuning setup.** We use the same adapter architecture as in the original VFM training, but instead of degraded observations, we map a fixed, learned spatial latent grid to $(\mu, \sigma)$ conditioned on the class label. In standard VFM training, an adaptive scaling function is applied to the entire loss to stabilize gradients. Since $-\lambda R(f_\theta(z), c)$ can take large negative values, applying this normalization to the full reward objective distorts gradient magnitudes unpredictably. We therefore apply adaptive normalization only to the flow-related terms during reward alignment tasks. Next, because we explicitly want to tilt the flow map toward high-reward regions, we pass gradients through the active trainable model during the reward loss calculation. This contrasts with standard VFM, which uses the EMA model to prevent the flow map from being updated by the observation loss. Following the reward alignment setup in Meta Flow Maps (MFM) (Potaptchik et al., 2026), we use HPSv2 (Wu et al., 2023) during the fine-tuning stage. The adapter is amortized over all $1,000$ classes of ImageNet, and the reward is calculated using a fixed prompt *"A high-resolution, high-quality photograph of a {class_name}"* as utilized in MFM. Finally, we initialize VFM with the large DMF-XL/2+ model (Lee et al., 2025) and fine-tune for $10,000$ iterations with a batch size of $64$ (corresponding to $\sim 0.5$ epochs and taking only $6$ hours). The rest of the VFM training follows the standard procedure and parameter choices described throughout the paper.

**Multi-step observation.** We observe that one-step samples achieve the highest reward scores under the fine-tuned model, whereas multi-step samples tend to regress toward the unconditional ImageNet distribution. We attribute this to two factors. First, the reward loss is evaluated directly on the one-step map $z \to f_\theta(z)$. Second, unlike standard VFM training, we evaluate the reward loss through the active flow map rather than the EMA model. This explicitly tilts the model's one-step predictions, while its intermediate vector fields remain largely anchored to the unconditional data distribution. This discrepancy pulls multi-step trajectories back toward the base ImageNet data manifold. A natural remedy, computing the

reward on a short $K$-step rollout (e.g., $K = 3$) during training, would propagate the reward signal into the intermediate velocity field and is left as future work.

**Evaluation.** In addition to training with the HPSv2 reward, we evaluate the reward scores of VFM outputs during training based on various alignment metrics, including HPSv2 (Wu et al., 2023), PickScore (Kirstain et al., 2023), and ImageReward (Xu et al., 2023). Figure 11 shows the reward score progression throughout training, calculated every 500 steps and averaged over 64 random generations. While we explore the effects of varying the reward strength $\lambda$, we find that a default value of $\lambda = 1$ achieves strong performance without the need for extensive hyperparameter tuning. As shown, VFM consistently boosts the reward across all alignment metrics.

### B.4. Additional Results

In this section, we present a comprehensive set of qualitative results on ImageNet $256 \times 256$ to further validate the effectiveness of Variational Flow Maps.

**Qualitative Comparisons.** Figures 12, 13, 14, 15, 16 provide side-by-side comparisons of VFM against seven state-of-the-art baselines across five distinct inverse problems. In all cases, VFM produces sharp, coherent, and consistent samples in a single forward pass, whereas baselines often exhibit artifacts or require hundreds of function evaluations to achieve comparable fidelity.

**Uncertainty Quantification.** A key advantage of VFM is its ability to learn a proper posterior distribution rather than collapsing to a single mode. In Figure 18, we visualize the pixel-wise mean and standard deviation computed from multiple posterior samples (10 samples). The uncertainty maps clearly highlight that VFM localizes variance in ambiguous regions (e.g., occluded areas or fine details lost to blur), which provides valuable information about the posterior that is typically infeasible to extract with slow or mode-collapsing baselines.

**Structured Noise ("Make Some Noise").** Figure 17 visualizes the internal operation of the noise adapter $q_\phi(z|y)$. We display the predicted latent mean $\mu$, the standard deviation $\sigma$, and the resulting reparameterized noise samples $z$. We observe strong structural patterns in the learned noise, which indicates that the adapter actively aligns the latent space to the data manifold. This validates our core premise: by "learning the proper noise" via optimization, we bridge the guidance gap without requiring iterative steering.

**Diversity and Mode Coverage.** In Figures 20 and 21, we examine diverse generation scenarios. While the baselines frequently fail or collapse to a single (often incorrect) solution, VFM successfully generates diverse, high-quality samples that are all consistent with the measurements. We observe that greater ill-posedness naturally leads to higher diversity in our generations, confirming that the model captures the multimodal nature of the posterior.

**Unconditional Generation.** Figure 19 presents additional curated unconditional samples generated by the trained flow map. It further highlights the generative quality of our backbone model.

**Reward Alignment.** Figure 22 provides additional uncurated samples generated by the fine-tuned flow map. VFM consistently samples from the target reward-tilted distribution. Furthermore, Figure 23 illustrates the evolution of generated images across different fine-tuning iterations using fixed latent seeds, which highlights the rapid and stable adaptation of the model.

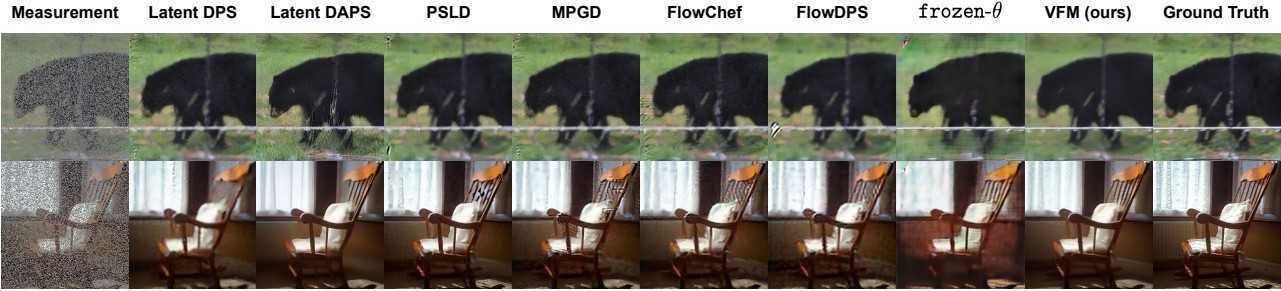

*Figure 12.* **Qualitative comparison on Random Inpainting.** We compare one-step VFM samples against seven baselines. VFM recovers fine details and texture consistent with the unmasked regions, while maintaining high perceptual quality.

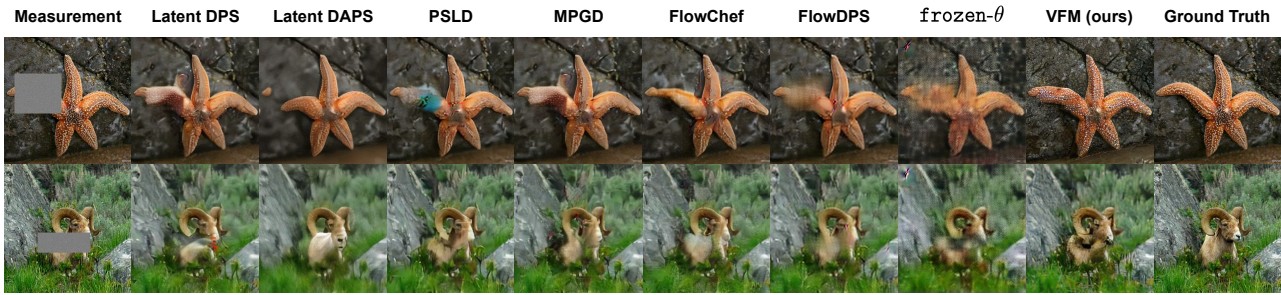

*Figure 13.* **Qualitative comparison on Box Inpainting.** Comparison of VFM against baselines for large occlusions. VFM generates plausible semantic content to fill the missing regions in a single step.

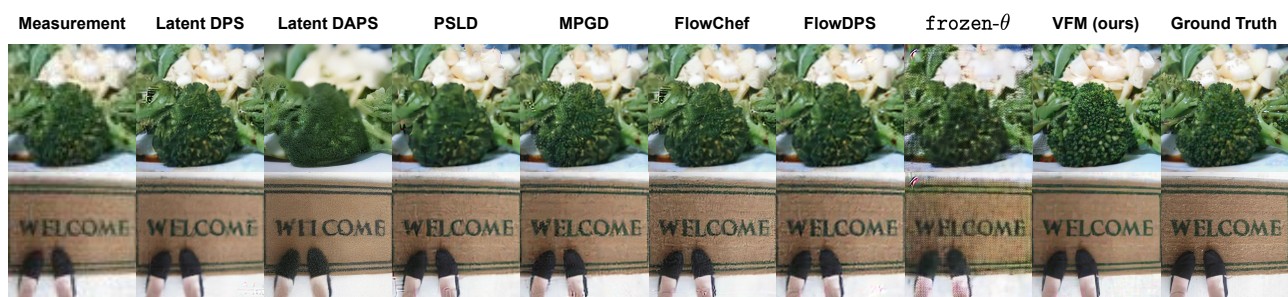

*Figure 14.* **Qualitative comparison on Super-Resolution** ($\times 4$). VFM effectively upsamples the low-resolution inputs, leading to sharp edges and realistic textures compared to the often over-smoothed baseline results.

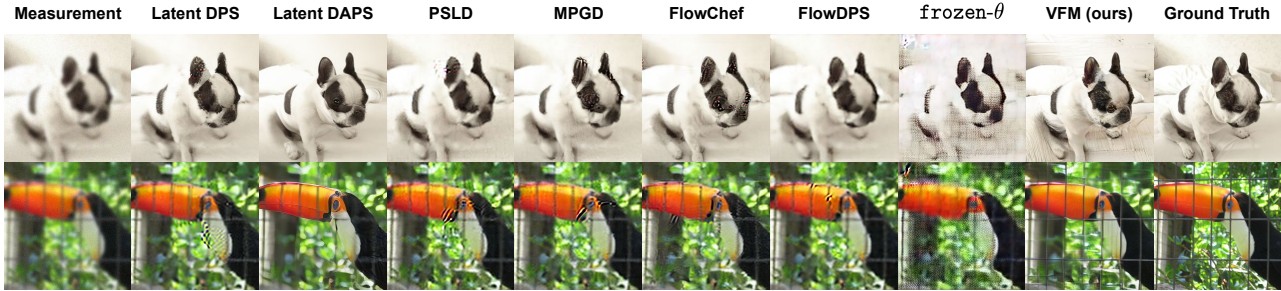

*Figure 15.* **Qualitative comparison on Gaussian Deblurring.** VFM successfully restores sharpness from heavily blurred observations ($\sigma = 3.0$), and it also avoids the artifacts common in guidance-based methods.

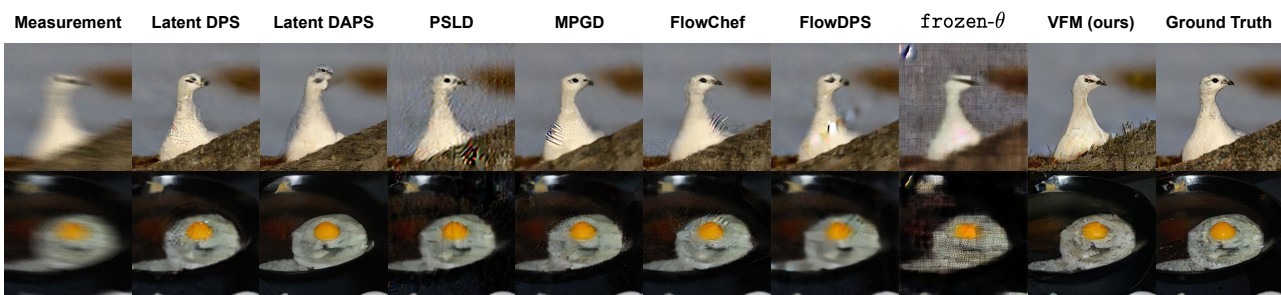

*Figure 16.* **Qualitative comparison on Motion Deblurring.** Comparison of deblurring performance on motion-blurred inputs. VFM resolves the motion streaks into coherent structures.

| Ground Truth | Measurement | mean | std | Latents 1 | Latents 2 | Latent 3 |
|---|---|---|---|---|---|---|

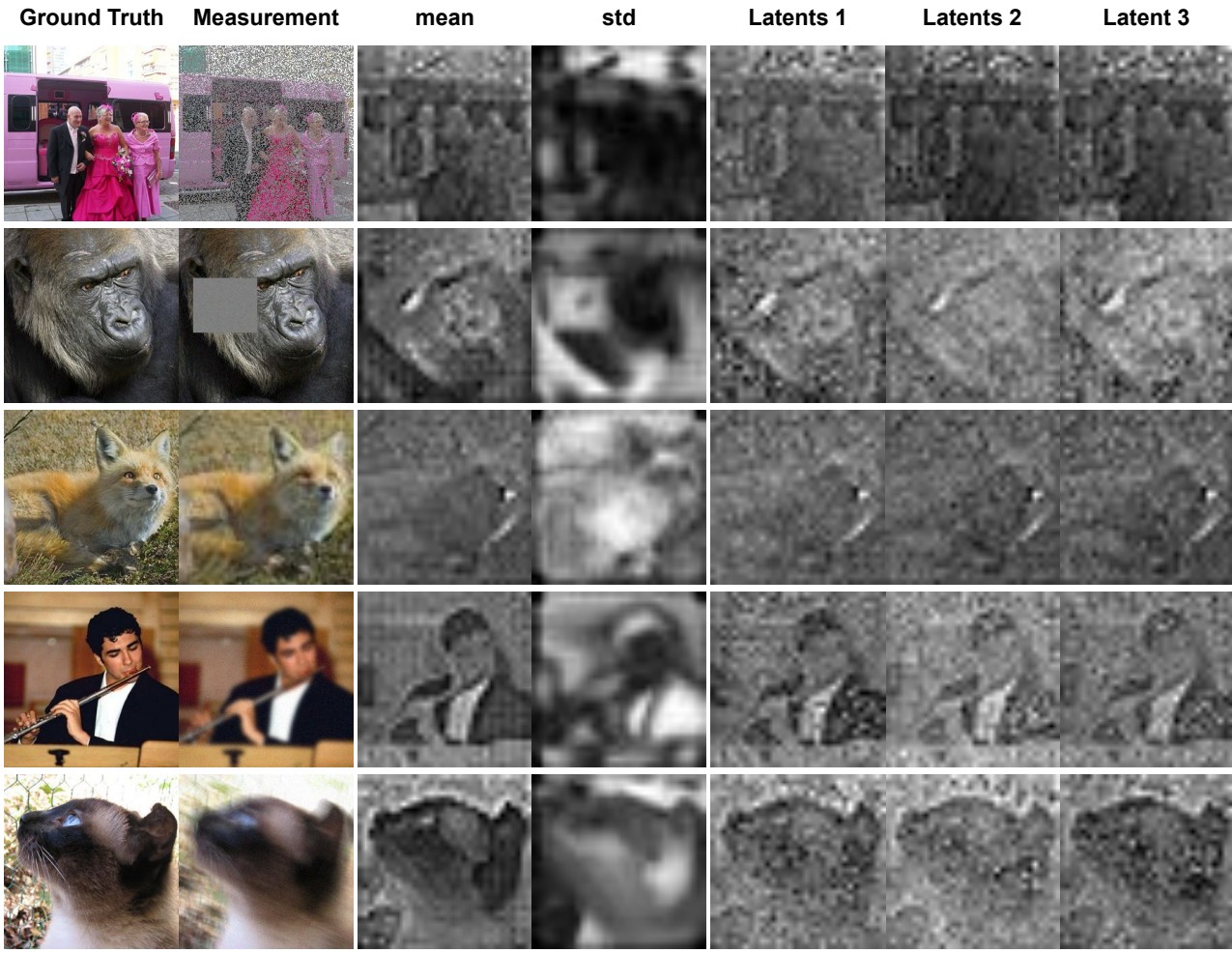

*Figure 17.* **Visualizing the Learned Noise Space.** We visualize the outputs of the noise adapter $q_\phi(z|y)$. From left to right: ground truth, measurement, the predicted latent mean $\mu$, standard deviation $\sigma$, and three independent latent samples drawn from the distribution. The visible structure in the "noise" confirms that the adapter optimizes the latent initialization to align with the conditional data manifold. From top to bottom, rows correspond to: random inpainting, box inpainting, super-resolution, gaussian deblurring, and motion deblurring.

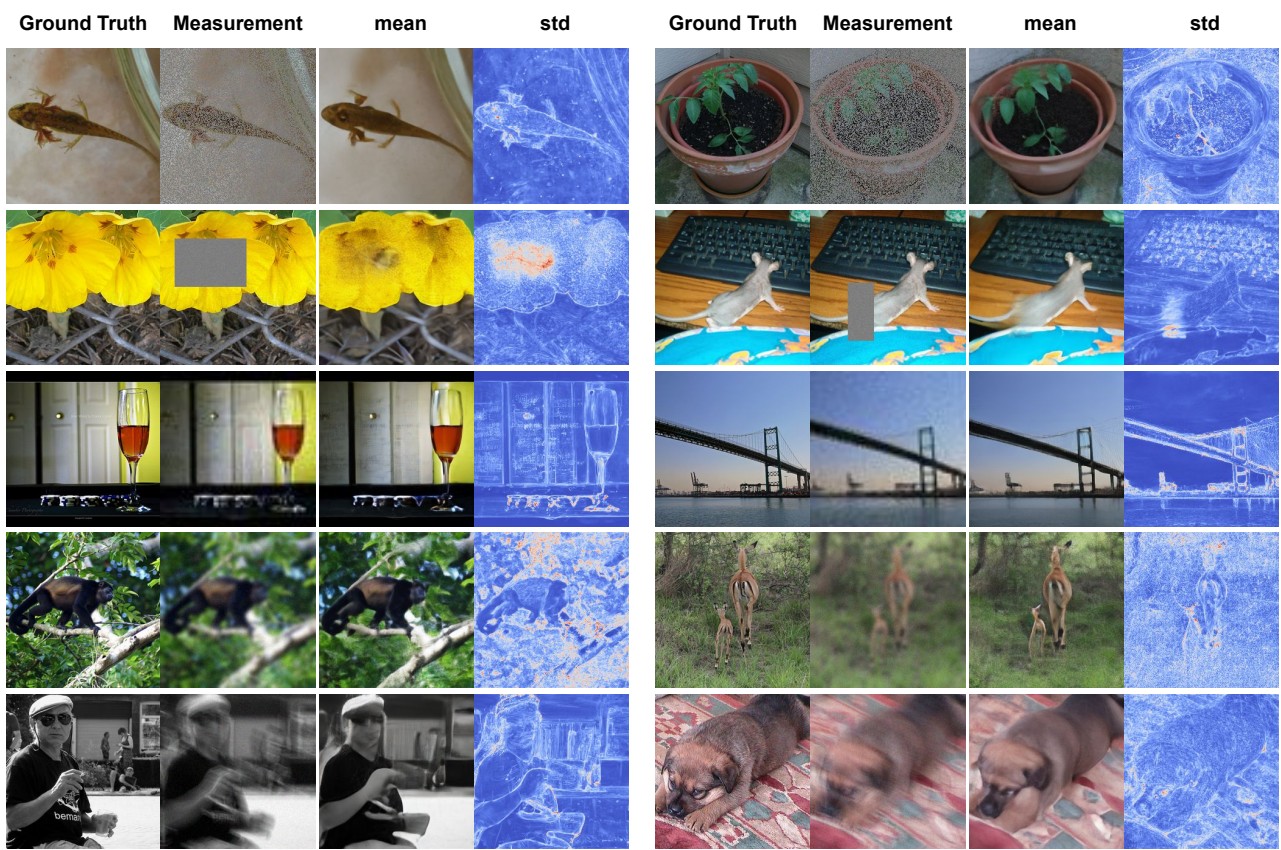

*Figure 18.* **Posterior Uncertainty Quantification.** We display the pixel-wise mean and standard deviation computed from 10 conditional samples generated by VFM. The standard deviation maps (right column) accurately capture the uncertainty inherent in the inverse problem, which highlights ambiguous regions where the model generates diverse solutions.

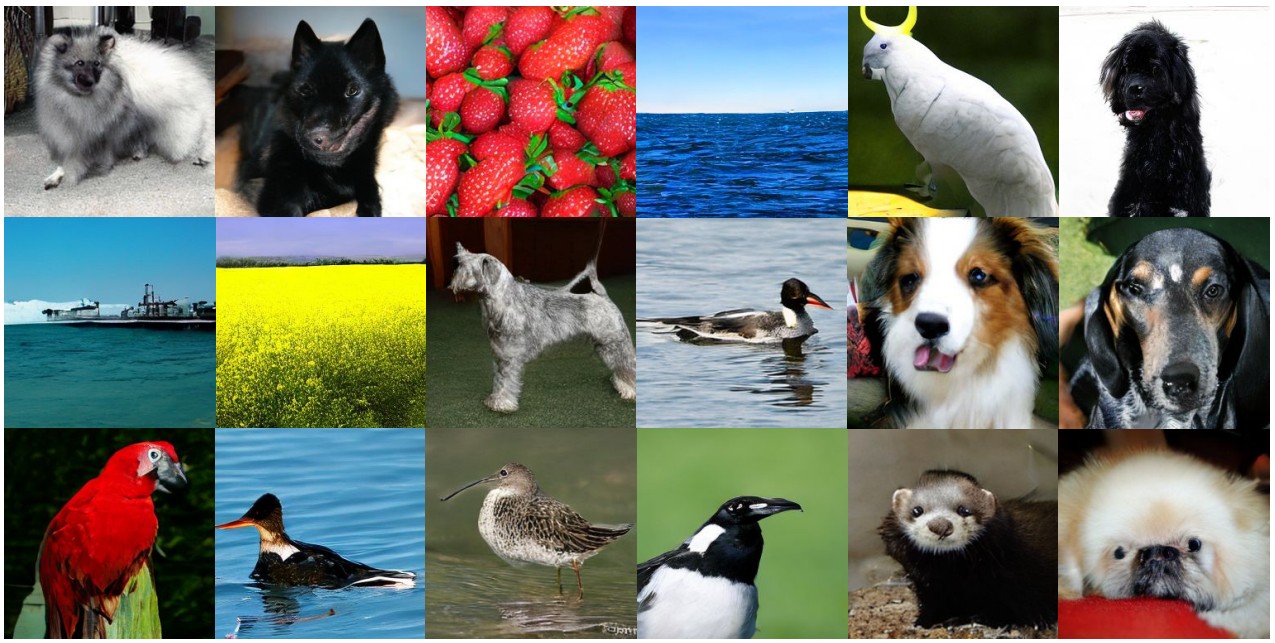

*Figure 19.* **Unconditional Samples.** Curated unconditional samples generated by the VFM.

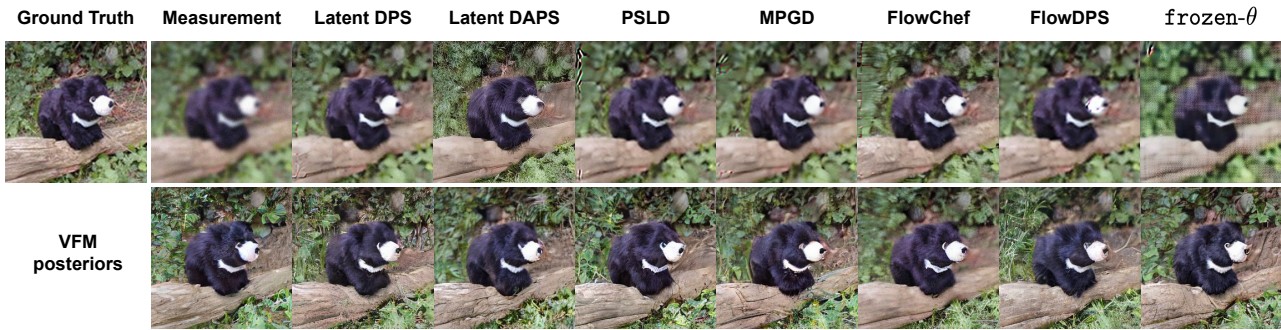

*Figure 20.* **Posterior Diversity (Sample Set 1).** Evaluation of sample diversity on gaussian deblurring. While baselines often collapse to a single mode or fail to produce valid results, VFM generates eight distinct, plausible, and measurement-consistent posterior samples.

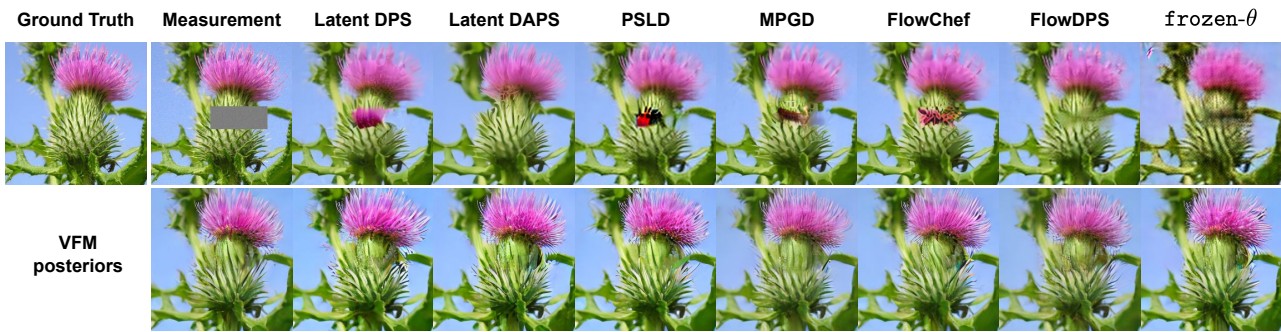

*Figure 21.* **Posterior Diversity (Sample Set 2).** Additional examples of diverse posterior sampling on box inpainting task. The high variance among the VFM samples reflects the multimodal nature of the posterior distribution for these ill-posed tasks.

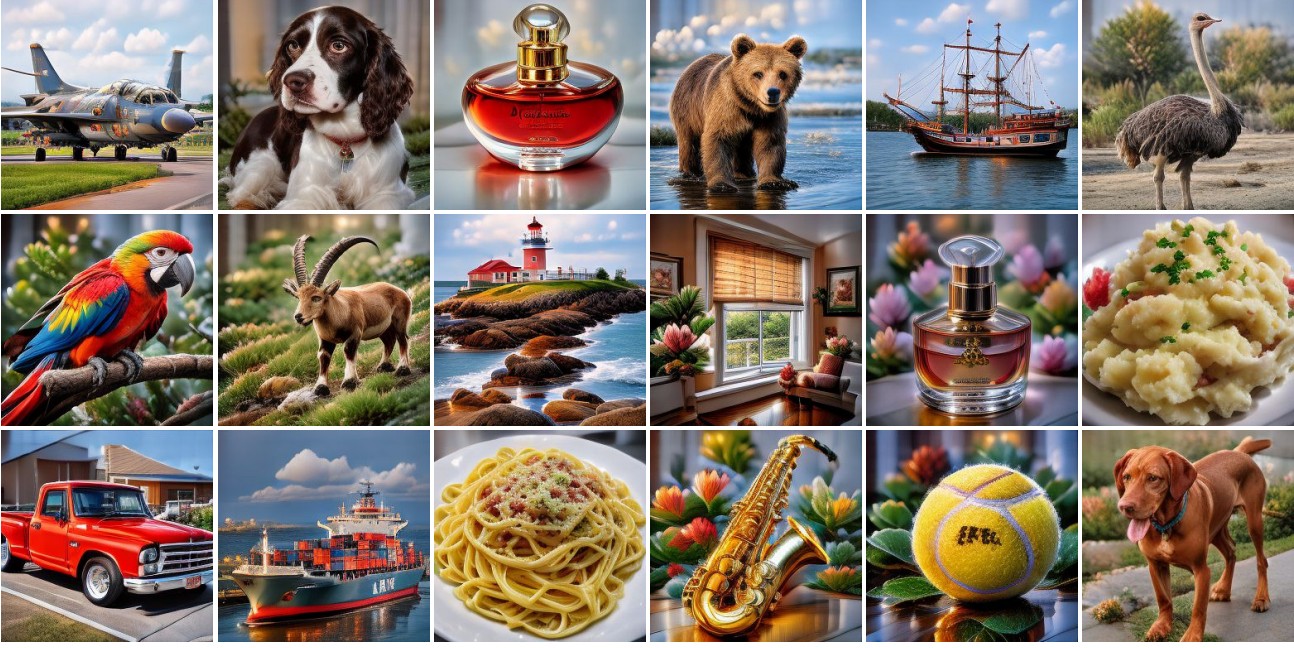

*Figure 22.* **Uncurated Samples from Reward Fine-Tuning** ($\lambda = 1$). Additional uncurated samples generated by the fine-tuned flow map. VFM consistently samples high-quality images from the target reward-tilted distribution, resulting in enhanced aesthetic and perceptual quality in a single forward pass.

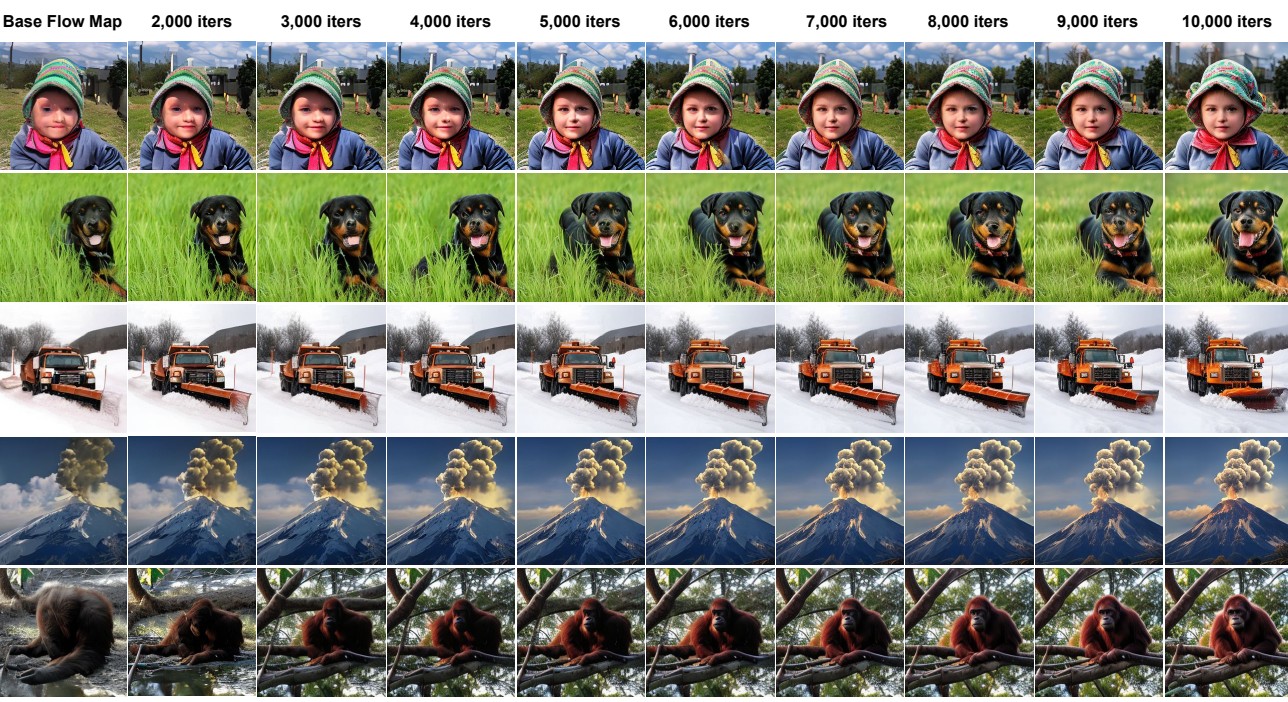

*Figure 23.* **Evolution of One-Step Generations during Reward Fine-Tuning.** We visualize the progress of generated samples across varying fine-tuning iterations using fixed latent seeds, all produced in a single neural function evaluation (1 NFE). These seeds are drawn from a pure standard Gaussian distribution. Although we tilt the original flow map to accommodate adapter-conditioned noises, the original noise space remains valid. As a result, this short fine-tuning process successfully enhances the one-step generative capabilities of the base DMF model even without the use of an adapter.

