# OpenReview forum: "Variational Flow Maps: Make Some Noise for One-Step Conditional Generation"
_ICML.cc/2026/Conference — ICML 2026 regular_

### Official Review · Reviewer_A5qF · 2026-02-24

**Soundness:** 3
**Presentation:** 2
**Significance:** 3
**Originality:** 4
**Overall Recommendation:** 5
**Confidence:** 4

**Summary:**

This paper introduces Variational Flow Maps (VFM), a framework for enabling conditional generation and posterior sampling using flow-map-based generative models in one or a few inference steps. Flow maps can generate high-quality samples in a single forward pass, but unlike diffusion models, they lack an explicit sampling trajectory, making it difficult to incorporate measurement constraints for conditional tasks such as inverse problems.

The key idea of the paper is to shift the conditioning perspective from guiding a sampling trajectory (as done in diffusion models) to instead learning an observation-dependent distribution over the initial latent noise. Specifically, given an observation y, the method learns a noise adapter qϕ(z∣y) that approximates the latent posterior p(z∣y). Samples from this learned noise distribution are then mapped deterministically to data space using a learned flow map x=fθ(z), producing approximate samples from p(x∣y) in one step.

To support this approach, the authors derive a principled variational objective inspired by VAE-style joint distribution matching. They match two factorizations of the joint distribution over data, observations, and latent variables, which leads to a training objective combining:
- A mean-flow structural loss to preserve flow-map consistency
- An observation consistency term enforcing measurement fidelity,
- A KL regularizer on the latent noise adapter.

A central contribution is the joint training of the flow map and the noise adapter. The authors argue theoretically (under linear-Gaussian assumptions) and demonstrate empirically that jointly adapting the generator reshapes the noise-to-data coupling, allowing a simple Gaussian variational posterior to approximate complex data posteriors more effectively.

Empirically, the method is validated on:
- A controlled 2D checkerboard inverse problem, illustrating improved posterior mode coverage and manifold preservation,
- Image inverse problems on ImageNet (e.g., box inpainting and Gaussian deblurring), where VFM achieves competitive or superior distributional metrics (FID, MMD, CRPS) compared to iterative guidance-based baselines,
- Unconditional generation, where the trained flow map remains competitive with strong one-step flow-map baselines.

Overall, the paper proposes a variational latent-space reformulation of conditional flow-map generation that enables fast one-step posterior sampling while maintaining strong empirical performance.

**Compliance With Llm Reviewing Policy:**

Affirmed.

**Final Justification:**

I appreciate the authors good work and response to my questions.

**Key Questions For Authors:**

1. How does VFM compare to strong 2-4 step distilled or consistency-based diffusion posterior samplers, rather than 250-step guidance methods?

2. Beyond CRPS, have you evaluated calibration using coverage tests or uncertainty diagnostics to verify that samples reflect correct posterior uncertainty?

3. How sensitive is performance to the diagonal Gaussian assumption for q(z∣y)? Would a more expressive posterior significantly improve results?

4.How robust is the method to choices of τ and α? Is performance stable across reasonable ranges?

5. Does joint conditional training degrade the learned unconditional distribution compared to a purely unconditional flow-map model trained for the same budget?

**Limitations:**

The paper includes a short impact statement focused primarily on computational efficiency and energy reduction, which is appropriate but limited in scope. It does not meaningfully discuss broader limitations or potential negative societal impacts.

Suggestions for improvement:

Include a clearer discussion of limitations, such as:
- The Gaussian posterior assumption,
- Sensitivity to hyperparameters (τ, α),
- Lack of strong guarantees on posterior correctness in high dimensions,
- Potential trade-offs between conditional and unconditional performance.
- Bias propagation from pretrained generative backbones (e.g., ImageNet/latent diffusion priors).

Providing a more comprehensive limitations and impact discussion would improve transparency and strengthen the paper.

**Strengths And Weaknesses:**

---
**Presentation**

the paper is logically structured:
- Motivation: the “guidance gap”
- Background: flow maps, inverse problems, variational inference
- Core method: joint latent inference
- Empirical validation

The shift from trajectory guidance to noise adaptation is communicated clearly and repeatedly.

Strong conceptual framing: the “guidance gap” terminology provides a helpful organizing idea. The VAE-style joint factorization perspective is well motivated and ties the method into familiar probabilistic modeling concepts.

Effective use of toy experiment.
The checkerboard experiment visually and quantitatively illustrates the key mechanism before scaling to ImageNet.



Section 3 derivation may feel dense.
The joint distribution matching and KL derivation, while correct, may be difficult to parse without strong VAE background. A short conceptual summary paragraph before the formal derivation could improve accessibility.

---

**Significance**

Conditional generation is central to inverse problems, restoration, and controllable synthesis. Flow maps are promising for one-step generation but lack conditioning mechanisms. This paper directly addresses that gap.

computational impact: reducing conditional sampling from hundreds of steps to one (or a few) has substantial practical implications for real-time and energy-efficient applications.

The method connects:
- Flow-map generative modeling
- Variational inference
- Conditional inverse problem solving

This conceptual bridge may inspire further work in latent-space posterior learning for deterministic generators.

One critique in significance is that the scope is somewhat specialized. The main benefit is for flow-map-based generative models. While this is an active area, it is narrower than general diffusion modeling.


---


**Originality**

The key originality lies in reframing conditional generation for deterministic one-step generators as amortized latent posterior inference, instead of trajectory guidance.

Creative combination of ideas. The method combines:
- Flow maps (mean flows)
- VAE-style joint matching
- Latent posterior approximation
- Structural ODE constraints

The integration is non-trivial and thoughtfully justified.

The theoretical and empirical argument that the generator must adapt to make posterior inference simple is an insightful contribution.

---

**Soundness**

- The core idea (reformulating conditional sampling for flow maps as variational inference in latent (noise) space) is conceptually clean and mathematically grounded. The VAE-style joint distribution matching in Section 3.1 is coherent and leads to a principled objective rather than an ad-hoc modification. The derivation of the loss from a KL divergence between two joint factorizations is standard and sound.

- Proposition 3.1 provides a clear theoretical justification (under linear-Gaussian assumptions) for jointly training the flow map and the noise adapter. While the assumptions are simplified, the result directly supports the central claim: freezing the generator can bias posterior inference, whereas joint optimization can recover the correct posterior mean.

- Replacing the explicit reconstruction term with the mean flow loss is well-justified through the upper-bound argument (Proposition 3.2). This maintains compatibility with the flow-map structure instead of naively treating the model as a generic decoder.

- The 2D checkerboard experiment is thoughtfully constructed and isolates the core mechanism.

- On ImageNet inverse problems, the authors report multiple metrics (FID, MMD, CRPS, LPIPS, PSNR, SSIM) and compare against several guidance-based baselines. The substantial inference speedup is clearly demonstrated.

*Weaknesses*

- Limited theoretical guarantees beyond simplified setting.
The key theoretical result holds under linear-Gaussian assumptions. While useful for intuition, it does not guarantee posterior correctness in realistic nonlinear high-dimensional settings. The paper does not provide analysis of bias or consistency beyond that simplified case.

- Calibration analysis is relatively shallow.
Although CRPS is reported, deeper uncertainty diagnostics (e.g., coverage tests, conditional variance analysis, credible-region calibration) are not explored. Since the method is explicitly positioned as posterior sampling, a more thorough uncertainty evaluation would strengthen soundness.

- Comparative baselines focus on iterative guidance.
The comparison is primarily against 250-step guidance-based samplers. Fewer comparisons are made against aggressively optimized few-step diffusion or distilled samplers tailored for fast conditional generation. While this does not invalidate the results, it leaves some ambiguity about the absolute competitiveness in low-step regimes.

---

**Overall**

The paper is technically sound, conceptually well-motivated, and empirically validated. Its main contribution is a principled variational reformulation that enables one-step conditional generation with flow maps, supported by theoretical insight and convincing experiments. While deeper posterior calibration analysis and stronger few-step baselines would further strengthen the empirical section, the work represents a meaningful and creative advancement in conditional generative modeling with fast samplers.

---

> ### Author Rebuttal · Authors · 2026-03-30
>
> We thank the reviewer for the thorough and constructive review. We are grateful that the reviewer found our approach "well-motivated" with "non-trivial and thoughtfully justified" synthesis.
>
>
> > **[W1] Specialized scope**
>
> We respectfully disagree that the scope is limited. We view VFM not as a method developed _for flow maps_, but as a method _using flow maps_ to address a broader gap: principled one/few-step conditional sampling with deterministic generators. The core idea, shifting from data-space guidance to noise-space optimization, applies to any deterministic generator (GANs, consistency models, etc.). We chose flow maps for their strong empirical foundation and clean theory (Proposition 3.2). As noted in Remark 3.3, VFM can also be instantiated with consistency model losses.
>
> Moreover, our new reward alignment experiments (**[LINK BELOW]**) demonstrate that VFM extends well beyond inverse problems to sampling from reward-tilted distributions, a task relevant across generative modeling, achieved in only ~6 hours of fine-tuning.
>
>
> > **[W2] Theoretical assumption in Proposition 3.1**
>
> We clarify that the linear-Gaussian assumption is used only in Proposition 3.1, not in Propositions 3.2 or 3.4. The role of Proposition 3.1 is to serve as a rigorous counterexample: it demonstrates that separate training fails to recover the posterior mean even in the simplest setting, while joint training provably succeeds. For general cases, we added Remark A.19 discussing how joint training extends beyond the linear setting via Jacobian alignment, allowing the generator to adapt its local geometry with the factorized variational posterior. Extending the formal analysis to the full nonlinear regime is an ambitious direction for future work.
>
>
> > **[W3] [Q2] Calibration analysis beyond CRPS**
>
> Our evaluation already includes several calibration diagnostics beyond CRPS: NLPD (a proper scoring rule, Appendix B.1.3), SACC (support accuracy- a coverage metric on the checkerboard), posterior MMD (kernel-based distributional distance), CRPS in dual embedding spaces (Inception and DINO), and uncertainty maps (Figure 16) showing pixelwise standard deviation from 10 posterior samples. We have additionally computed LPIPS variance (see response to Reviewer 5dS2 and **[LINK BELOW]**).
>
>
> > **[W4] [Q1] 2-4 step distilled posterior samplers**
>
> To our knowledge, no established few-step distilled posterior samplers are directly comparable. Existing methods either (a) handle only specific conditioning (e.g., class-conditional) or (b) require task-specific tuning and retraining per observation y. VFM naturally amortizes across multiple inverse problems in a single training. We chose 250-step guidance baselines as the strongest existing approach and also added the recent LFlow [1] baseline (**[LINK BELOW]**).
>
>
> > **[Q3] Gaussian assumption for adapter**
>
> Using more flexible adapters on a frozen generator has been explored [2], but their training is significantly more complex and hard to scale. Our complementary approach keeps the adapter simple and compensates via joint training. Proposition 3.1 shows that joint training enables a diagonal Gaussian to recover the exact posterior mean, whereas separate training almost surely fails. Empirically, this simple adapter achieves best-in-class FID/MMD/CRPS on ImageNet 256x256, confirming that joint training reshapes the coupling to make posteriors representable by simple distributions.
>
>
> > **[Q4] Robustness of $\tau$ and $\alpha$**
>
> The method is quite robust. With EMA, performance is very stable for $\tau \geq 1$, the blue curves in Figures 5 and 6 are flat across more than two orders of magnitude. $\alpha$ controls the trade-off between unconditional and conditional quality. Higher $\alpha$ improves posterior fit at the cost of unconditional generation, and vice versa (see Figure 9). Our improved pipeline for images uses $\alpha=0.8$ with a two-phase schedule that achieves strong performance on both fronts.
>
>
> > **[Q5] Unconditional performance**
>
> There is a mild trade-off (Section 4.2 & Figure 4). VFM-B/2 achieves FID 10.77 (1-step) / 9.22 (2-step) vs. 5.63 for DMF-B/2 trained purely unconditionally, still competitive with other one-step baselines (iCT: 34.24, Shortcut: 40.30) despite fewer training epochs (180 vs. 240). Our improved two-phase pipeline (30 epochs unconditional, then 20 epochs with $\alpha=0.8$) accepts a negligible additional FID cost on unconditional generation in exchange for substantially stronger conditional performance across all benchmarks. See **[LINK BELOW]** for results.
>
>
> > **[L1] Comprehensive limitation discussions**
>
> We will expand the limitations section in the camera-ready to discuss all the points.
>
>
>
>
> **NEW RESULTS:** [LINK HERE](https://drive.google.com/file/d/1IgYO3D9BPpFTF1Y2CNsq-vDyHuI4kLOG/view?usp=sharing)
>
> [1] https://arxiv.org/abs/2511.06138
>
> [2] https://arxiv.org/abs/2502.06999

---

> > ### Author Rebuttal · Reviewer_A5qF · 2026-04-01
> >
> > Thank you.

---

### Official Review · Reviewer_5dS2 · 2026-02-28

**Soundness:** 3
**Presentation:** 3
**Significance:** 2
**Originality:** 3
**Overall Recommendation:** 4
**Confidence:** 4

**Summary:**

The authors propose Variational Flow Maps (VFM), a framework designed to improve one-step conditional generation by addressing the over-smoothing and blurriness issues inherent in deterministic distillation methods. By formulating a variational inference objective, the method augments a deterministic flow map with a parameterized noise injection mechanism to capture the residual stochasticity of the target distribution. The approach is evaluated on tasks including image super-resolution, inpainting, and text-to-image distillation, demonstrating improvements in overall generation quality.

**Compliance With Llm Reviewing Policy:**

Affirmed.

**Final Justification:**

I recommend this paper to be accepted. The authors has addressed my concerns and other reviewer's questions, It's an interesting one and a good paper.

**Key Questions For Authors:**

- Can you clarify the fundamental conceptual differences between the proposed noise injection mechanism and a standard Variational Autoencoder applied directly to the residual output space?
- Have you evaluated the diversity of the generated samples quantitatively (e.g., using LPIPS variance for the same conditioning input) to demonstrate that the injected noise models meaningful structural stochasticity rather than merely adding high-frequency artifacts?

**Limitations:**

Yes

**Strengths And Weaknesses:**

A key strength of the submission lies in its sound theoretical formulation, which casts the residual stochasticity as a variational inference problem, alongside its empirical validation across multiple conditional generation tasks. The presentation is generally clear, with logical transitions from the limitations of deterministic flow matching to the proposed variational objective. But there are notable weaknesses regarding its originality and practical significance. The proposed noise injection mechanism conceptually resembles standard conditional VAE reparameterization applied to the output space, which somewhat diminishes the architectural novelty. Furthermore, while the paper demonstrates performance improvements, the comparisons against recent state-of-the-art one-step adversarial distillation methods lack sufficient depth, leaving it unclear whether the visual gains fully justify the complexity of training an additional parameterized noise network. Finally, the authors do not thoroughly investigate the computational overhead introduced by the auxiliary noise network during the inference phase, which is a critical metric for any one-step generation method aiming for strict deployment efficiency.

---

> ### Author Rebuttal · Authors · 2026-03-30
>
> We thank the reviewer for recognizing the "sound theoretical formulation" and clear presentation. We clarify some aspects that may not have been fully conveyed and address specific points.
>
>
> > **[W1] [Q1] Differences to VAE**
>
> We acknowledge the VAE resemblance. However, we want to emphasize that VFM is not a straightforward application of conditional VAE to flow maps. Our contribution is to recast inverse problems as posterior inference over the latent noise, a formulation that naturally invokes VAE-like machinery but introduces several crucial differences:
>
> **Three-variable joint matching.** VFM matches two factorizations of the three-variable joint p(x, y, z) (Eq. 10) instead of p(x, z) as in VAEs, simultaneously aligning data x, observations y, and latent noise z. This yields the data-fidelity term $L_{data}$ (Eq. 13) that is absent in standard VAEs. This couples the adapter and flow map more tightly.
>
> **Structural constraints on the decoder via mean flow loss.** In a VAE, the decoder is a generic neural network trained with a reconstruction loss. In VFM, the decoder is a flow map that must satisfy the Eulerian condition of ODE flows. We enforce this by replacing the standard reconstruction loss with the upper-bound mean flow loss $L_{MF}$ (Prop. 3.2). Without it, joint training degenerates to the unconstrained failure mode (Figure 2b), where the flow map produces many off-manifold samples.
>
> **Joint training.** A key insight of VFM, formalized in Prop. 3.1, is that joint training allows flow map to reshape its noise-to-data coupling so that even a simple diagonal Gaussian adapter can approximate complex multimodal posteriors. Figure 2 visually demonstrates that the frozen baseline (standard VAE-like approach) fails to capture bimodality, while VFM with joint training succeeds.
>
> **Noise space perspective.** VFM doesn’t inject noise into the output. Instead, we learn a distribution over the initial noise z in the noise space of the generative model, and then map z → x = f(z). The posterior in noise space may be much simpler than in data space, making it amenable to a simple Gaussian approximation.
>
> > **[W2] Comparisons against adversarial distillation methods**
>
> We believe the reviewer may be referring to methods such as Adversarial Diffusion Distillation [1]. These target a different problem: distilling a pre-trained diffusion teacher into a fast student generator using GAN-style adversarial training, designed for unconditional or text-conditional generation. They are not designed for Bayesian posterior sampling from inverse problems, and adapting them to produce calibrated posterior samples (correctly capturing the full distribution p(x|y), not just generating a single plausible output) would require substantial, non-obvious modifications. We therefore believe a direct comparison would not be meaningful for the posterior sampling problem we address.
>
> That said, we have added the LFlow baseline, a recent guidance-based method for flow-based inverse problem solving, to our comparisons. VFM substantially outperforms it across metrics (see **[LINK BELOW]**).
>
> > **[W3] Computational overhead during the inference phase**
>
> The noise adapter adds negligible overhead. The adapter has 10M parameters vs. 130M for the flow map backbone (<8%). All wall-clock times in Tables 1-2 already include the adapter inference. We have also evaluated the separate times, and adapter takes 0.006s, flow map takes 0.008s, total ~0.015s per sample. This is instantaneous compared to guidance methods (7-47s per sample). Once the adapter produces $\mu$ and $\sigma$, generating multiple posterior samples only requires reparameterization $(z = \mu + \sigma \cdot \epsilon)$ plus one flow map forward pass per sample, no re-running of the adapter
>
> > **[Q2] Diversity of generated samples (LPIPS variance)**
>
> Thank you for this suggestion. We computed the LPIPS standard deviations across 10 independent sampling runs for VFM and all baselines. VFM exhibits meaningful diversity, the LPIPS variance is sufficiently large to confirm genuine structural variation (not just high-frequency artifacts), while concentrated enough to indicate measurement consistency (see **[LINK BELOW]**).
>
> We also wish to highlight that our paper already reports several distributional metrics that directly assess sample diversity and calibration: (i) FID and MMD measure distributional distance, where VFM achieves the best scores; (ii) CRPS (computed in both Inception and DINO embedding spaces) is a proper scoring rule that jointly evaluates calibration and sharpness, it rewards diversity while penalizing samples that deviate from the truth; (iii) the uncertainty maps in Figure 16 show pixelwise standard deviation from 10 posterior samples, confirming that VFM correctly localizes uncertainty in ambiguous regions.
>
>
>
>
> **NEW RESULTS:** [LINK HERE](https://drive.google.com/file/d/1IgYO3D9BPpFTF1Y2CNsq-vDyHuI4kLOG/view?usp=sharing)
>
> [1] https://arxiv.org/abs/2311.17042

---

> > ### Author Rebuttal · Reviewer_5dS2 · 2026-04-01
> >
> > Thanks for the author's response, I will maintain my score.

---

> > > ### Author Response · Authors · 2026-04-02
> > >
> > > Dear Reviewer 5dS2,
> > >
> > > We are very glad to hear that our responses addressed your original concerns. Since the start of the discussion period, we have continued to strengthen the paper with new results that we believe significantly broaden its impact.
> > >
> > > Specifically, we have introduced **the first principled framework for one-step reward-aligned sampling**, which achieves alignment in just ~0.5 epochs, and expanded our testing to **HDR imaging** to show how the model handles non-linear inverse problems. With the addition of the **LFlow baseline** and a new diversity analysis, VFM shows strong improvements across metrics, all while **halving the training cost**.
> > >
> > > Considering these additions, we kindly ask whether you might consider revisiting your score to reflect the expanded scope of the work. If any concerns remain, we would be happy to address them with further experiments or clarifications.
> > >
> > >
> > > Best regards,
> > >
> > > The Authors

---

### Official Review · Reviewer_Bfde · 2026-03-09

**Soundness:** 1
**Presentation:** 2
**Significance:** 2
**Originality:** 3
**Overall Recommendation:** 4
**Confidence:** 2

**Summary:**

The paper explores training an encoder to predict the noise given condition. The noise can be given to control an unconditional flow map model for conditional generation. The paper finds that it is better to train both the encoder and the flow map model jointly, and terms the method Variational Flow Maps (VFM).

**Compliance With Llm Reviewing Policy:**

Affirmed.

**Final Justification:**

The reviewer resolved some of my questions. I am raising the score to weak accept considering the work's merit and other reviewers' feedback.

**Key Questions For Authors:**

First, the introduction states that flow map models excel at unconditional tasks but not for conditional generation.

Line 50: "This “guidance gap” has limited flow maps to unconditional settings, leaving their potential for conditional generation largely unexplored."

This statement is not factually correct as one can simply train a conditional flow map model: f(x_s, s, t, c). The original consistency models and meanflow models are all trained for class-conditional ImageNet generation, proving that flow maps can be used for conditional generation.

A potential application is where an unconditional flow map model has to be frozen, in which case having an encoder to predict the conditional noise distribution would make sense. But in this paper, the proposed method requires joint training of both the flow map and the encoder. Then what is an example application of the proposed method, where simply training a conditional flow map model cannot solve?

If we do not enforce the hypothetical constraint where an unconditional flow map model must be frozen, the applications used in the paper (inpainting and debluring) can both be simply trained as conditional flow map models, and not using the proposed variational approach. The paper has mentioned amortizing multiple inverse problems in Sec 3.2, but this can also be trained as class-conditional flow maps which support multiple conditions.

**Limitations:**

The paper proposes a way to train jointly a flow map inverse encoder to turn an unconditional flow map into a conditional generator. However, as my question has previously stated, I am not seeing the practical applications for the proposed method besides academic novelty.

**Strengths And Weaknesses:**

I am not very convinced on the soundess of the method and hope the authors to help clarify. The confusion is stated in the questions section.

---

> ### Author Rebuttal · Authors · 2026-03-30
>
> We appreciate the reviewer’s questions and the opportunity to clarify the motivation and practical value of VFM. We hope the following detailed response addresses the core concerns.
>
>
> > **[Q1] Positioning against conditional flow maps**
>
> We fully agree that class-conditional flow maps exist and work well for label-based conditioning. However, the conditioning we address, namely, Bayesian inverse problems, is fundamentally different from class conditioning. We clarify our position along several axes:
>
> **Modularity and scalability via decoupling.** Training a single conditional flow map to handle all conditioning modalities simultaneously (class labels, text prompts, inpainting masks of varying shapes, blur kernels of varying widths, super-resolution factors, HDR tone mapping, etc...), would require entangling all these modalities into one generator. This means the model must learn the Cartesian product of every interaction between every condition type and the corresponding output, leading to a very data-hungry method requiring substantial architectural modifications (e.g., new conditioning inputs, cross-attention layers, etc.). VFM decouples this: the backbone flow map learns the data distribution, while a lightweight noise adapter (only 10M parameters vs. 130M for the flow map) learns to find the appropriate latent noise for each observation. This is more parameter-efficient and modular.
>
> **Training burden is modest:** We also note that the training cost of VFM is minimal compared to conditional training. We improved the training pipeline and now VFM can be fine-tuned over partially trained flow maps for 20 epochs for a short time and still perform the best among all the metrics. This suggests that fine-tuning from stronger checkpoints would be even cheaper. Please see the **[LINK BELOW]** for improved VFM results within significantly reduced training time.
>
> **VFM operates in noise space, not by conditioning the generator.** VFM reformulates the problem in noise space: given an observation $y$, VFM learns a distribution over initial noise vectors $z$ such that mapping $z \mapsto x=f(z)$ produces samples consistent with both the observation and the data prior. The key insight is that the data-space posterior $p(x|y)$ can be highly multimodal and complex, but its corresponding noise-space posterior $p(z|y)$ may be much simpler due to the structure of the noise-to-data mapping. This pull-back to noise space is fundamentally what makes a simple Gaussian adapter sufficient. This perspective is a core conceptual contribution of our work that we believe is valuable for the community.
>
> We agree that our original submission did not sufficiently articulate these advantages, and we will expand the discussion in the updated manuscript to clarify these points.
>
>
> > **[Q2] An example that conditional flow map cannot solve**
>
> A conditional flow map $f(z,c)$ learns a direct map from conditions to samples without reference to any objective evaluating sample consistency with the condition. This makes it unable to handle _reward alignment_, which is a compelling application that VFM solves naturally in a principled way: The goal is to sample from a reward-tilted distribution $p_{reward}(x|c) \propto p_{data}(x) \exp(\lambda R(x, c))$, where $R(x, c)$ is a differentiable reward model (e.g., a human preference score) given image $x$ and context $c$. A conditional flow map has no mechanism to incorporate such a reward signal, as it only learns input-output mappings from paired data. One way could be expensive training from scratch, but it needs data samples from the reward-tilted distribution, which is unrealistic to have access to. On the other hand, VFM is very cheap (0.5 epochs) and only needs a differentiable reward function + any data-context pairs, not necessarily from the tilted distribution. VFM provides a rigorous ELBO-based objective that naturally decomposes into: (i) reward maximization pushing the adapter toward high-reward latent regions, (ii) KL regularization preventing latent collapse, and (iii) a data-fidelity term anchoring the generator to the data manifold. Starting from a pre-trained DMF-XL/2+ model, VFM achieves strong reward alignment in 0.5 epochs (~6 hours), producing high-quality reward-aligned samples in a single forward pass. We train with the HPSv2 reward, but at test time, the fine-tuned model also boosts PickScore and ImageReward consistently, demonstrating genuine alignment rather than reward hacking. We explored different $\lambda$ values (reward strength) and observed consistent improvements across all metrics. To our knowledge, this is the first principled framework for one-step reward-aligned generation with flow maps. Please see the **[LINK BELOW]** for results and metrics.
>
>
> **NEW RESULTS:** [LINK HERE](https://drive.google.com/file/d/1IgYO3D9BPpFTF1Y2CNsq-vDyHuI4kLOG/view?usp=sharing)

---

> > ### Author Rebuttal · Reviewer_Bfde · 2026-04-02
> >
> > Looks good. Changed to weak accept.

---

> > > ### Author Response · Authors · 2026-04-04
> > >
> > > Dear Reviewer Bfde,
> > >
> > > Thank you very much for revisiting your assessment and raising your score. We are glad the reward alignment application and positioning clarifications helped resolve your concerns.
> > >
> > > Since our rebuttal, we have continued strengthening the paper with new results on **non-linear inverse problems (HDR imaging)**, a **halved training cost** through our improved two-phase pipeline, and an **additional recent baseline (LFlow)** that VFM substantially outperforms. ([LINK](https://drive.google.com/file/d/1IgYO3D9BPpFTF1Y2CNsq-vDyHuI4kLOG/view?usp=sharing))
> > >
> > > If there are any remaining questions, we would be very happy to address them.
> > >
> > > Thank you again for your time and the constructive feedback, which has helped us improve the manuscript.
> > >
> > >
> > > Best regards,
> > >
> > > The Authors

---

### Official Review · Reviewer_2QEF · 2026-03-13

**Soundness:** 3
**Presentation:** 3
**Significance:** 3
**Originality:** 3
**Overall Recommendation:** 5
**Confidence:** 4

**Summary:**

This paper presents Variational Flow Maps (VFMs), a framework for solving inverse problems using flow maps, which serve as counterparts to consistency models for flow matching. Instead of guiding the generation path, VFMs focus on finding an observation-guided noise input to the flow matching process, enabling conditional generation in one or a few steps. The authors propose a joint training strategy for the noise adapter and the flow map that involves three terms: (i) data fidelity for the given inverse problem, (ii) a VAE-style KL loss for the noise adapter, and (iii) a mean flow loss. They also amortize the inference across different forward operators. The framework is first benchmarked on a two-dimensional example to evaluate the effect of jointly training the flow map and noise adapter, as well as the structural constraint imposed by the mean flow loss. They show that VFMs successfully capture the bimodal nature of the data. Finally, they validate their framework on image restoration tasks, where VFMs produce highly realistic results compared to other methods, despite lower data fidelity in terms of PSNR.

**Compliance With Llm Reviewing Policy:**

Affirmed.

**Final Justification:**

I believe this work is a valuable contribution to the solution of inverse problems with efficient generative models. Given that the authors would explain the limitations and trade offs of their method, I would keep my recommendation to accept.

**Key Questions For Authors:**

1. Can VFM be extended to handle non-linear inverse problems?
2. The flow matching solvers for inverse problems are evolving fast; there have been many methods that outperform FlowChef and FlowDPS. I would suggest adding some more recent methods to your benchmarks.

**Limitations:**

yes

**Strengths And Weaknesses:**

The paper is well written, and the design choices are well explained, and the 2d ablation example is nice. VFM can open the door to more efficient (in terms of inference time) flow matching inverse solvers.  However, the VFM needs retraining, whereas most of the comparison models for image restoration rely on pre-trained models.

---

> ### Author Rebuttal · Authors · 2026-03-30
>
> We thank the reviewer for their careful reading and for recognizing that VFM "can open the door to more efficient flow matching inverse solvers". We are glad the reviewer found the paper "well written" and the design choices "well explained".
>
> > **[W1] Retraining vs. Guidance**
>
> We agree that VFM requires a one-off fine-tuning phase, unlike purely test-time guidance methods. However, we view this as a favorable trade-off for many practical scenarios:
>
> **Amortized inference cost:** Guidance-based methods must re-run an expensive iterative optimization (250+ NFEs with classifier-free guidance) for every new sample. VFM invests cost once during training and then produces well-calibrated posterior samples instantaneously (~0.015s) at test time. When the application requires generating many conditional samples (e.g., uncertainty quantification, real-time restoration, or interactive editing), then the amortized cost of VFM is substantially lower.
>
> **Training burden is modest:** Since submission, we have also improved the training pipeline, leading to a smaller fine-tuning cost. Concretely, starting from a pre-trained SiT model (80 epochs), we first train with the mean flow loss alone for 30 epochs ($\alpha=0$) to consolidate the one-step transport, and then couple the adapter and flow map with the full VFM loss for only 20 additional epochs ($\alpha=0.8$). This halves the total VFM training cost compared to our original submission without the $\alpha=0$ warmstart  (50 vs. 100 epochs of fine-tuning), produces superior results on all inverse problem benchmarks without the projection trick, and sacrifices only a negligible FID on unconditional generation. This suggests that even a partially trained flow map can be efficiently adapted for conditional tasks; fine-tuning from stronger checkpoints would be even cheaper. Please see the **[LINK BELOW]** for improved VFM results within significantly reduced training time. This suggests that strong flow maps trained for 100’s of epochs can be turned into strong VFM’s within just a few epochs.
>
> **Principled posterior learning:** VFM provides a variational objective (an ELBO-based bound) that learns the full posterior distribution, rather than seeking a single point estimate per query. This is a crucial advantage over guidance methods which approximate the posterior through ad-hoc gradient updates during sampling.
>
> Moreover, our new reward alignment experiments further demonstrate that VFM’s training cost is modest: starting from a pre-trained DMF-XL/2+ model, we achieve strong reward alignment in under 0.5 epochs (~6 hours), enabling one-step sampling from reward-tilted distributions
>
>
> > **[Q1] Non-linear inverse problems**
>
> Yes. The VFM framework is agnostic to the structure of the forward operator A. The observation loss $L_{obs}$ (Eq. 14) evaluates $||y − A(f(z))||^2$ for any differentiable operator A, and no linearity assumption is required for the training procedure. We have conducted two new experiments demonstrating this:
>
> **HDR imaging:** A nonlinear inverse problem where the forward operator simulates sensor saturation, scaling the scene radiance and applying hard clipping to model the irreversible loss of highlight and shadow detail that occurs in real camera pipelines. VFM outperforms all baselines on this task. We fine-tuned our B/2 flow map checkpoint for only 10 additional epochs. Please see the **[LINK BELOW]** for results.
>
> **General reward alignment:** In VFM, the observation loss can be replaced by a differentiable reward function R(x, c) in a principled way, where x is the image input and c is the text prompt. This is highly nonlinear (e.g., HPSv2 [1], a neural network-based human preference scorer). Starting from a pre-trained DMF-XL/2+ model, VFM achieves strong alignment across multiple reward metrics (HPSv2, PickScore, ImageReward) in a single forward pass after only ~10K iterations of fine-tuning (corresponding to only 0.5 epochs within 6 hours of fine-tuning). To our knowledge, this is the first principled and scalable framework for one-step reward-aligned sampling with flow maps. Please see the **[LINK BELOW]** for results and metrics.
>
>
> > **[Q2] Recent baseline method**
>
> We appreciate this suggestion. Our original submission already includes seven competitive baselines, all carefully tuned with exhaustive hyperparameter sweeps on the same SiT-B/2 backbone (see Appendix B.2.2). Following the reviewer’s suggestion, we have additionally re-implemented and benchmarked LFlow [2], a recent guidance-based method for inverse problems. VFM substantially outperforms LFlow across the metrics. We will incorporate this baseline in the camera-ready version. Please see the **[LINK BELOW]** for results.
>
>
>
>
> **NEW RESULTS:** [LINK HERE](https://drive.google.com/file/d/1IgYO3D9BPpFTF1Y2CNsq-vDyHuI4kLOG/view?usp=sharing)
>
> [1] https://arxiv.org/abs/2306.09341
>
> [2] https://arxiv.org/abs/2511.06138

---

> > ### Author Rebuttal · Reviewer_2QEF · 2026-04-02
> >
> > I would like to thank the authors for their explanations in the rebuttal. However, there are a few points I would like to address regarding the response:
> > (1) Retraining and inference time is certainly a trade-off, and whether it is favorable depends on the use case. Such retraining (or fine-tuning) injects information about the forward operator directly into the model. While I acknowledge that you amortize this over a set of forward operators, I would still be surprised if the model can generalize to entirely unseen forward operators (e.g., Fourier sampling) out-of-the-box, unlike methods that solely depend on pretrained models and guidance.
> > (2) There are also strategies to improve the generation path of flow matching methods, for example, with rectified flow matching [1], which is used by the latest versions of Stable Diffusion. The inverse solvers based on these models require significantly fewer steps (e.g., ~40 steps) for solving inverse problems with pretrained models.
> > (3) Additionally, I do not agree with the generalization that all guidance methods are "ad-hoc." Methods like DAPS [2] (cited in your paper), as well as later methods like Flower [3] and FlowLPS [4] for flow matching, do not rely on gradient corrections and instead follow principled ancestral sampling perspectives. Also, many of the gradient-based methods are motivated by theory, of course, up to certain approximations.
> > (4) I appreciate the effort put into ensuring fair comparisons with seven baseline methods and the extensive hyperparameter tuning involved. Thanks for adding the LFlow results also. I believe there is still room to better clarify where VFM stands specifically in terms of image restoration quality. (e.g. see the Table 1 of [4] for a overview of recent improvements of methods over FlowDPS and FlowChef). I understand and value that running these comparisons is very time-consuming. My feedback here is genuinely to help clarify the big picture of how these different methods interact; I do not believe that only state-of-the-art methods should be published, and there is clear merit in your unique perspective. Also I am curious, how much the performance of baselines drop with less steps e.g. if you drop the 250 steps to 100 steps or even 10 steps (I imagine they should do pretty bad with 10 steps unlike your method).
> > (5) The new setup for training you introduced in the rebuttal sounds interesting and is consistent with the findings of [5].
> > Despite the points raised above, I see merit in this work and I maintain my positive acceptance score, as I believe the core contribution is valuable to the community. However, I strongly advise that you more thoroughly discuss these trade-offs and limitations in the newer version of the manuscript.
> > References: [1] https://arxiv.org/abs/2209.03003 [2] https://arxiv.org/abs/2407.01521 [3] https://arxiv.org/abs/2509.26287 [4] https://arxiv.org/abs/2512.07150 [5] https://arxiv.org/abs/2510.20771

---

> > > ### Author Response · Authors · 2026-04-06
> > >
> > > Dear Reviewer 2QEF,
> > >
> > > Thank you for the thoughtful follow-up and for maintaining your positive score. We appreciate the constructive nature of these points, which will help us improve the manuscript. We address each below.
> > >
> > > (1) We fully agree that both paradigms have distinct strengths, and the favorable choice depends on the application. VFM is unlikely to generalize out-of-the-box to entirely unseen forward operators (e.g. Fourier sampling), as you note. However, we emphasize that VFM's primary advantage lies in the amortized regime. It is when one needs to draw many posterior samples for a fixed class of problems (e.g. uncertainty quantification, real-time restoration), the one-time training cost is quickly offset by the ~0.015s per-sample inference. We will discuss this trade-off more explicitly in the camera-ready.
> > >
> > > (2) This is an excellent point. We agree that better-crafted generative priors (e.g. rectified flows) can reduce the step count for guidance-based methods. Importantly, the same improvements benefit VFM as well, stronger pretrained flow maps would serve as better initializations, likely requiring even less fine-tuning while maintaining instantaneous inference. Our current comparisons use the same SiT-B/2 backbone (130M parameters) across all methods to ensure a fair, apples-to-apples evaluation, but we acknowledge that both approaches gain from advances in the underlying generative model. We will clarify this in the manuscript.
> > >
> > > (3) We will revise our language and expand the related work section to properly reflect these distinctions and the papers you have pointed out.
> > >
> > > (4) Thank you for this suggestion. We agree this comparison would strengthen the paper. Due to time constraints during the discussion period, we were unable to run these experiments, but we will include a study of baseline performance degradation at lower step counts (e.g. 100 and 10 steps) in the camera-ready version. We expect this to further highlight VFM's advantage in the low-NFE regime.
> > >
> > > (5) We will expand the manuscript with a thorough discussion of the trade-offs raised above, including limitations, the complementary nature of guidance and amortized approaches, and the settings where each paradigm is most appropriate.
> > >
> > > We sincerely appreciate your engagement throughout the review process. Your suggestions have improved the scope and clarity of our work, and we are glad you see merit in the core contribution of shifting conditional generation to noise space for one-step sampling. All points will be carefully reflected in the final manuscript.
> > >
> > >
> > > Best regards,
> > >
> > > The Authors

---

### Decision · Program_Chairs · 2026-04-30

**Decision:**

Accept (regular)

**Comment:**

This paper proposes Variational Flow Maps (VFM), a framework for one-step conditional generation with flow maps by learning a latent posterior over noise, with applications to inverse problems and reward-aligned sampling.

The reviewers agree the paper is technically sound and well motivated, with strong conceptual framing (reformulating conditioning as latent posterior inference), principled joint training of the flow map and adapter, convincing ablations, and substantial speedup versus iterative guidance. Initial concerns centered on positioning relative to conditional flow maps, limited few-step baselines, calibration analysis depth, and theory beyond linear-Gaussian settings.

The rebuttal effectively addressed these concerns, clarifying the modularity and noise-space perspective, adding LFlow baseline, LPIPS diversity analysis, non-linear inverse problem results (HDR imaging), reduced training cost, and applications to reward alignment. One reviewer maintained reservations about out-of-distribution generalization and comparison scope, but all reviewers ultimately resolved concerns or moved toward acceptance. The discussion converged on a positive consensus. Given this, the recommendation is Accept.